# Vertical bedrock shifts reveal summer water storage in Greenland ice sheet

Jiangjun Ran[1,21✉], Pavel Ditmar[2,21], Michiel R. van den Broeke[3], Lin Liu[4], Roland Klees[2], Shfaqat Abbas Khan[5], Twila Moon[6], Jiancheng Li[7,8,9], Michael Bevis[10], Min Zhong[11], Xavier Fettweis[12], Junguo Liu[13,14], Brice Noël[12], C. K. Shum[10], Jianli Chen[15,16,17], Liming Jiang[18,19] & Tonie van Dam[20]

The Greenland ice sheet (GrIS) is at present the largest single contributor to global-mass-induced sea-level rise, primarily because of Arctic amplification on an increasingly warmer Earth[1–5]. However, the processes of englacial water accumulation, storage and ultimate release remain poorly constrained. Here we show that a noticeable amount of the summertime meltwater mass is temporally buffered along the entire GrIS periphery, peaking in July and gradually reducing thereafter. Our results arise from quantifying the spatiotemporal behaviour of the total mass of water leaving the GrIS by analysing bedrock elastic deformation measured by Global Navigation Satellite System (GNSS) stations. The buffered meltwater causes a subsidence of the bedrock close to GNSS stations of at most approximately 5 mm during the melt season. Regionally, the duration of meltwater storage ranges from 4.5 weeks in the southeast to 9 weeks elsewhere. We also show that the meltwater runoff modelled from regional climate models may contain systematic errors, requiring further scaling of up to about 20% for the warmest years. These results reveal a high potential for GNSS data to constrain poorly known hydrological processes in Greenland, forming the basis for improved projections of future GrIS melt behaviour and the associated sea-level rise[6].

Increased meltwater runoff constitutes the largest contributor (roughly 55%) to post-2000 GrIS mass loss[1–5,7]. En route to the ocean, meltwater may be temporarily stored in surface lakes (supraglacially), inside firn (the layer of compressed snow) or in ice cavities (englacially), at the ice–bedrock interface (subglacially) or as groundwater[8–20] (Fig. 1). Most of this buffered water storage (BWS) is gradually released to the ocean before the onset of the next melt season. BWS affects ice-sheet evolution in several ways. In the interior accumulation zone, liquid water typically percolates into the firn layer, in which it refreezes or recharges firn aquifers. Over semi-impermeable ice in the marginal ablation zone, meltwater enters supraglacial lakes and streams, ultimately draining to the ice sheet–bedrock interface through moulins and crevasses[21–24]. In the subglacial environment, BWS induces high basal water pressure, creating a temporary lubrication effect and ice-flow acceleration, particularly at the beginning of the melt season[25–28]. When the melt season progresses, the accumulation of water creates an efficient subglacial drainage system[29–32], reducing basal water pressure. But these drainage systems and the glacier bed are highly heterogeneous, and high basal water pressure can persist if the drainage system is hydraulically poorly connected to the channels[32].

In situ observations on the GrIS remain too sparse to spatiotemporally resolve the highly heterogeneous (basal) water accumulation and flow[32]. Ice-penetrating radar detects properties indicative of basal water but lacks information on water volume and pressure[33]. Satellite gravimetry quantifies water storage and release but its relatively coarse sampling prevents a regionally resolved assessment of Greenland hydrology[34]. Here we apply a new method to quantify the spatiotemporal evolution of Greenland BWS: continuous monitoring of elastic vertical bedrock displacements by 22 Greenland GNSS Network (GNET) stations (Fig. 2). Elastic bedrock deformation happens instantaneously when mass is redistributed. The accumulation of mass at the surface (for example, growth of the seasonal snow cover) generally leads to

[1]Department of Earth and Space Sciences, Southern University of Science and Technology, Shenzhen, China. [2]Department of Geoscience and Remote Sensing, Delft University of Technology, Delft, The Netherlands. [3]Institute for Marine and Atmospheric Research, Utrecht University, Utrecht, The Netherlands. [4]Department of Earth and Environmental Sciences, Faculty of Science, The Chinese University of Hong Kong, Hong Kong, China. [5]Department of Geodesy and Earth Observation, DTU Space—National Space Institute, Technical University of Denmark, Kongens Lyngby, Denmark. [6]National Snow and Ice Data Center, Cooperative Institute for Research in Environmental Sciences, University of Colorado, Boulder, Boulder, CO, USA. [7]School of Geosciences and Info-Physics, Central South University, Changsha, China. [8]MOE Key Laboratory of Geospace Environment and Geodesy, School of Geodesy and Geomatics, Wuhan University, Wuhan, China. [9]Hubei Luojia Laboratory, Wuhan University, Wuhan, China. [10]Division of Geodetic Science, School of Earth Sciences, Ohio State University, Columbus, OH, USA. [11]School of Geospatial Engineering and Science, Sun Yat-sen University, Zhuhai, China. [12]Department of Geography, University of Liège, Liège, Belgium. [13]Yellow River Research Institute, North China University of Water Resources and Electric Power, Zhengzhou, China. [14]Henan Provincial Key Laboratory of Hydrosphere and Watershed Water Security, North China University of Water Resources and Electric Power, Zhengzhou, China. [15]Department of Land Surveying and Geo-Informatics, The Hong Kong Polytechnic University, Hong Kong, China. [16]Research Institute for Land and Space, The Hong Kong Polytechnic University, Hong Kong, China. [17]Hong Kong Polytechnic University Shenzhen Research Institute, Shenzhen, China. [18]State Key Laboratory of Geodesy and Earth's Dynamics, Innovation Academy for Precision Measurement Science and Technology, Chinese Academy of Sciences, Wuhan, China. [19]College of Earth and Planetary Science, University of Chinese Academy of Sciences, Beijing, China. [20]Department of Geology and Geophysics, College of Mines and Earth Science, University of Utah, Salt Lake City, UT, USA. [21]These authors contributed equally: Jiangjun Ran, Pavel Ditmar. ✉e-mail: ranjj@sustech.edu.cn

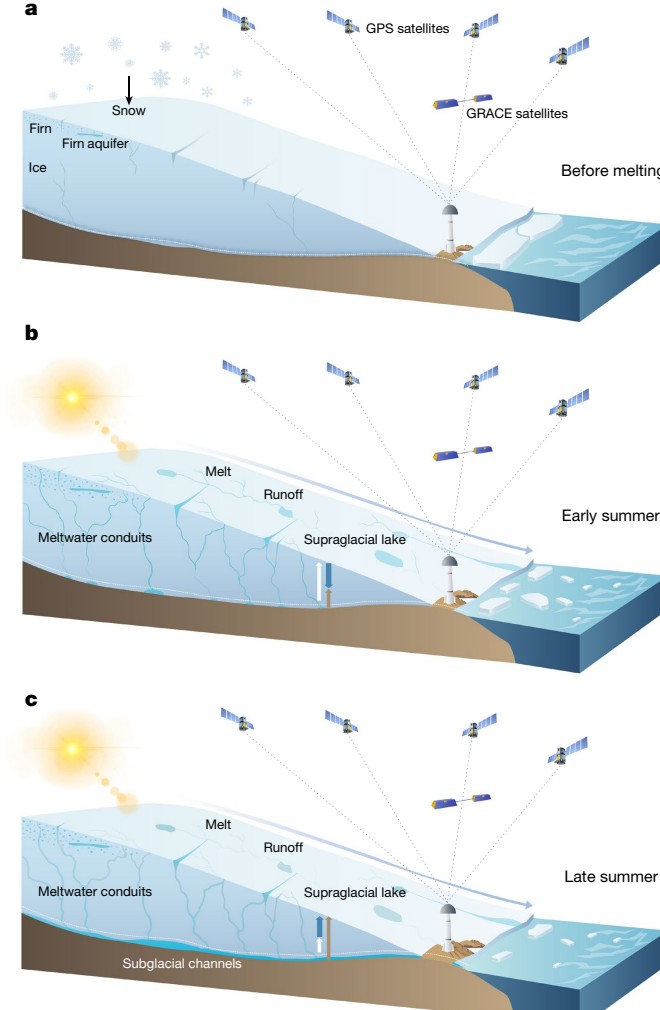

**Fig. 1 | Evolution of water storage and associated bedrock displacements within the GrIS at different stages of the melt season. a**, Ice sheet before melt season. Water mass is minimal; actual vertical position of the bedrock is consistent with that computed on the basis of background models (dashed white line). **b**, Early phase of the melt season. Liquid water rapidly accumulates because discharge into the ocean is minimal, so that the actual vertical position of the bedrock shows only a minor uplift (brown arrow); position of the bedrock based on background models, which do not take the water accumulation into account, is above the true position, showing a rapid uplift (white arrow); the separation between the actual and calculated positions increases, so that the residual bedrock displacement becomes more and more negative (blue arrow directed downwards), reflecting a continuing water accumulation. **c**, Late phase of the melt season. Both accumulated and newly produced water is subject to rapid discharge into the ocean through an efficient system of englacial and subglacial channels; position of the bedrock based on the background models is still above the true position, but the separation between the actual and calculated positions decreases owing to a decreasing water mass, so that the residual bedrock displacement becomes less and less negative (blue arrow directed upwards).

crustal subsidence, whereas mass removal causes crustal uplift. In this way, each GNET station offers quantitative information on regional mass changes in glaciers, ice caps, the ice sheet and aquifers within a roughly 200 km radius (see Methods). Correcting for known nuisance signals yields time series of residual vertical displacements. A notable correction concerns the glacier surface mass balance (SMB), the seasonal accumulation and ablation of snow and ice. For this, SMB models are used, which only account for local, shallow meltwater storage by capillary retention and refreezing in seasonal snow and firn; further

meltwater is assumed to reach the ocean instantly. In reality, BWS causes a marked runoff delay: we expect the increase in BWS in the early melt season to result in a downward residual bedrock displacement, which slowly reduces to zero towards the end of the melt season, as meltwater is gradually released into the ocean.

In this study, we address three questions: (1) how does GrIS BWS evolve during the melt season?; (2) are there spatial variations in the duration of GrIS BWS?; (3) can GNET data be used to improve runoff estimates from SMB models?

## Seasonal cycle and its spatial variations

We produce time series of vertical bedrock displacements ('shifts') at 22 GNSS stations from GNET over the period 2009–2015. We isolate the BWS signal by subtracting SMB-related and other nuisance signals from the total displacements observed (Extended Data Table 1). Mass variations caused by SMB processes are provided by the RACMO2.3p2 regional climate model[35] covering the entire GrIS. Figure 2 shows the mean annual cycles of detrended residual vertical displacements for all GNET stations under consideration. The pattern is similar for all stations: a slow downward motion from February to April (corresponding to the accumulation of stored water), which accelerates in May and peaks in July. In general, residual downward motion corresponds to an accumulation of stored water that is not shallow and/or local, that is, not accounted for in the SMB models, and vice versa. Therefore, we interpret this signal as BWS accumulation within roughly 200 km around the GNSS station starting from the onset of the melt season (Fig. 1), which is unaccounted for in the SMB models. After July, the stations show relatively constant upward motion until February the following year, which is attributed to a gradual reduction of BWS through discharge into the ocean. We conducted a comprehensive analysis, including validation with independent GRACE satellite gravimetry data (Fig. 3), which demonstrates that this signal is real and not an artefact resulting from errors in models or data, or a seasonality of ice discharge (Methods).

The time series also reveal a spatial variability superimposed onto the mean annual cycle of residual vertical displacements (Fig. 2). Details of vertical motion differ among the stations, notably starting from July. The stations in the south and southeast typically show a sharp and quick bedrock uplift after July, evidence of a rapid loss of BWS there. By contrast, most of the remaining stations show a slower BWS loss until September or October and an accelerated loss only later. Finally, many stations show a reduction of BWS loss rate by the end of winter.

## Quantification of BWS

To quantify BWS variations within the GrIS, we propose an analytic function to fit the residual vertical displacement time series (Methods). The function assumes BWS to decay exponentially, with the exponent being inversely proportional to the parameter $T_{st}$, which is termed as 'water storage time' and fitted to the data. This parameter indicates for how long the water is buffered inside the ice sheet during and after the melt season. Extended Data Fig. 1a shows the time series of residual vertical displacements and its approximation with the computed analytic function using the GNET KAGA station as an example.

Notably, the introduced analytic function takes into account possible inaccuracies in the runoff magnitude estimated by the adopted SMB model. It is assumed that the true runoff is related to the modelled runoff by a scaling factor. To determine its value, the modelled runoff time series are scaled with empirical factors calculated per year using nonlinear optimization (Methods). The resulting estimates of water-mass variations are referred to as 'calibrated'. For comparison, we also estimate variations in the water mass without applying this scaling (referred to as 'uncalibrated').

On the basis of the proposed analytic function, we compute time series of the vertical displacements caused by BWS. These estimates

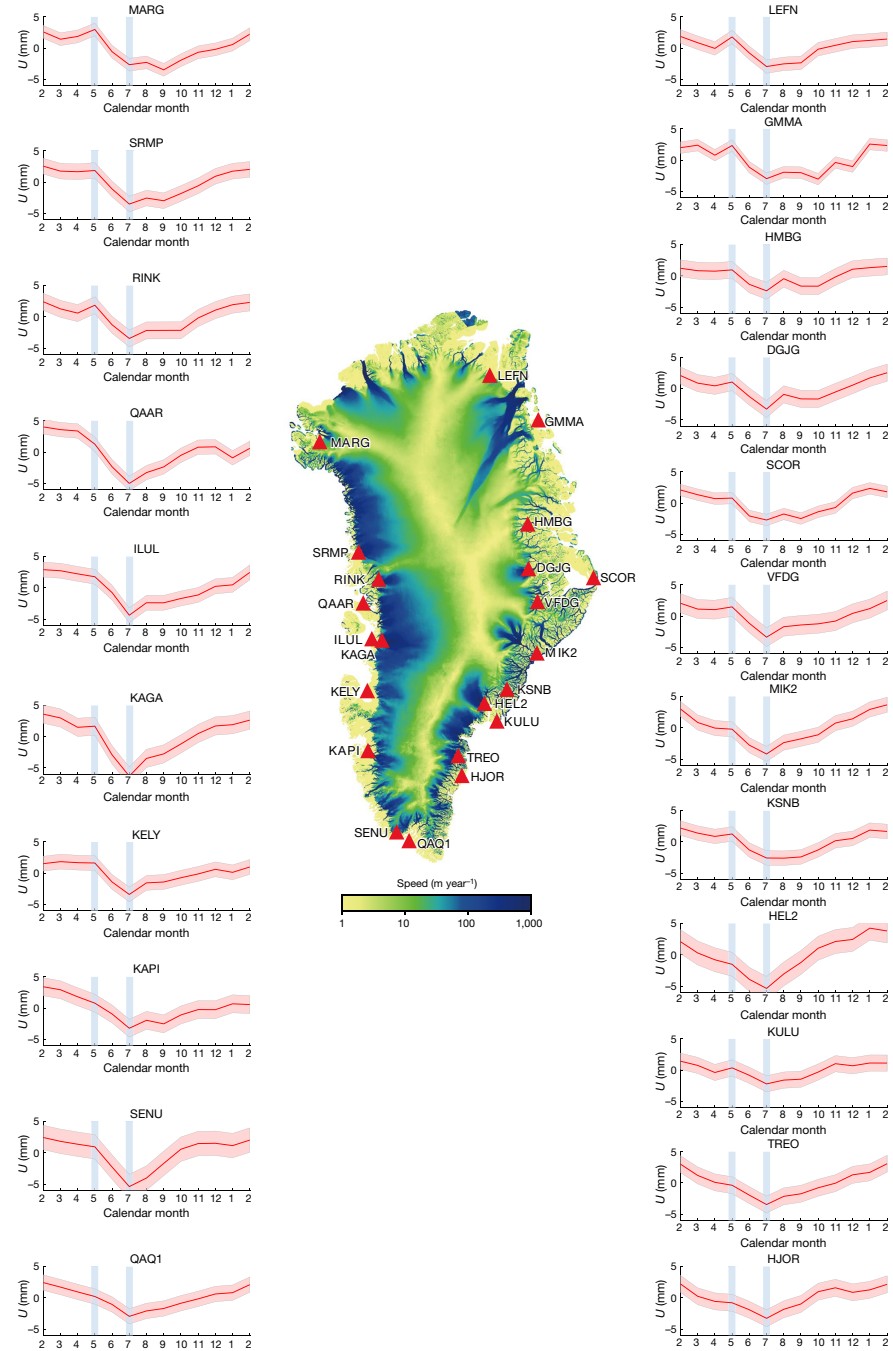

**Fig. 2 | The mean annual cycles of residual vertical displacements at GNET sites.** Note that displacements are obtained after subtracting the loading signals resulting from modelled SMB and non-ice processes (Methods), as well as a subsequent detrending. The red shading depicts the one-sigma uncertainty. Vertical blue lines denote May (as the onset of the melt season) and July (as the month of the peak water storage). The map in the centre shows mean ice-flow velocities during 1985–2018 from NASA's Making Earth System Data Records for Use in Research Environments (MEaSUREs) programme[39]. Extended Data Fig. 3 shows an example of how the residual vertical displacements are computed.

account for variations in the total BWS, that is, water stored in all ice-sheet compartments, including snow/firn, moulins, lakes, basal water storage, as well as groundwater storage below the ice sheet. Taking the KAGA station as an example, we show the displacements based on both calibrated and uncalibrated estimates of BWS variations in Extended Data Fig. 1b. Both time series reveal the largest BWS in 2012, a year of extreme melt in Greenland[2]. For the calibrated BWS estimates, the displacement at the KAGA station reaches 14 mm. Similar features are found in the time series from other GNET stations, particularly those located in southern and southwestern Greenland: from HJOR to QAAR (Fig. 4).

Most of the stations outside the northern part of Greenland show the second largest BWS in 2010, another year of extreme summer melt[36].

## Evaluation of modelled runoff estimates

Comparison between modelled and observed vertical displacements indicates that applying a scaling factor to SMB-modelled runoff is necessary to improve agreement. The calibrated estimates of BWS show larger temporal variations at the KAGA station than the uncalibrated ones (Extended Data Fig. 1b): for example, the calibrated estimate in

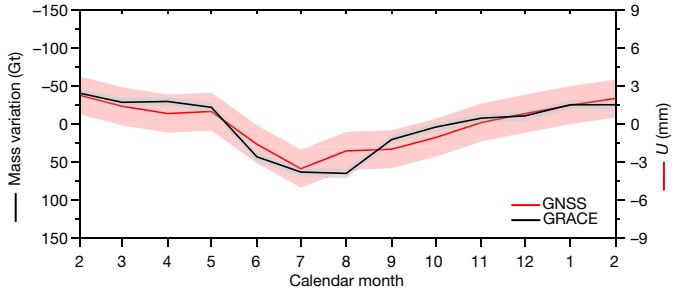

**Fig. 3 | Comparison of mean seasonal cycles of two phenomena related to BWS in Greenland.** Vertical displacements from GNSS are shown in red and mass anomalies from GRACE are shown in black. The monthly vertical displacements represent the 22 GNET station mean values. The monthly mass anomalies are a Greenland-wide integration of water-related mass anomalies (see Methods). The time interval under consideration is January 2009–December 2015. The shading shows 68% confidence intervals. The GRACE scale is inverted for clarity.

temperatures by showing a notable ($R = 0.42$) correlation between the scaling factors and the mean annual summer temperature anomalies for each GNSS station using ERA5 (ref. 37) (Extended Data Fig. 2).

## Water storage times

The estimated water storage times $T_{st}$ are between 3 and 13 weeks for most stations, with a 55-day (roughly 8-week) average (Fig. 6 and Extended Data Table 3). In northeastern Greenland (stations from LEFN to VFDG), the water storage time is slightly above the average: $64 \pm 16$ days (that is, about 9 weeks). Western Greenland (stations from KAPI to SRMP) is characterized, on average, by the same water storage time, but the station-to-station variations are larger ($64 \pm 20$ days). In the south and southeast regions (stations from MIK2 to SENU), the average water storage time is halved: $31 \pm 12$ days. The longest water storage time (129 days) is observed at station MARG in the extreme northwest of the GrIS.

## Discussion

Our study reveals new insights into the spatial and temporal variability of Greenland water storage within the GrIS. GNET GNSS data are used as a new source of valuable information on BWS within the GrIS. The results show that, across the GrIS ablation zone, the BWS reaches its maximum in July and gradually decreases thereafter. Furthermore, the vertical displacements clearly show interannual variations in BWS. For instance, it is particularly large in high-melt summers, such as those of 2010 and 2012.

The BWS after calibration shows larger interannual variations than modelled runoff from the regional climate model RACMO2.3p2 suggest. We quantify this further by a noticeable correlation ($R = 0.42$) between the estimated scaling factors and summer temperature anomalies (Extended Data Fig. 2d). In other words, the runoff scaling factors for the high-melt (warmer) years, when the BWS is large, are larger than for low-melt (colder) years. For the years with highest summer temperatures (for example, 2012), the upscaling may reach about 20%. This can

the extraordinarily warm 2012 summer is larger than the uncalibrated estimate by about 20%. This implies that the scaling factors are relatively large during high-melt summers. Analysis of the other GNET stations supports this conclusion: scaling factors in warm years (2010 and 2012) are typically larger than in 'ordinary' melt years (2011 and 2014), to say nothing about relatively low-melt years (2013 and 2015); see Fig. 5 and Extended Data Table 2 (year 2009 is not considered as it represents the 'initialization' year (Methods)). This difference between low-melt and high-melt summers is particularly pronounced for northern and northeastern stations. It is also notable that differences in average scaling factor within a single year but for different GrIS regions are typically smaller than year-to-year scaling-factor variations (Extended Data Table 2). This is in spite of the fact that the scaling factors are estimated for each GNET station independently. This finding demonstrates the robustness of the scaling factors (particularly if averaged over a sufficiently large GrIS region). We confirm the scaling factor dependence on summer

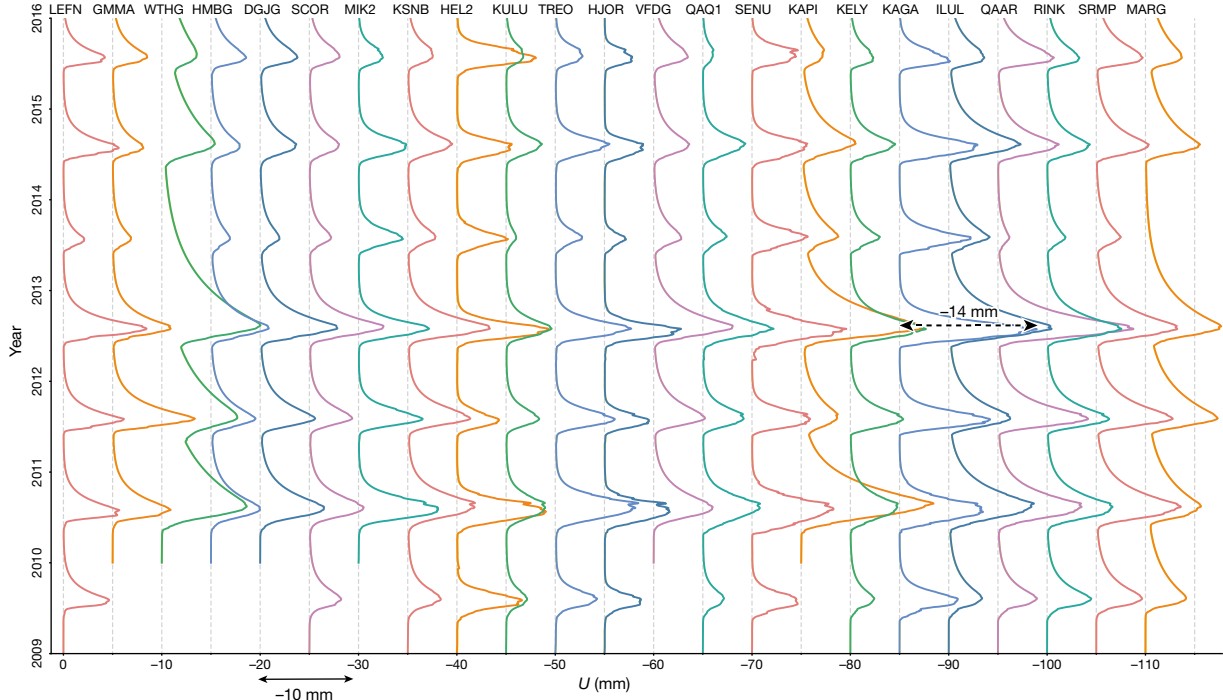

**Fig. 4 | Water-related vertical displacements for all GNET stations studied.** Vertical displacement time series from the proposed analytic model, based on calibrated BWS estimates.

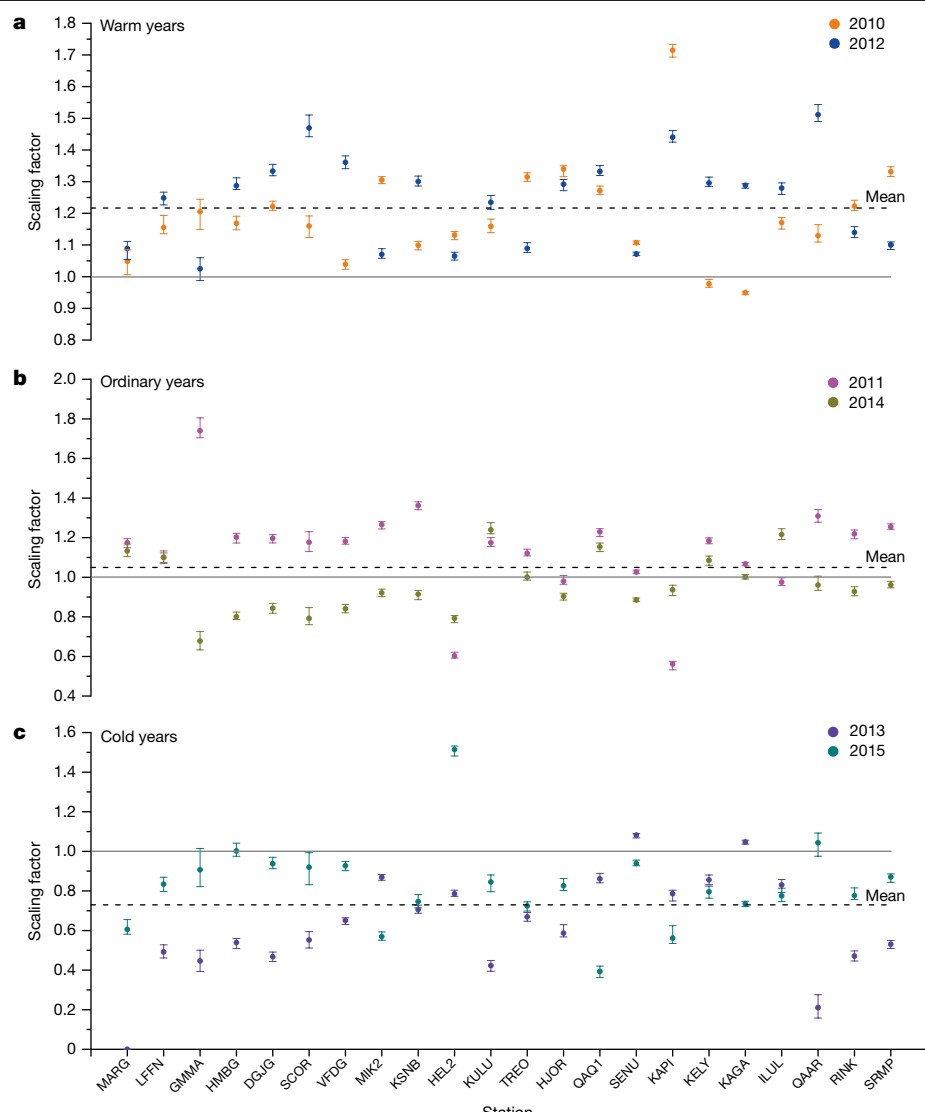

**Fig. 5 | Scaling of SMB model runoff estimates.** Scaling factors for warm (**a**), ordinary (**b**) and cold (**c**) years, to be applied to the SMB model runoff estimates to achieve a best match with GNET station observed vertical displacements. Error bars are defined on the basis of the 1st and the 3rd quartiles as the lower and upper bounds, respectively, from 100 Monte Carlo runs.

be interpreted as evidence that the adopted regional climate model either underestimates melting or overestimates water retention in 'warm' years (or both). The latter might be explained by an unaccounted or too slow modelled firn degradation in 'warmer' years, which reduces the fraction of produced liquid water that can be retained in the firn layer. As a result, the actual runoff in 'warmer' years is higher than that of the regional climate model. Such an interpretation may also explain the relatively poor correlation between the estimated scaling factors and summer temperature anomalies. Strong firn degradation in a 'warm' year probably has a long-term impact: it may affect water retention not only in that year but also in the years to follow, independently of their summer temperatures.

GNET data thus offer a new method to improve GrIS SMB estimates from regional climate models. Those models are at present the tool of choice for estimating GrIS-integrated surface melt rate. Despite their generally good and consistent performance, considerable uncertainties remain in the modelled melt products[38]. Having independent estimates of adjustments required to improve/calibrate the melt products from regional climate models, as provided in this study, is therefore highly valuable for the Greenland mass balance research community. Among others, regional climate models require adjustments in this way for

abnormally warm summers. This is particularly relevant in view of projected Arctic warming[1-5]. Extremely high summer temperatures today may become normal in the foreseeable future. Thus, good model performance for warmer years is critical to project ice-sheet behaviour and associated sea-level rise across coming decades.

GNET data also allow us to quantify BWS on seasonal timescales. We found that the GrIS average water storage time is about 8 weeks, although with important spatial variations. In the northeastern and western regions, it is slightly above the average (about 9 weeks). In the southeastern GrIS, on the other hand, water storage time is relatively short, implying that the hydrological regimes are regionally different. Most probably, this is because the southeastern GrIS is characterized by high accumulation rates, steep topographic relief, a relatively narrow ablation zone and short distances from surface melt locations to the ocean. This results in—on average—rapid drainage, despite widespread occurrence of firn aquifers in this region. Further investigations are needed to shed more light onto the observed differences in GrIS hydrological regimes.

To conclude, our study demonstrates how spatiotemporal variations in BWS within the GrIS can be sensed with GNSS-based vertical displacement data, which offer a higher spatial and temporal resolution than

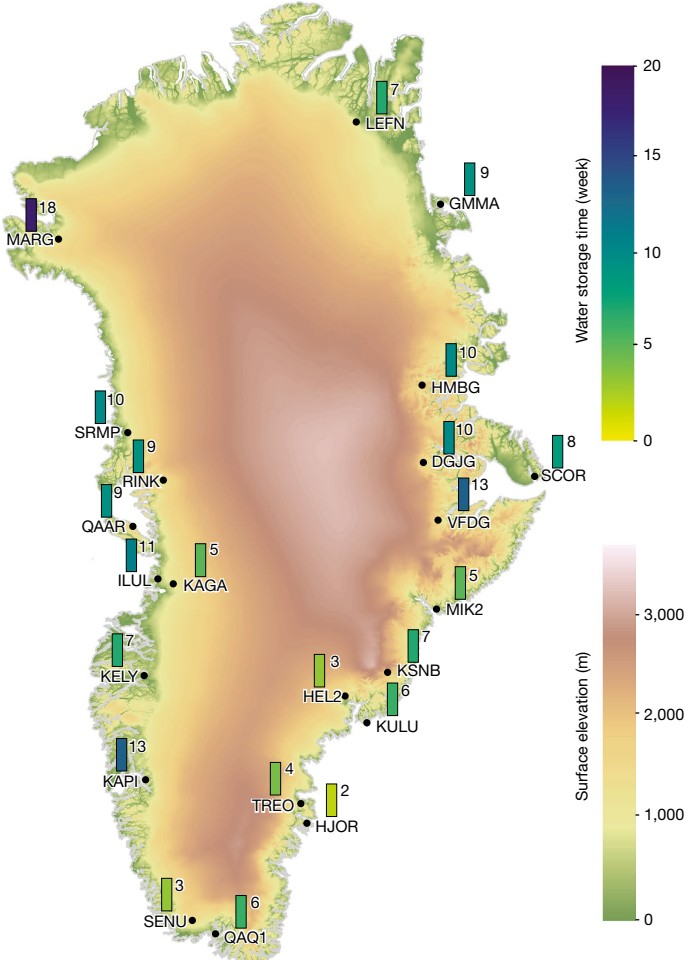

**Fig. 6 | Water storage times.** Water storage time at each GNET station estimated from the proposed analytic model. Ice thickness from MEaSUREs is shown as a base map.

satellite gravimetry. This opens the door for wider use of GNSS data for observation and better understanding of hydrological processes within the GrIS and other ice bodies on Earth. This will be particularly important for an accurate forecasting of the future behaviour of the GrIS and other ice bodies, as well as the associated sea-level rise.

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

## Methods

### GNSS data selection and preprocessing

The GNET consists of about 54 sites, most of which are located along the coast. Here we consider a subset comprising 22 stations that are mounted on bedrock and located close to outlet glaciers. GNSS stations on bedrock are sensitive to all of the processes that cause mass changes on/in/under the ice close to the station. A few stations (for example, UPVK, NUUK and so on) are excluded because they suffer from large mass variation signals from the ocean. In particular, this concerns stations in the northwestern and southeastern parts of Greenland.

We use coordinates and uncertainties from the GNSS daily solutions released by Technical University of Denmark, which are computed with the scheme proposed in ref. 40 in the International Global Navigation Satellite System Service 2014 frame, after removing tidal deformations related to the solid Earth and oceans. The vertical displacements observed at GNET sites include not only the loading signals caused by the BWS changes but also several nuisance signals. The latter represent the crustal deformation caused by atmospheric pressure loading (ATM), land water storage (LWS) on ice-free land, non-tidal ocean loading (NTOL), SMB-related signals and thermal elastic expansion (TEM). To isolate variations in the BWS loading, we correct the observed vertical displacements for the nuisance signals using background models (Extended Data Table 1). The ATM, LWS and NTOL loading changes were computed at University of Luxembourg with the scheme proposed in ref. 41. The SMB contribution was computed by the time integration of signals from the RACMO2.3p2 model[35] for the entire ice sheet. The set of SMB signals consists of precipitation, runoff, sublimation and snow drift components. To compute the vertical displacements at the GNET sites caused by a given component, we used the Green's function method[42,43]. Bedrock deformations are also caused by thermal expansion owing to temperature variations. In this study, we account for this process by using the output from ref. 44, which is computed from the ERA5 model with a finite element method. Subtraction of all these nuisance signals results in a time series of residual vertical displacements, which is the input for the further analysis.

As an example, Extended Data Fig. 3 shows the subtracted signals and residual vertical displacements for station KAGA located near Jakobshavn Isbræ (Greenlandic: Sermeq Kujalleq), one of the largest GrIS outlet glaciers. SMB and ATM signals are the largest among those subtracted[43].

### Uncertainties of the mean annual cycle of residual vertical displacements

Next, we estimate uncertainties of the mean annual cycle of residual vertical displacements, which is shown in Fig. 2. We identify various nuisance signals and categorize them into those that can be completely neglected (for example, ice discharge, lakes) and those that need to be included in the uncertainty analysis. Note that we consider separately shallow groundwater in tundra areas and deep groundwater beneath the ice sheet, as detailed in the 'Contribution of groundwater storage in Greenland' section. To that end, we compare the 'primary' time series computed as explained above with a family of alternative ones. To produce the latter time series, we replace either the input GNSS data themselves or one of the background models of nuisance signals with one or two alternative ones. The alternative set of input GNSS data was provided by the Nevada Geodetic Laboratory at the University of Nevada, Reno[45]. The alternative background models[44–48] are listed in Extended Data Table 1, along with the primary ones. The resulting ATM and NTOL loading signals were downloaded from the International Mass Loading Service (IMLS)[46]. Alternative SMB loading signals were computed in house at DTU Space on the basis of the alternative SMB model (MAR) in the same way as in the case of the primary one. The alternative time series of bedrock deformations owing to thermal expansion were also computed in ref. 44 from ERA5 but with a harmonic method.

Noise in the input GNSS data and in each of the nuisance signals is quantified by computing the standard deviation between the 'primary' time series and the corresponding alternative one[34,49] (see the caption of Extended Data Fig. 4). The total uncertainty of the mean annual cycle of residual vertical displacements is computed as the root sum square of the standard deviations of noise from all possible sources considered in this study, that is, GNSS data and models of ATM, NTOL, LWS, SMB, TEM and GWS signals.

Notably, the GWS signal in tundra areas is not a part of the SMB models (SMB is only defined over glacial ice). Therefore, this signal is not subtracted from GNSS data when residual vertical displacements are computed. To quantify the impact of that signal, we use hydrological models. We consider the difference between mean annual vertical displacements per calendar month with and without groundwater signal subtracted using PCR-GLOBWB and WGHM models. This basically results in the vertical displacements owing to modelled groundwater signal alone. We show that the root mean square signals computed from the PCR-GLOBWB and WGHM models are 0.25 mm and 0.02 mm, respectively, which is small compared with the BWS signal (see Extended Data Fig. 4).

A comparison of the total uncertainty computed for each station with the signal revealed in the mean annual cycle of residual vertical displacements demonstrates that the latter far exceeds the noise level (see Fig. 2) and, therefore, cannot be explained by inaccuracies in the input data or exploited models. Furthermore, we have chosen the KAGA station as the representative one to demonstrate that the use of alternative models or GNSS data leads to only minor changes in the observed annual cycle of residual vertical displacements (Extended Data Fig. 4).

### Contribution of ice discharge

To obtain an upper bound of the contribution of seasonal ice discharge variations to the observed vertical displacements, we use Jakobshavn Isbræ because it shows the largest ice flow velocity seasonality among GrIS outlet glaciers[50]. Note that, although the total change of mass owing to discharge is comparable with that resulting from SMB, its seasonal variability is much smaller[51]. We used in our analysis a time series of ice-discharge-related time series of mass anomalies obtained in ref. 52. That time series was obtained by combining two datasets: (1) monthly ice discharge at the Jakobshavn Isbræ flux gates (Extended Data Fig. 5a) computed from monthly ice velocities[53] and ice thickness values[54]; (2) annual rates of mass-variation change from altimetry and SMB-based vertical displacements. Corresponding mass anomalies were obtained by weighting the latter annual patterns with the monthly ice-discharge estimates. Finally, the solid-rock vertical displacements were derived by means of the Green's function method[42,43], for the location of KAGA station, which is only about 1 km from the 2015 Jakobshavn Isbræ calving front. We see that the mean contribution of Jakobshavn Isbræ ice discharge to vertical displacements does not exceed 1 mm (Extended Data Fig. 5), which is small compared with the magnitude of residual vertical displacements.

### Contribution of lakes

There are many lakes in the Arctic region, none of which were considered in the LWS products used in this study. To further quantify the potential water-mass impact of lakes onto GNSS loading signal in Greenland, we consider three types of lake: lakes in the pan-Arctic region (north of 60° N) in general; supraglacial lakes (SGL) on the GrIS; and proglacial lakes in the coastal part of Greenland.

**Lakes in pan-Arctic region.** Specifically, the monthly water storage data of some large lakes were directly downloaded from a recently published dataset[55]. Also, we used the intra-annual lake-level datasets from three portals: Hydroweb (https://hydroweb.theia-land.fr/), Global Reservoirs and Lakes Monitor (G-REALM; https://ipad.fas.usda.gov/cropexplorer/global_reservoir/) and Database for Hydrological Time

Series of Inland Waters (DAHITI; https://dahiti.dgfi.tum.de/en/), respectively[56–58]. For months without lake area data in the dataset provided in ref. 55, we follow the empirical relationship between lake water level and lake surface area from a similar previous study[59] by interpolating the missing lake-area data with sampler-level data and converting the lake area and level data to water storage changes.

It is worth mentioning that the considered lakes (38 in total, indicated by blue circles in Extended Data Fig. 6a) are typically large, with 33 of them having a surface area greater than 500 km$^2$, accounting for 61.6% of the total surface area of Arctic lakes extracted in ref. 55. For the remaining smaller inland lakes[60] (red circles in Extended Data Fig. 6a), the area data are very limited. Therefore, we use a scale-up strategy[59] to estimate their impact on mass loading.

Taking KAGA station as an example, we find that the vertical displacement induced by the load of Arctic lake water storage during the study period (2009–2015) ranges from −0.10 to 0.04 mm (Extended Data Fig. 7a). This magnitude is sufficiently small to allow for neglecting the potential influence of these lakes.

**SGL on the GrIS.** We also assessed the vertical displacement caused by water mass loading from SGL on the GrIS. First, we divided the GrIS into 10 × 10-km equal-area cells and identified all the cells with SGL using approximately 300,000 high-spatiotemporal-resolution optical images from Sentinel-2 and Landsat 8/9 over 2017–2022, as shown in Extended Data Fig. 6b. Then, the monthly SGL area changes during the melting season (that is, May–September) were derived. Note that it is problematic to obtain the monthly SGL area change for the entire GrIS before the launch of Sentinel-2B satellites in 2017 owing to a high cloud contamination. We assumed, however, that the SGL area changes over the 2009–2015 interval were similar to those over the 2017–2022 interval. When converting the SGL area changes to mass changes, we assumed that the maximum depth of GrIS supraglacial lakes is around 8.5 m (refs. 61,62), which allowed us to estimate the upper limit of water mass changes (Extended Data Fig. 7b). The result shows that, even for the maximal depth of 8.5 m, the magnitude of loading signal caused by SGL mass changes is about 0.3 mm. Thus, supraglacial lakes provide only a minor contribution to the total BWS signal.

**Proglacial lakes in Greenland.** Similar to SGL, we also consider the impact of proglacial lakes in Greenland using the HydroLAKES database (Extended Data Fig. 6c). In total, 2,687 proglacial lakes are taken into account. Most of the proglacial lakes are smaller than 5 km$^2$, whereas the largest one could reach roughly 100 km$^2$. The loading signal caused by proglacial lakes is small. At the KAGA station, for instance, it is on the order of only 0.02 mm (Extended Data Fig. 7c).

### Contribution of groundwater storage in Greenland

In the context of the impact from groundwater storage[63] in Greenland on the loading signals, we distinguish two types of groundwater:
1. Shallow groundwater in tundra areas, which results from snow melting and rainfalls there. In principle, this is an ordinary component of the terrestrial water storage, which is described by various hydrological models, including those addressed in the manuscript (PCR-GLOBWB and WGHM); see Extended Data Fig. 7d,e. According to M. Bierkens, who is a developer of PCR-GLOBWB, it is very difficult to model the groundwater accurately in this region, but the model outcome is enough for a first-order estimate of its magnitude (personal communication, 2024). In this study, we analyse the uncertainty of groundwater estimates in the 'Uncertainties of the mean annual cycle of residual vertical displacements' section.
2. Deep groundwater below the ice sheet (its presence was detected by a hydrological well down to the depth of hundreds of metres[63]). It is a product of ice sheet melting (both at the surface and at the ice sheet base). To the best of our knowledge, little is known about variations in the deep groundwater mass. Here we consider the signal

from deep groundwater as a part of the total BWS signal we detect in GNSS data. Unfortunately, elastic loading data do not allow the deep groundwater to be separated from the rest of the BWS.

### Validation of the results using GRACE data

To validate the results based on elastic loading data, we compared them with water mass changes extracted from satellite gravimetry data. We used four GRACE-based mascon data products. Three of them are off-the-shelf products that were released by: (1) the Center for Space Research (CSR RL06 v02) of the University of Texas at Austin[64]; (2) the Jet Propulsion Laboratory (JPL RL06 v02)[65]; and (3) the Goddard Space Flight Center (GSFC RL06 v1.0)[66–68]. The fourth is the mascon product computed in house[34,69,70]. We corrected the time series for glacial isostatic adjustment using the model in ref. 71, subtracted the SMB signal and detrended the results. Both the GRACE-based monthly estimates of BWS and the monthly water-related elastic displacements were averaged over entire Greenland. On this basis, BWS mean seasonal cycles were obtained. The mean of the four GRACE-based mean seasonal cycles, as well as the mean seasonal cycles of elastic displacements, are shown in Fig. 3 for entire Greenland (the GRACE scale is inverted for clarity). The standard deviations $\delta_{GRACE}$ of the GRACE-based estimates were computed as

$$\delta_{GRACE} = \frac{1}{2}\sqrt{\frac{d_1^2 + d_2^2 + d_3^2 + d_4^2}{3}}, \tag{1}$$

in which $d_i$ (with $i$ = 1, 2, 3, 4) represents the root mean square difference between the estimates based on the $i$th variant of the mascon data product and the mean ones. The factor 1/2 is present because of the fact that we address the error in the mean of the four time series, rather than errors in the individual ones.

The seasonal cycles based on GRACE and GNSS data are remarkably similar. In particular, both datasets show a mass increase from May to July/August, with a subsequent mass loss until February the following year. Minor differences between the GRACE-based and GNSS-based results can be explained by random errors and the different spatial resolutions of these two data types. We interpret the revealed similarity as a confirmation that both types of data show the signal of the same origin: an accumulation and release of water within the GrIS.

### Analytic model of the BWS signal in GNSS elastic loading data

In this section, we explain the analytic function proposed to describe the BWS signal in the residual vertical displacement data. At most of the GNSS stations, a prominent signal in the residual vertical displacements is an upward trend, which reflects a slow mass loss caused by ice discharge. Superimposed to this slow mass loss, there is a seasonal BWS signal at many GNSS sites, which peaks in the middle of the summer (Fig. 2). We hypothesize that this signal is because of BWS. Our interpretation stems from the fact that buffered water, which is not refrozen in place, is not a part of SMB and, therefore, is not described by SMB models.

The total mass balance for grid cell $j$ of an SMB model can be represented as[34]:

$$\frac{dM^{(j)}(t)}{dt} = -D^{(j)}(t) + B^{(j)}(t) + \frac{dS^{(j)}(t)}{dt}, \tag{2}$$

in which $M^{(j)}(t)$ is the total mass, $D^{(j)}(t)$ is ice discharge, $B^{(j)}(t)$ is SMB and $S^{(j)}(t)$ is the BWS. In RACMO2.3p2, SMB is computed as a combination of four components[2]: SMB = P − SU − ER − R, in which P is total precipitative flux (sum of snowfall and rainfall), SU is sublimation, ER is erosion of snow by divergence of the drifting snow transport and R is runoff.

Figure 2 shows that, at many stations, the trend after the end of the melt season does not appear as a continuation of the trend observed before the melt season. This implies that the net change in the ice mass

during the melt season is different from what could be expected if ice discharge were the only cause of that mass change (at least, under the assumption that the ice discharge is constant over the considered time period). Because the effect of temporal variations in ice discharge is probably minor (Fig. 3), we believe that there must be another explanation for this mismatch. We hypothesize that it can be explained by a difference between the mass loss owing to actual water runoff $R$ and runoff $R_0$ from the SMB model. We assume that the true runoff, $R$, is related to the modelled runoff, $R_0$, as

$$R = (1 + \epsilon_k)R_0, \tag{3}$$

in which factor $\epsilon_k$ accounts for errors in the modelled runoff and is estimated per year. This factor is assumed to be spatially invariant in the vicinity of a given GNSS station. Then, the true SMB can be represented as SMB = P − SU − ER − R = P − SU − ER − $(1 + \epsilon_k)R_0$ = SMB$_0$ − $\epsilon_k R_0$, in which SMB$_0$ = P − SU − ER − $R_0$ is the modelled SMB. For a cell $j$, the true SMB contribution $B^{(j)}(t)$ to the total mass balance can be written as

$$B^{(j)}(t) = B_0^{(j)}(t) - \epsilon_k R_0^{(j)}(t), \tag{4}$$

so that the mass balance equation given by equation (2) can be rewritten as:

$$\frac{dM^{(j)}(t)}{dt} = -D^{(j)}(t) + B_0^{(j)}(t) - \epsilon_k R_0^{(j)}(t) + \frac{dS^{(j)}(t)}{dt}. \tag{5}$$

The integration of equation (5) over time yields the mass at a given time $t$ relative to the mass at the initial epoch $t_0$. Let us assume for simplicity that $t_0$ coincides with the beginning of a calendar year. Then, the result of the integration is mass variation $M^{(j)}(t)$, which is defined under the assumption that $M^{(j)}(t_0) = 0$:

$$M^{(j)}(t) = -\int_{t_0}^{t} D^{(j)}(\tau)d\tau + \int_{t_0}^{t} B_0^{(j)}(\tau)d\tau$$
$$- \sum_{k=1}^{K(t)} \epsilon_k \int_{t_{k_0}}^{t_{k_e}} R_0^{(j)}(\tau)d\tau + S^{(j)}(t) - S^{(j)}(t_0), \tag{6}$$

in which $K(t)$ is the number of the year containing the current time $t$, $t_{k_0}$ is the time at the beginning of the $k$th year and $t_{k_e}$ is either the time at the end of the $k$th year (if $k < K$) or the current time $t$ (if $k = K$). Notice that the integration of the term $\epsilon_k R_0^{(j)}(t)$ over time implies that the modelled runoff is integrated separately in each year; the results are scaled with factors $\epsilon_k$ and summed over all years (up to the year containing the current time $t$).

We assume that the contribution of the ice discharge does not change over time: $-D^{(j)}(t) = A^{(j)}$. Then, equation (6) can be simplified as:

$$M^{(j)}(t) = C^{(j)} + A^{(j)}t + \int_{t_0}^{t} B_0^{(j)}(\tau)d\tau - \sum_{k=1}^{K(t)} \epsilon_k \int_{t_{k_0}}^{t_{k_e}} R_0^{(j)}(\tau)d\tau + S^{(j)}(t), \tag{7}$$

in which $C^{(j)} = A^{(j)}t_0 - S^{(j)}(t_0)$. The mass variations given by equation (7) result in vertical elastic deformations of the solid Earth. Assuming a linear relationship between mass variations and elastic deformations, we can rewrite equation (7) in terms of vertical elastic deformations at the location of a GNSS station:

$$m(t) = c + at + \int_{t_0}^{t} b_0(\tau)d\tau - \sum_{k=1}^{K(t)} \epsilon_k \int_{t_{k_0}}^{t_{k_e}} r_0(\tau)d\tau + s(t), \tag{8}$$

in which different signals in the time series of vertical elastic deformations (denoted with lowercase letters) are associated with the corresponding mass signals (denoted with capital letters in equation (7) and before). Technical details of the transformation of surface mass load

into elastic vertical deformations can be found in the section below entitled 'Computation of elastic vertical deformations and spatial sensitivity of GNSS loading data'. The left-hand side of equation (8) contains the residual GNSS measurements before the correction for the SMB signal. On the right-hand side, we see, among others, the elastic deformations associated with the SMB model, $(\int_{t_0}^{t} b_0(\tau)d\tau)$, and the elastic deformations associated with the computed runoff $(\int_{t_{k_0}}^{t_{k_e}} r_0(\tau)d\tau)$. Both signals are computed on the basis of the RACMO2.3p2 model output. The unknown constant factors $a$, $c$ and $\epsilon_k$ can be estimated using least squares from the observed time series $m(t)$. The only term that requires a further discussion is the signal $s(t)$ associated with the BWS.

Let us consider the total BWS $S(t)$ in the drainage basins located around the current GNSS station (more specifically, in the drainage basins that substantially affect the elastic deformations at the current GNSS station). Temporal variations of that BWS mass are equal to the difference between the total runoff $R(t)$ in those drainage basins (which describes the rate of liquid water production) and the rate $Q(t)$ (which describes discharge of water from the drainage basin into the ocean):

$$\frac{dS(t)}{dt} = R(t) - Q(t). \tag{9}$$

Because transport of water from the location of production to the location of discharge can take weeks or even months, the BWS $S(t)$ can be substantial. Let us assume that the discharge into the ocean is proportional to the BWS (that is, drainable water storage), which is a commonly used assumption in hydrology[72]:

$$Q(t) = \beta S(t), \tag{10}$$

in which $\beta$ is a certain constant proportionality coefficient. The substitution of this expression into equation (9) yields:

$$\frac{dS(t)}{dt} = R(t) - \beta S(t) \tag{11}$$

or

$$\frac{dS(t)}{dt} + \frac{1}{T_{st}}S(t) = R(t), \tag{12}$$

in which $T_{st} = \beta^{-1}$. Assuming that the runoff $R(t)$ is given, we can readily find the solution of the differential equation given by equation (12) as

$$S(t) = S(t_0)e^{-\frac{t-t_0}{T_{st}}} + \int_{t_0}^{t} R(\tau)e^{-\frac{t-\tau}{T_{st}}}d\tau. \tag{13}$$

From this expression, it follows that the parameter $T_{st}$ can be interpreted as the characteristic time of BWS. We refer to it as the 'water storage time'.

The first term in equation (13) represents the impact of the initial BWS, $S(t_0)$. Because $t_0$ is assumed to coincide with the beginning of a calendar year (here year 2009), we set $S(t_0) = 0$ (to minimize the impact of this assumption, we ignore year 2009 in the subsequent analysis as an 'initialization' year). Then, we obtain:

$$S(t) = \int_{t_0}^{t} R(\tau)e^{-\frac{t-\tau}{T_{st}}}d\tau. \tag{14}$$

Equation (14) allows us to introduce an approximate relationship between runoff and BWS in terms of loading signals.

If the spatial pattern of runoff was similar to the spatial pattern of BWS, equation (14) could be rewritten in terms of the loading signal directly (as it was already done when equation (8) was introduced). In practice, of course, this is not the case. Notably, vertical deformation

reduces as the distance between the GNSS station and the location of the surface load increases[73]. Therefore, the buffered water signal measured at a GNSS station can show systematic deviations from the values predicted on the basis of the runoff in line with equation (14). To take this effect into account, we introduce an empirical scaling factor $\theta$ per GNSS station, so that an approximate relationship between runoff and BWS in terms of loading signals is:

$$s(t) = \theta \int_{t_0}^{t} r(\tau) e^{-\frac{t-\tau}{T_{st}}} d\tau. \tag{15}$$

Taking into account that the true runoff is defined as a scaled variant of the modelled one (see equation (3)), we can rewrite the expression above as:

$$s(t) = \theta \sum_{k=1}^{K(t)} (1 + \epsilon_k) \int_{t_{k_0}}^{t_{k_e}} r_0(\tau) e^{-\frac{t-\tau}{T_{st}}} d\tau. \tag{16}$$

After the substitution of this equation into equation (8) and the isolation of the known terms on the right-hand side, we finally obtain:

$$c + at - \sum_{k=1}^{K(t)} \epsilon_k \int_{t_{k_0}}^{t_{k_e}} r_0(\tau) d\tau + \theta \sum_{k=1}^{K(t)} (1 + \epsilon_k) \int_{t_{k_0}}^{t_{k_e}} r_0(\tau) e^{-\frac{t-\tau}{T_{st}}} d\tau$$
$$= m(t) - \int_{t_0}^{t} b_0(\tau) d\tau \tag{17}$$

By considering this equation for all times $t$ within the interval under consideration, we can form a system of nonlinear equations containing $n + 4$ unknown parameters per GNSS station: $c$, $a$, $\theta$, $T_{st}$ and $\epsilon_k$ ($k = 1,...,n$), in which $n$ is the number of years in the considered time interval.

To estimate all the unknown parameters, an iterative least-squares adjustment could be directly applied. In the course of a preliminary study, we realized, however, that there is a trade-off between water storage time $T_{st}$ and the mean value of the corrections $\epsilon_k$. Each of the two can be used to explain signals in the input data, whereas an attempt to estimate them simultaneously frequently results in unphysical estimates (for example, a nearly zero water storage time). To solve that problem, we have introduced a constraint that forces the mean value of the corrections $\epsilon_k$ to be equal to zero. This constraint can be interpreted as an assumption that the runoff estimates provided by the SMB model are correct on average in the study period (even though they still may contain errors in individual years). Furthermore, we watch that $\epsilon_k \geq -1$. A violation of this inequality implies that the true runoff in a given year is negative. Then, in line with equation (14), the estimated BWS becomes negative as well. Of course, all of that is unphysical. In a few cases when this still happens, we refrain from estimating the true runoff. Instead, we force the corresponding estimates of $\epsilon_k$ to be exactly equal to −1. This corresponds to a zero runoff and a zero BWS.

Once all of the unknown parameters are estimated, the vertical displacements caused by variations in BWS can be readily computed with equation (16). For a comparison, we also present the vertical displacements computed under the assumption that the SMB-based runoff estimates are correct, so that $\epsilon_k = 0$. Under this assumption, equation (16) simplifies to:

$$s(t) = \theta \int_{t_0}^{t} r_0(\tau) e^{-\frac{t-\tau}{T_{st}}} d\tau. \tag{18}$$

To distinguish the BWS-related vertical displacements computed with equations (16) and (18), we call them 'calibrated' and 'uncalibrated', respectively.

Notably, the last term on the left-hand side of the functional model given by equation (17) describes the accumulation and discharge of the BWS, which is a short-term process. This signal declines exponentially after the end of the melt season (that is, after the runoff-related signal $r_0(t)$ turns to zero). This is fully consistent with the behaviour of BWS,

which is primarily produced as a result of ice/firn/snow melting and ends up as discharge into the ocean. This term controls, in the first instance, the estimated water storage time $T_{st}$ and empirical coefficient $\theta$. By contrast, the third term on the left-hand side of equation (17) describes the long-term effect of inaccuracies in the runoff estimated as part of the SMB. The effect of these inaccuracies does not vanish in the course of time. The estimated corrections $\epsilon_k$ are mostly controlled by this term, whereas the impact of the fourth term on those estimates is minor. To demonstrate that, we have considered a modified functional model that lacks the BWS-related signal (that is, the aforementioned fourth term):

$$c + at - \sum_{k=1}^{K(t)} \epsilon_k \int_{t_{k_0}}^{t_{k_e}} r_0(\tau) d\tau = m(t) - \int_{t_0}^{t} b_0(\tau) d\tau. \tag{19}$$

Of course, such a functional model is not applicable in the course of the melt season and immediately thereafter. Therefore, we have limited the input data time series to either 6 months per year (November–April) or even to 4 months per year (December–March). This allowed us to obtain two alternative estimates of corrections $\epsilon_k$ (as well as those based on the original functional model given by equation (17)). The mean of the three estimates, as well as the associated standard deviation, is reported in terms of scaling factor $(1 + \epsilon_k)$ per station per year in Extended Data Table 4. We can see that the standard deviation in most cases is less than 0.15. Only one GNET station—GMMA—may show standard deviations larger than 0.3. This means that the obtained estimates of corrections $\epsilon_k$ are sufficiently robust; intrinsic uncertainties associated with the spatial distribution of BWS during the melt season have only a minor effect.

### Accuracy of the obtained estimates
Uncertainties for all reported estimates have been quantified. This concerns both water storage times (Fig. 6 and Extended Data Table 3) and the scaling factors to be applied to the SMB-model-based runoff estimates (Fig. 5 and Extended Data Fig. 2d). The input for the error propagation procedure was defined as the standard deviation of errors in the residual displacements. To that end, the post-fit residuals obtained after fitting the residual displacements with the analytic function given by equation (17) were considered as realizations of the aforementioned errors, which were assumed to be uncorrelated. In view of a nonlinearity of the inversion procedure and a skewness of the resulting error probability density functions, the error propagation was implemented by means of Monte Carlo simulations[74], in which 100 realizations of random errors were generated for each station. The resulting uncertainty intervals were quantified with the 1st quartile and the 3rd quartile as the lower and upper bounds, respectively.

### Computation of elastic vertical deformations and spatial sensitivity of GNSS loading data
Because the solid earth is an elastic body, it experiences vertical deformations in response to a changing surface mass load. Let that load be defined in terms of equivalent water height (EWH) as $h(\varphi, \lambda)$, in which $\varphi$ and $\lambda$ is geographical colatitude and longitude, respectively. We compute the resulting vertical deformations[42] $U(\varphi, \lambda)$ as the convolution of the surface mass load sources $m(\varphi'', \lambda'')$ with the Green's function $G(\psi)$:

$$U(\varphi, \lambda) = \iint m(\varphi'', \lambda'') G(\psi) d\sigma \tag{20}$$

with

$$G(\psi) = \frac{a_e}{m_e} \sum_{l=0}^{\infty} h_n' P_n(\cos\psi), \tag{21}$$

in which $\psi$ is the spherical angular distance between the points $(\varphi, \lambda)$ and $(\varphi'', \lambda'')$; $\sigma$ is the integration area; $a_e$ and $m_e$ are the mean radius and mass of the Earth, respectively; $h_n'$ is the $n$th degree load Love number; and $P_n(\cos\psi)$ are fully normalized Legendre polynomials. As follows

from equation (20), vertical deformations are proportional to the magnitude of surface mass load.

To demonstrate a possible spatial variability of elastic vertical deformations, we have considered several surface mass loads, each of which is homogeneously distributed over a disc of a given radius. The thickness of each disc is defined in terms of EWH such that the deformation at its centre reaches 5 mm, which is similar to the magnitude observed in real data (Fig. 2). We can see (Extended Data Fig. 8a) that the spatial variability of the resulting deformations strongly depends on the spatial extent of the surface load. We believe that a disc of 200 km (or larger) radius, with a total mass of (at least) 35 Gt, gives the best approximation of the actual surface load distribution. This is because the total BWS mass per drainage system estimated from GRACE satellite gravimetry data is on the order of 20–40 Gt (ref. 34), whereas the shape of the actual surface load distribution probably resembles half a disc rather than a disc (most of the GNSS stations are located near the coast, whereas the surface load over the ocean is nearly constant). This simple example also shows that the position of any realistic surface load relative to a given observation point must move laterally by at least a few tens of kilometres to change the elastic vertical deformation at the observation point substantially. This means, for instance, that a local redistribution of meltwater within the firn layer cannot affect deformations observed at a given GNSS station.

Also, we performed a sensitivity study to clarify how mass changes over the entire GrIS, including outlet glaciers hundreds of kilometres away, may affect the observed mass loading signal in practice[75]. By taking the detrended SMB in July 2012 (Extended Data Fig. 8b) as the GrIS mass change signal, we analyse the sensitivity of GNSS loading observed at the KAGA station as an example. The processing strategy is to take the KAGA station as the centre, create ring-shaped zones outward with a step width of 50 km, calculate the vertical displacement caused by SMB inside each ring and then normalize with the SMB signal from entire Greenland. The obtained sensitivity curve (Extended Data Fig. 8c) reveals a substantial contribution from the 'near field': SMB changes within 200 km from KAGA contribute about 80% to the detrended SMB loading displacements. There is little sensitivity to SMB beyond 500 km from KAGA. This is evidence of a relatively high spatial resolution compared with GRACE data: the spatial resolution of the latter is on the order of 400 km (in terms of wavelengths, when the spherical harmonic expansion to degree 96 is considered) or even worse.

## Data availability

Water-related vertical displacements data for all GNET stations analysed in this study can be found at https://doi.org/10.5281/zenodo.8313978 (ref. 85). The GNSS loading data used were provided by the International Mass Loading Service (http://massloading.net/; accessed 1 January 2022), the Nevada Geodetic Laboratory (http://geodesy.unr.edu/; accessed 1 January 2022) and Technical University of Denmark (https://ftp.space.dtu.dk/pub/abbas/GNET/; accessed 1 January 2022). The ice-flow velocities and ice-thickness base maps are provided by NASA's Making Earth System Data Records for Use in Research Environments (MEaSUREs) programme (https://doi.org/10.5067/IMR9D3PEI28U; accessed 1 January 2022). The SGL area changes data are provided at https://doi.org/10.5281/zenodo.10398558 (ref. 86). Source data are provided with this paper.

## Code availability

The MATLAB scripts used to process seasonal elevation changes to plot the main figures are available at https://doi.org/10.5281/zenodo.13836132 (ref. 87). The code of the GRACE MASCON approach to produce mass change of GrIS is also released at https://doi.org/10.5281/zenodo.13836135 (ref. 88).

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

**Acknowledgements** We acknowledge the editor and reviewers for valuable comments. We thank the National Natural Science Foundation of China (42388102, 42322403, 42174096, 42361144001, and 41974094) for the support. M.R.v.d.B. acknowledges support from the Netherlands Earth System Science Centre (NESSC). L.L. was supported by CUHK Direct Grant for Research (4053481 and 4053592). S.A.K. acknowledges support from the Carlsberg Foundation—Semper Ardens Advance programme (CF22-0628). We thank the Danish Agency for Data Supply and Infrastructure (SDFI) for providing GNET GNSS data. B.N. was supported by the Fonds de la Recherche Scientifique de Belgique (F.R.S.-FNRS). The computation of residual GNSS data was supported by the Center for Computational Science and Engineering at the Southern University of Science and Technology. We thank M. Bierkens and Y. Luan for the discussion of groundwater storage and pure elastic deformation, respectively. Computational resources for running the MAR model have been provided by the Consortium des Équipements de Calcul Intensif (CÉCI), financed by F.R.S.-FNRS under grant no. 2.5020.11 and by the Walloon Region and by the Tier-1 supercomputer (Lucia) of the Walloon Region, infrastructure financed by the Walloon Region under grant agreement no. 1910247. J.Liu. acknowledges support from the Henan Provincial Key Laboratory of Hydrosphere and Watershed Water Security and Strategic Priority Research Program of the Chinese Academy of Sciences (XDA20060402).

**Author contributions** J.R. designed, initiated and coordinated the study. P.D. developed the analytic model describing the water storage signal in residual displacement data. J.R., P.D., L.L., M.R.v.d.B. and J.Li interpreted the results and wrote the paper. J.Li, R.K., S.A.K., T.M., M.Z., C.K.S., X.F., J.C., J.Liu. and L.J. contributed to discussions of summer water storage mechanism and edited the paper. B.N. and M.R.v.d.B. provided the RACMO2.3p2 data, X.F. provided the MARv3.9 data and S.A.K. and M.B. contributed to collecting and processing GNET GNSS data. T.v.D. provided the ATM, LWS and NTOL loading time-series data.

**Competing interests** The authors declare no competing interests.

**Additional information**
**Correspondence and requests for materials** should be addressed to Jiangjun Ran.

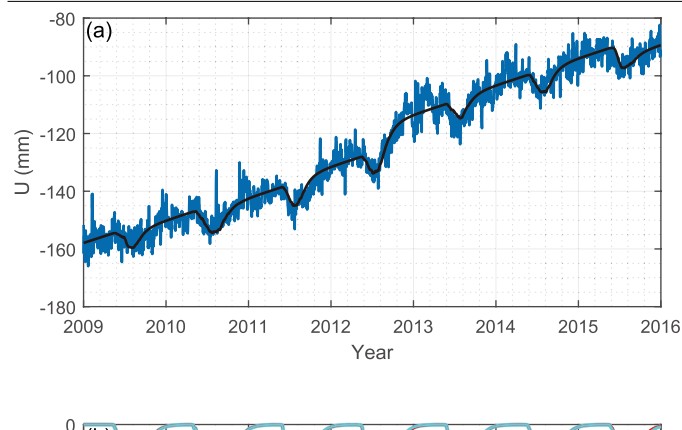

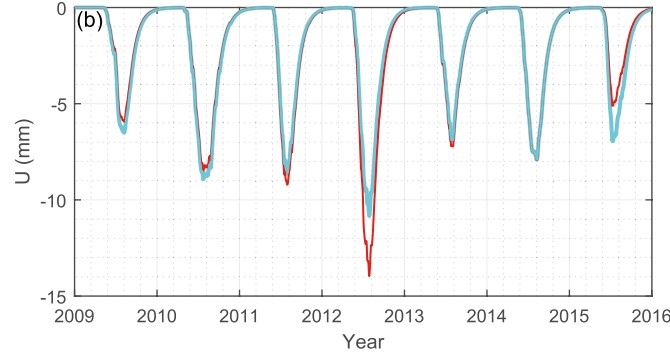

**Extended Data Fig. 1 | Time series and analysis of residual vertical displacements at the KAGA station. a**, The observed displacements (blue) versus the displacements estimated using the proposed analytic function (black). **b**, Calibrated (red) and uncalibrated (light blue) estimates of the displacements caused by BWS variations.

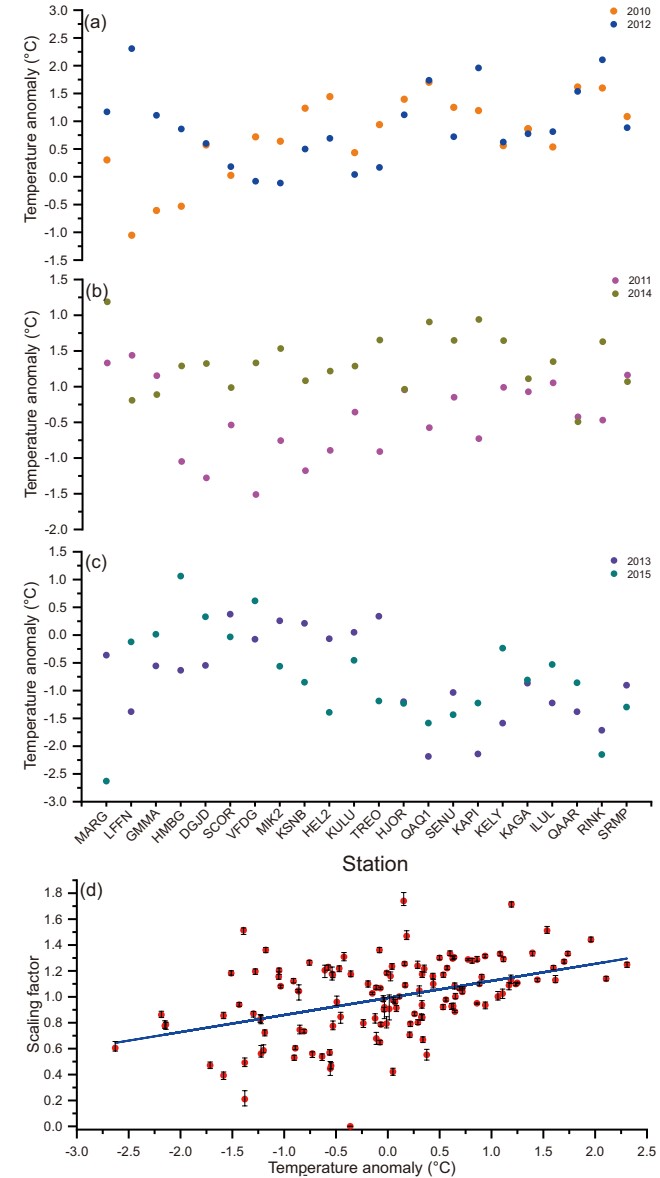

**Extended Data Fig. 2 | Summer temperature anomalies and their correlation with estimated scaling factors. a–c**, 2010–2015 summer temperature anomalies for different GNET stations. **d**, Scaling factor versus temperature anomaly (all stations and years in the 2010–2015 interval are considered). The blue line represents the empirical relationship between the two quantities estimated by means of linear regression. The estimated correlation coefficient is 0.42. The observed slope of the blue line is statistically significant (the two-tailed $P$-value is $2.7 \times 10^{-9}$). Error bars are defined on the basis of the 1st and the 3rd quartiles as the lower and upper bounds, respectively, from 100 Monte Carlo runs.

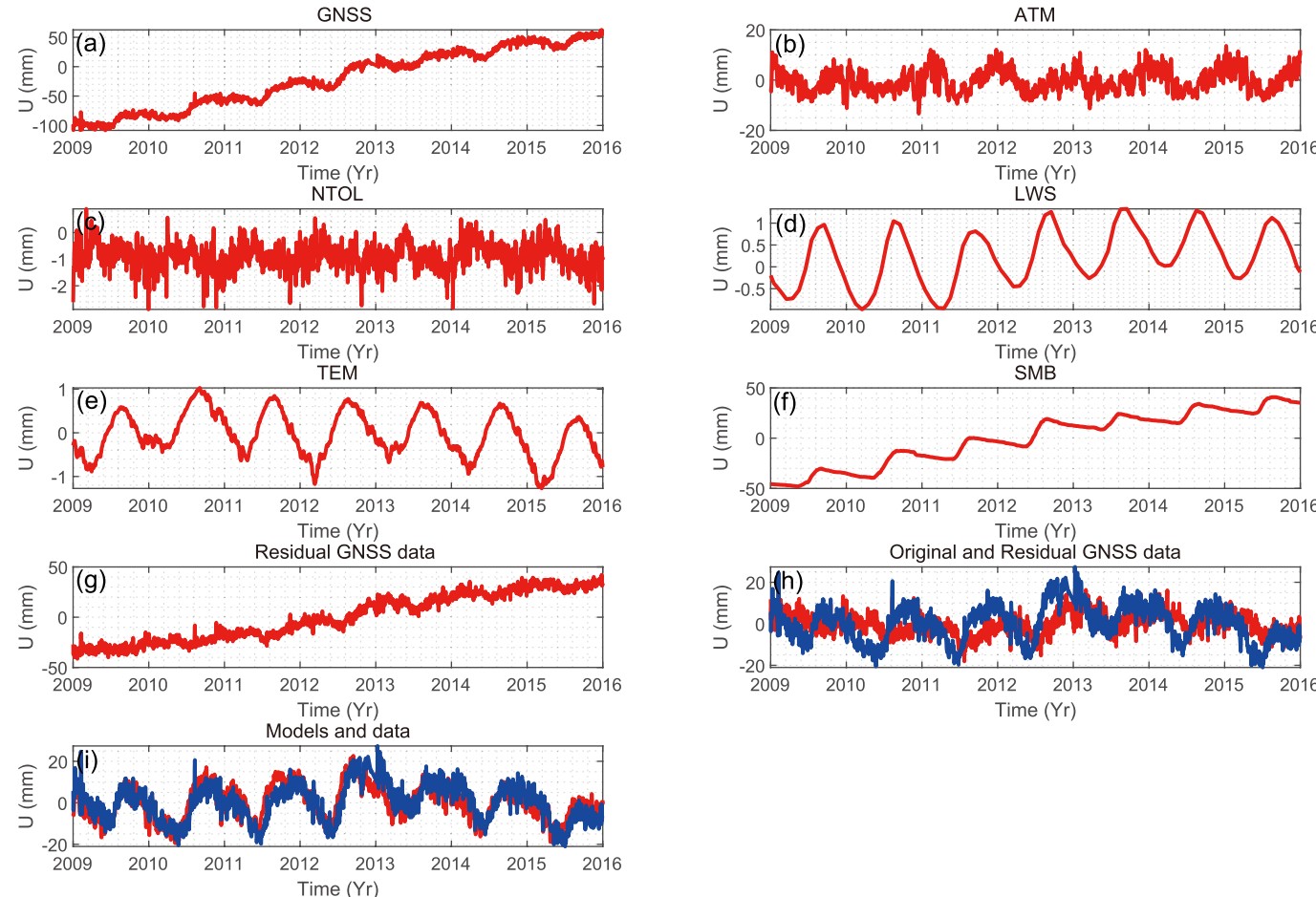

**Extended Data Fig. 3 | Vertical bedrock displacements and the contribution of nuisance signals.** Time series of observed vertical displacements at the KAGA station (panel **a**) and time series calculated for different nuisance signals, as listed in Extended Data Table 1: ATM, NTOL, LWS, TEM and SMB (panels **b**–**f**), as well as the residual displacements after subtracting the calculated nuisance signals (panel **g**), the observed (blue) versus the residual displacements (red) after detrending (panel **h**) and the observed (blue) versus modelled displacements (red) after detrending (panel **i**). Note the different vertical scales.

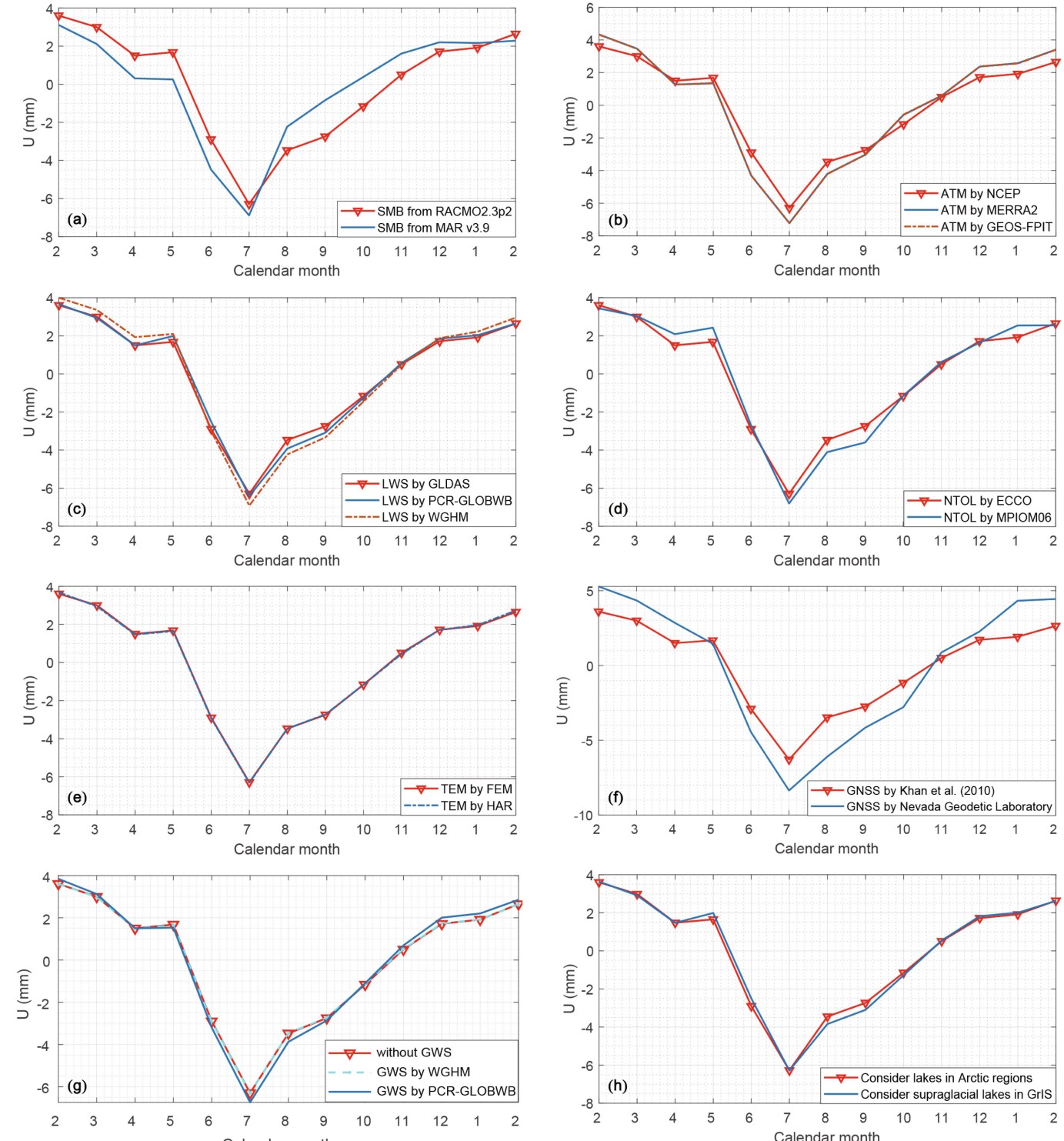

**Extended Data Fig. 4 | Estimation of individual contributors to the error budget for the residual vertical displacement mean annual cycle at the KAGA station.** For that purpose, either the primary GNSS data themselves or one of the primary background models of nuisance signals (red lines) are replaced with one or two alternatives (green and black lines). The panels present different estimates of SMB loading variations (the estimated uncertainty is 0.71 mm) (**a**); ATM loading variations (0.35 mm) (**b**); LWS loading (0.21 mm) (**c**); NTOL loading variations (0.25 mm) (**d**); TEM variations (0.02 mm) (**e**); GNSS solutions (1.04 mm) (**f**); groundwater loading (0.24 mm) (**g**); and water mass loading from lakes in Arctic regions (0.008 mm) and supraglacial lakes in GrIS (0.02 mm) (**h**).

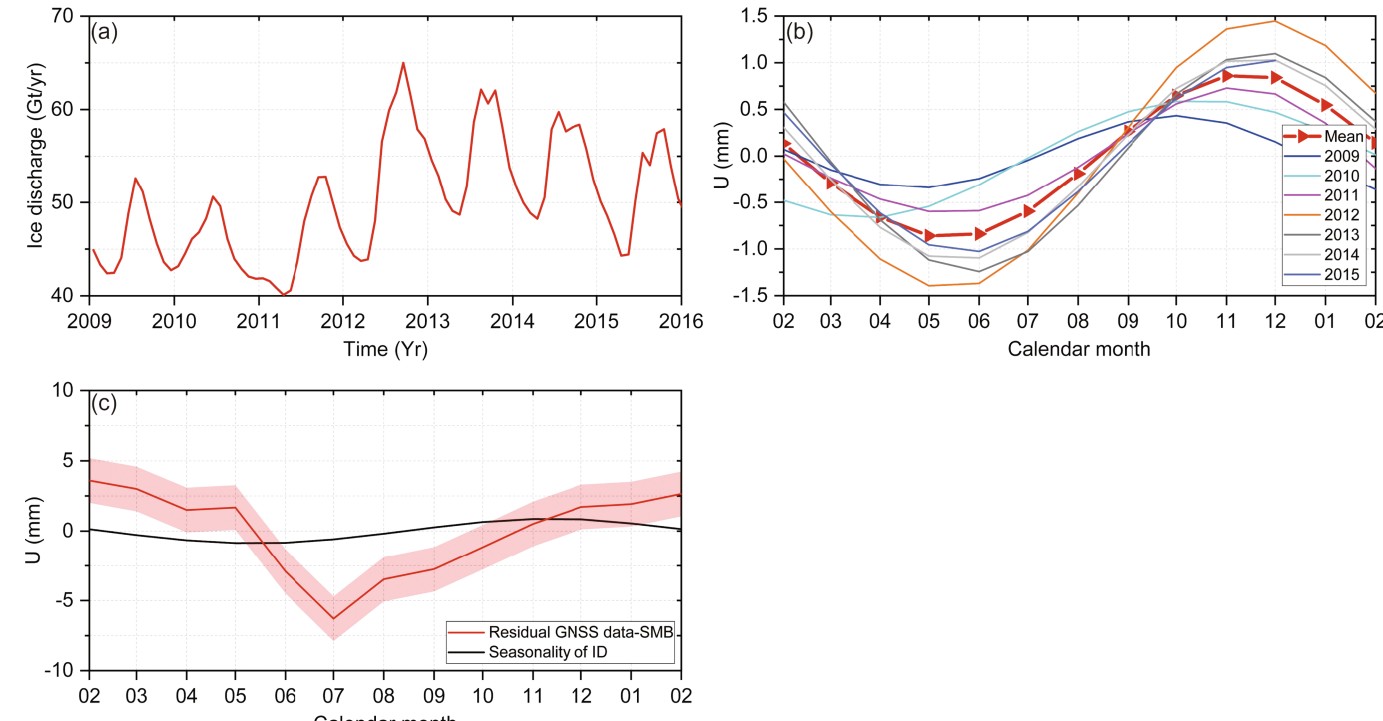

**Extended Data Fig. 5 | Seasonality of ice discharge and associated vertical displacements at the KAGA station. a**, The ice discharge estimated by King et al.[76] for the Jakobshavn Isbræ glacier from 2009 to 2015. **b**, The annual cycles of vertical displacements at the KAGA station caused by ice discharge variations at Jakobshavn Isbræ, showing the mean (thick red curve) and individual years from 2009 to 2015 (thin curves). **c**, Observed monthly mean vertical displacements at the KAGA station (red line), including 68% confidence interval (red shadowing) and similar displacements computed from Jakobshavn Isbræ ice discharge (black line).

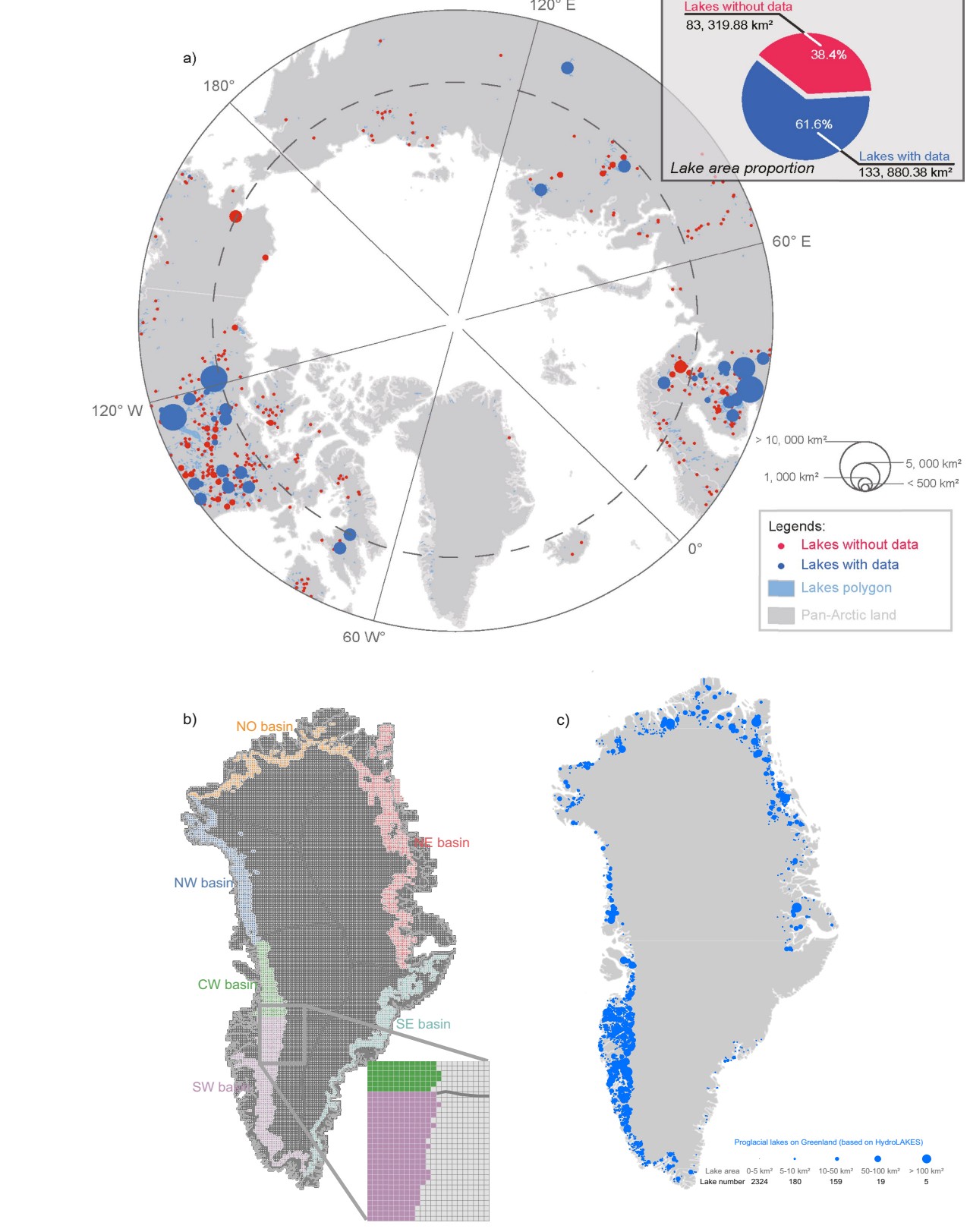

**Extended Data Fig. 6 | Three types of lake considered in this study. a**, Lakes in the pan-Arctic region. **b**, Spatial distribution of cells with SGLs. The cells with SGLs in 2017–2022 are identified using optical images from Sentinel-2 and Landsat 8/9 satellites. The cells without SGLs are shown in grey. **c**, Spatial distribution of proglacial lakes in Greenland.

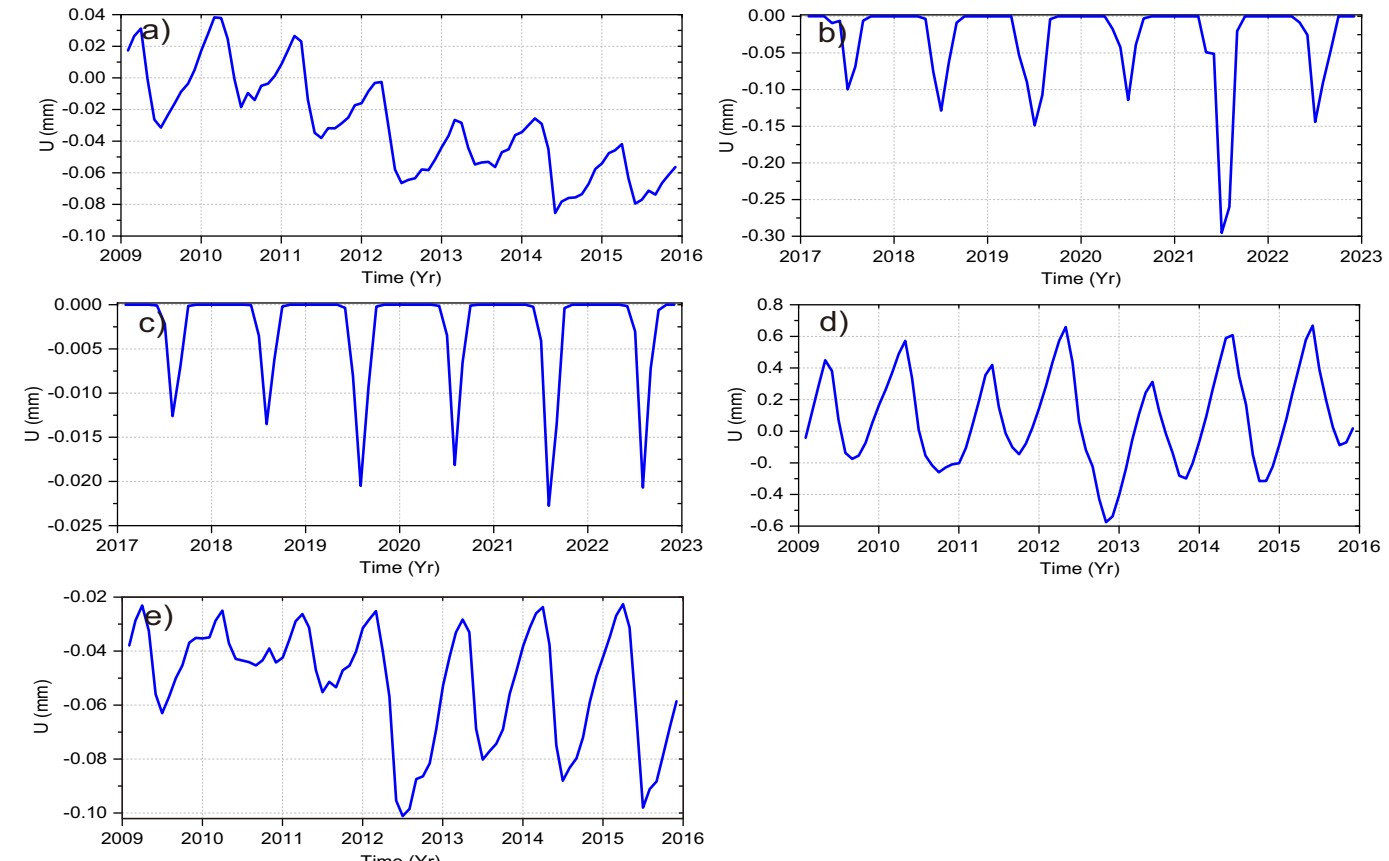

**Extended Data Fig. 7 | Vertical displacements at the KAGA station owing to different components. a**, For water mass changes in lakes in the pan-Arctic region. **b**, For SGL mass changes within the entire GrIS. **c**, For proglacial lake mass changes over entire Greenland. **d**, For groundwater components modelled by PCR-GLOBWB. **e**, For groundwater components modelled by WGHM.

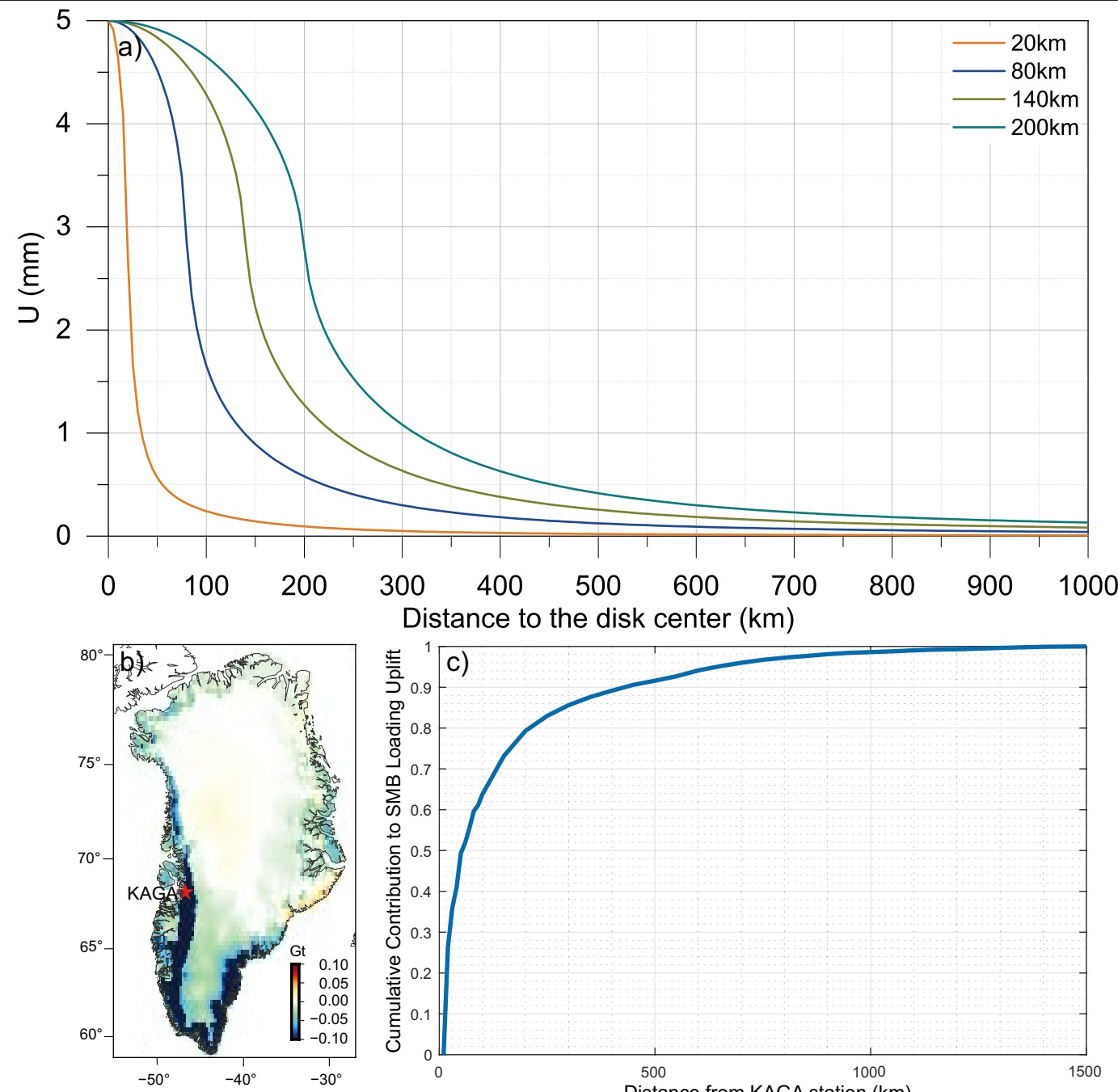

**Extended Data Fig. 8 | Analysis of the spatial sensitivity of GNSS loading signal. a**, Deformations caused by a loading of discs of several radii at different distances from the disc centre. The thickness of each disc is defined in terms of EWH such that the deformation at its centre reaches 5 mm (namely, 147 cm, 53 cm, 35 cm and 28 cm for discs of 20 km, 80 km, 140 km and 200 km radius, respectively). The corresponding disc masses are 1.8 Gt, 11 Gt, 22 Gt and 35 Gt, respectively. These results are consistent with Fig. 1 of ref. 77. **b**, Detrended SMB mass anomalies in July 2012, which are used as input. **c**, Cumulative contribution to the detrended uplift at the KAGA station owing to SMB loading, depending on the radius of the buffer zone around that station. All of the numbers are normalized with the loading uplift caused by the SMB signal from entire Greenland.

**Extended Data Table 1 | Background models applied to correct for nuisance signals in the GNSS data vertical displacements**

| Signal | Models | Temporal sampling | Spatial resolution |
|---|---|---|---|
| Atmospheric loading (ATM) | NCEP[78] | 6 hours | 2.5° × 2.5° |
| | *MERRA2[79]* | 6 hours | 0.5°× 0.625° |
| | *GEOS-FPIT[80]* | 3 hours | 0.5°× 0.625° |
| Non-tidal ocean loading (NTOL) | ECCO[81-82] | 12 hours | 1°× 1° |
| | *MPIOM06[83]* | 6 hours | 2′ × 2′ |
| Land water storage loading (LWS) | GLDAS/Noah[84] | 3 hours | 0.25° × 0.25° |
| | *PCR-GLOBWB[47]* | 1 month | 0.5° × 0.5° |
| | *WGHM[48]* | 1 month | 0.5° × 0.5° |
| Surface mass balance (SMB) | RACMO2.3p2[35] | 1 day | 1×1 km (GrIS); 5.5×5.5 km (tundra) |
| | *MAR v. 3.9[38]* | 1 month | 7.5×7.5 km |
| Thermal elastic expansion (TEM) | FEM from ERA-5[44] | 24 hours | 0.25° × 0.25° |
| | *HAR from ERA-5[44]* | 24 hours | 0.25° × 0.25° |

Alternative background models, which were only used to quantify uncertainties of estimated mean annual cycles of vertical displacements, are presented in italics. References 78–84.

**Extended Data Table 2 | Average annual scaling factors (1+$\epsilon_k$)**

|      | N-NE          | S-SE          | W             | Greenland     |
|------|---------------|---------------|---------------|---------------|
| 2010 | 1.16±0.11     | 1.22±0.06     | 1.21±0.06     | 1.20±0.15     |
| 2011 | 1.22±0.12     | 1.10±0.08     | 1.08±0.07     | 1.13±0.17     |
| 2012 | 1.26±0.11     | 1.18±0.06     | 1.29±0.06     | 1.24±0.14     |
| 2013 | 0.39±0.16     | 0.75±0.09     | 0.68±0.10     | 0.60±0.22     |
| 2014 | 0.89±0.14     | 0.98±0.08     | 1.01±0.09     | 0.96±0.19     |
| 2015 | 0.83±0.20     | 0.82±0.12     | 0.79±0.13     | 0.82±0.27     |

These scaling factors are to be applied to the SMB model runoff estimates to make the resulting vertical displacements best match GNET station observations. The average scaling factors are computed for entire Greenland, as well as for three separate regions: (1) north and northeast (stations from MARG to VFDG); (2) south and southeast (stations from MIK2 to SENU); and (3) west (stations from KAPI to SRMP). The uncertainties are computed as the standard derivations of scaling factors for GNET stations in different regions.

**Extended Data Table 3 | Estimated water storage time for each GNET station**

| station | water storage time (days) | water storage time (weeks) | distance to ice sheet (km) |
|---|---|---|---|
| MARG | $129^{+5}_{-10}$ | 18 | 9.143 |
| LEFN | $46^{+3}_{-3}$ | 7 | 3.661 |
| GMMA | $60^{+7}_{-5}$ | 9 | 57.342 |
| HMBG | $70^{+6}_{-5}$ | 10 | 6.515 |
| DGJG | $68^{+4}_{-5}$ | 10 | 13.250 |
| SCOR | $54^{+3}_{-4}$ | 8 | 70.382 |
| VFDG | $88^{+5}_{-3}$ | 13 | 15.788 |
| MIK2 | $33^{+2}_{-2}$ | 5 | 15.678 |
| KSNB | $52^{+2}_{-3}$ | 7 | 29.487 |
| HEL2 | $19^{+1}_{-1}$ | 3 | 2.398 |
| KULU | $40^{+3}_{-3}$ | 6 | 67.859 |
| TREO | $29^{+1}_{-2}$ | 4 | 1.236 |
| HJOR | $14^{+1}_{-1}$ | 2 | 20.712 |
| QAQ1 | $40^{+2}_{-2}$ | 6 | 46.075 |
| SENU | $24^{+1}_{-1}$ | 3 | 0.549 |
| KAPI | $93^{+4}_{-4}$ | 13 | 36.422 |
| KELY | $51^{+4}_{-3}$ | 7 | 40.910 |
| KAGA | $32^{+2}_{-1}$ | 5 | 5.649 |
| ILUL | $79^{+5}_{-3}$ | 11 | 33.137 |
| QAAR | $62^{+3}_{-2}$ | 9 | 69.408 |
| RINK | $60^{+5}_{-4}$ | 9 | 12.650 |
| SRMP | $73^{+3}_{-2}$ | 10 | 1.421 |

Water storage time uncertainties are computed as the 1st quartile (lower bound) and the 3rd quartile (upper bound) from 100 Monte Carlo runs.

**Extended Data Table 4 | Scaling factors (1+$\epsilon_k$) based on three estimation strategies per year per GNET station**

| Station | 2010 | 2011 | 2012 | 2013 | 2014 | 2015 |
|---|---|---|---|---|---|---|
| GMMA | 1.07±0.12 | 2.16±0.43 | 0.93±0.09 | 0.54±0.14 | 0.47±0.18 | 0.82±0.39 |
| HMBG | 1.12±0.04 | 1.27±0.08 | 1.26±0.03 | 0.58±0.05 | 0.72±0.08 | 1.02±0.01 |
| DGJG | 1.21±0.03 | 1.28±0.08 | 1.30±0.04 | 0.52±0.06 | 0.75±0.08 | 0.95±0.02 |
| VFDG | 1.02±0.02 | 1.22±0.04 | 1.34±0.02 | 0.66±0.02 | 0.82±0.05 | 0.93±0.05 |
| MIK2 | 1.33±0.03 | 1.33±0.06 | 1.02±0.04 | 0.80±0.08 | 0.91±0.01 | 0.61±0.04 |
| KAPI | 1.49±0.20 | 0.79±0.20 | 1.37±0.06 | 0.81±0.08 | 0.93±0.07 | 0.61±0.05 |
| KAGA | 0.953±0.004 | 1.09±0.02 | 1.30±0.02 | 1.02±0.02 | 0.98±0.02 | 0.739±0.009 |
| MARG | 1.09±0.06 | 1.26±0.04 | 1.17±0.04 | 0.64±0.13 | 1.08±0.08 | 0.70±0.04 |
| LEFN | 1.08±0.07 | 1.15±0.05 | 1.21±0.04 | 0.53±0.07 | 1.07±0.03 | 0.76±0.10 |
| SCOR | 1.05±0.09 | 1.22±0.04 | 1.49±0.03 | 0.64±0.08 | 0.64±0.14 | 1.01±0.08 |
| KSNB | 1.089±0.009 | 1.44±0.07 | 1.26±0.03 | 0.66±0.04 | 0.86±0.08 | 0.85±0.13 |
| HEL2 | 1.15±0.03 | 0.61±0.02 | 0.93±0.12 | 0.90±0.10 | 0.66±0.12 | 1.53±0.03 |
| KULU | 1.12±0.03 | 1.15±0.02 | 1.28±0.04 | 0.26±0.17 | 0.79±0.03 | 0.96±0.10 |
| TREO | 1.29±0.03 | 1.17±0.04 | 1.04±0.07 | 0.70±0.05 | 0.94±0.06 | 0.78±0.05 |
| HJOR | 1.33±0.02 | 0.980±0.001 | 1.29±0.02 | 0.53±0.09 | 0.85±0.06 | 0.84±0.06 |
| QAQ1 | 1.27±0.02 | 1.19±0.04 | 1.328±0.005 | 0.86±0.05 | 1.13±0.02 | 0.45±0.07 |
| SENU | 1.12±0.02 | 1.01±0.02 | 1.08±0.01 | 0.97±0.10 | 0.95±0.06 | 0.96±0.02 |
| KELY | 0.91±0.08 | 1.194±0.009 | 1.29±0.02 | 0.83±0.04 | 1.04±0.04 | 0.86±0.06 |
| ILUL | 1.02±0.13 | 1.12±0.14 | 1.25±0.03 | 0.91±0.07 | 1.19±0.06 | 0.82±0.06 |
| QAAR | 0.98±0.13 | 1.42±0.12 | 1.37±0.13 | 0.45±0.21 | 0.95±0.07 | 1.02±0.13 |
| RINK | 1.18±0.05 | 1.27±0.04 | 1.17±0.05 | 0.43±0.06 | 0.86±0.07 | 0.82±0.06 |
| SRMP | 1.26±0.06 | 1.28±0.03 | 1.12±0.02 | 0.55±0.02 | 0.89±0.06 | 0.94±0.06 |

Each of the scaling factors under consideration is estimated in three different ways: (1) using equation (17); (2) using equation (19), while limiting the input data time series to 6 months per year (November–April); and (3) also using equation (19) but after limiting the input data time series to only 4 months per year (December–March). The mean of the three estimates and the associated standard deviation is reported. The values with standard deviations larger than 0.15 and 0.30 are highlighted in yellow and red, respectively.