## [Peer Review File · Nature]

Manuscript Title: Vertical bedrock shifts reveal summer water storage in Greenland ice sheet

Reviewer Comments & Author Rebuttals

Reviewer Reports on the Initial Version:

Referee #1 (Remarks to the Author):

This paper details a novel approach that allows the authors to quantify the meltwater storage within the Greenland Ice Sheet, and the characteristic timescale for drainage. Starting with observed bedrock vertical motions, they remove models for a variety of other effects, which isolates the seasonal water storage signal. The paper is clear and well-written, and includes a comprehensive assessment of uncertainties. The results are robust and compelling, and I think they will be of very broad interest. For me, the paper is very exciting because it illustrates for the first time the capability to measure something new and relevant about the mass transport within the Greenland Ice Sheet, and a similar approach could be applied to data from Antarctica or other deglaciating regions. So the paper is highly original and of high significance.

The only substantive issue I think needs to be improved relates to Extended Data Figure 4. Because of the differing scales and the overall linear trend, it is a bit hard to visualize the relationship between the “nuisance models” and the remaining signal. I think this can be addressed easily enough by an additional figure or additional panels to the existing figure. In particular, I recommend adding the detrended original and residual time series, which will have a scale that may allow visual correlation with the nuisance models, and a panel that compares the data with the sum of the models (this likely also needs to be detrended for reasons of vertical scale).

Otherwise, I have only a few very minor wording suggestions.

Line 63. Change “along the transportation route” to “along its transportation route”

Line 85. Change “does not allow one to answer one” to “does not provide an answer to one”

Line 188. Change “data is used” to “data are used”

Line 195. Insert space before the left parenthesis “factors(” to “factors (”

Figure 2. If possible, add to the caption something like “Extended Data Fig. 4 shows an example of how the residual vertical displacement is computed.”

Line 355. Change “in-house” to “in-house at DTU-Space” or otherwise indicate where “in-house” means.

Referee #2 (Remarks to the Author):

The subject of this paper, seasonal storage and release of liquid water in the Greenland ice sheet, is a key component of many unsolved science problems related to the ice sheet's mass balance and ice flow dynamics. Because meltwater cannot exit the ice sheet instantaneously, the time/space evolution of hydrologic systems revolves around water storage. Storage is, therefore, a key state variable of all supraglacial, englacial, and subglacial hydrologic processes and models thereof. Water storage at the bed is a critical aspect of sliding mechanics and geochemical processes. Yet, the processes are too complex, and our understanding is too limited to explicitly include the many hydrologic processes (e.g., the seasonal evolution of a subglacial drainage system) in any large-scale SMB/runoff model.

Different observational methods, such as radar, GPS signals, GRACE, remote sensing imagery, radar and radiometry, have been used to determine water storage in various compartments. Each method presents resolution and interpretation challenges, and several offer only snapshots rather than continuous time series. Breakthroughs with direct measurements of storage thus have the potential to make a strong impact on many different aspects of Greenland hydrology.

Concerns:

1. Hydrologic interpretation.

Throughout the paper, the description of the water storage mechanism is quite puzzling.

Examples include:

-- Figure 1 depicts englacial lakes (and subglacial lakes that are actually shown as englacial)

— I know of no evidence for lakes existing inside the ice.

-- The text concerning loading is confusing because a phase change is not a mass change, so water mass loading must be due to the redistribution of water from one place to another (not stated).

-- The manuscript intro/discussion is structured around water storage in the firn compartment, but why this is driving the loading signal versus other compartments is not clear. Melt across most of the firn area will result in a short-distance mass redistribution, vertically within the upper 10 m. Only a relatively narrow reach of the far lower accumulation area has substantial horizontal flow and accumulation of water, but is there evidence that this is enough to drive the loading signal? Regardless, all runoff is accumulated toward the margin. Whereas a SMB model may simulate a large amount of meltwater generation across the accumulation zone, the subglacial drainage system, not the firn compartment, has the most accumulating water mass and is therefore the most likely candidate for loading. Water at the bed cannot be dismissed by invoking fast conduit drainage. A large body of literature on subglacial hydrology debates issues such as a) how far inward on the ice sheet fast-draining conduits can actually develop (e.g., Werder et al., 2013); and, b) how much area of the bed sits between conduits and does not drain quickly (e.g., Hoffman et al., 2016).

2. Signal robustness.

Because the water storage signal is a small residual that emerges after many large corrections to the original data, the robustness of the signal is dependent on the confidence of the removal of so-called nuisance signals. Perhaps some of the following issues are partly or solely responsible for the spatial variability of the results.

-- Prior work by Lui et al. (2017) demonstrated the difficulty of SMB corrections in Greenland due to far-field SMB influences. In other words, where does the SMB signal come from? Lui et al. (2017) argue that SMB from glaciers 1200 km away can impact Greenland GNSS signals. In fact, they found that correcting for SMB adversely inflated the residual. The manuscript does not make clear how this problem was handled nor is the Lui et al. (2017) paper cited.

-- The analysis does not appear to consider the seasonal loading signal caused by ice flow across the ablation zone (c.f., unloading by calving). The speed of the ice sheet in the ablation zone increases during summer, typically by a factor of 2-3 (see Zwally et al. (2002) for the seminal paper, with much confirming work published since then). Speed typically peaks mid-summer, similar to the GNSS loading signal. Ice acceleration causes horizontal advection of ice thickness gradients — more ice is moved low that must be slowly beaten back by ablation. How much of the residual could be due to this effect?

-- The product used for LWS corrections is a global product and is not necessarily well-suited for Arctic regions dominated by permafrost hydrology. Some areas have many lakes fluctuating lake levels. Further, the product has no treatment of groundwater, whereas Liljedahl et al., (2021) show very large head swings in the groundwater system along the ice margin. LWS is a big correction and uncertainties will vary substantially around the ice sheet.

Originality:

Storage--

Seasonal storage of meltwater is well-established. For example, prior work has measured time series of 'bed separation,' the local water storage in the subglacial drainage system (e.g., Andrews et al. 2014). These studies have the advantage of pinpointing the storage location but exist for only a few points and time periods. Other work has used radar to identify the spatial variability of water storage at the bed over 10s to 100s of km length scales (Chu et al., 2016). Ran et al. (2018) used GRACE data to show a time delay, peaking in midsummer, between RACMO's meltwater generation and total mass loss (i.e., water storage). The finding is required since RACMO, MAR, and other regional climate models do not simulate water transport through hydrologic pathways and so they instantly remove water from the ice sheet. Thus, the disconnect peaks midsummer when meltwater generation is at maximum.

>This paper presents time series of storage/release at 22 regions around the ice sheet. However, since the compartment/precise area driving the signal is not clear, it is also not clear how the results can be used to advance understanding of hydrologic systems.

Scaling factor--

Large melt biases and uncertainties in RACMO/MAR due to factors such as clouds, albedo, and other is well established (e.g., Fettweis et al., 2020).

>This paper demonstrates that a scaling factor is required to force RCM melt to better match GNSS residuals. The scaling factors are highly variable, centered around 1 (Fig. 4), and has a relatively weak correlation with melt amount. How to incorporate the result into hydrologic

studies is unclear, and so the results would seem to be most relevant to RCM testing and validation studies.

====

Werder MA, Hewitt IJ, Schoof CG and Flowers GE (2013) Modeling channelized and distributed subglacial drainage in two dimensions. *Journal of Geophysical Research: Earth Surface* 118(4), 2140–2158. doi:10.1002/jgrf.20146.

Hoffman, M.J., Andrews, L.C., Price, S.A., Catania, G.A., Neumann, T.A., Lüthi, M.P., Gulley, J.D., Ryser, C., Hawley, R.L., & Morriss, B.F. (2016). Greenland subglacial drainage evolution regulated by weakly connected regions of the bed. *Nature Communications*, 7.

Liu, L., Khan, S.A., Dam, T.V., Ma, J.H., & Bevis, M.G. (2017). Annual variations in GPS-measured vertical displacements near Upernavik Isstrøm (Greenland) and contributions from surface mass loading. *Journal of Geophysical Research: Solid Earth*, 122, 677 - 691.

Zwally, H.J., Abdalati, W., Herring, T.A., Larson, K.M., Saba, J.L., & Steffen, K. (2002). Surface Melt-Induced Acceleration of Greenland Ice-Sheet Flow. *Science*, 297, 218 - 222.

Liljedahl, L.C., Meierbachtol, T.W., Harper, J.T., van As, D., Näslund, J., Selroos, J., Saito, J., Follin, S., Ruskeeniemi, T., Kontula, A., & Humphrey, N.F. (2021). Rapid and sensitive response of Greenland's groundwater system to ice sheet change. *Nature Geoscience*, 14, 751 - 755.

Andrews, L.C., Catania, G.A., Hoffman, M.J., Gulley, J.D., Lüthi, M.P., Ryser, C., Hawley, R.L., & Neumann, T.A. (2014). Direct observations of evolving subglacial drainage beneath the Greenland Ice Sheet. *Nature*, 514, 80-83.

Li, W., Shum, C.K., Li, F., Zhang, S., Ming, F., Chen, W., Zhang, B., Lei, J., & Zhang, Q. (2020). Contributions of Greenland GPS Observed Deformation From Multisource Mass Loading Induced Seasonal and Transient Signals. *Geophysical Research Letters*, 47.

Chu, W., Schroeder, D.M., Seroussi, H., Creyts, T.T., Palmer, S.J., & Bell, R.E. (2016). Extensive winter subglacial water storage beneath the Greenland Ice Sheet. *Geophysical Research Letters*, 43, 12,484 - 12,492.

Ran, J., Vizcaíno, M., Ditmar, P., van den Broeke, M.R., Moon, T.A., Steger, C.R., Enderlin, E.M., Wouters, B., Noël, B.P., Reijmer, C.H., Klees, R., Zhong, M., Liu, L., & Fettweis, X. (2018). Seasonal mass variations show timing and magnitude of meltwater storage in the Greenland Ice Sheet. *The Cryosphere*.

Fettweis, X., et al., (2020). GrSMBMIP: intercomparison of the modelled 1980–2012 surface mass balance over the Greenland Ice Sheet. *The Cryosphere*.

Author Rebuttals to Initial Comments:

Referee #1 (Remarks to the Author):

This paper details a novel approach that allows the authors to quantify the meltwater storage within the Greenland Ice Sheet, and the characteristic timescale for drainage. Starting with observed bedrock vertical motions, they remove models for a variety of other effects, which isolates the seasonal water storage signal. The paper is clear and well-written, and includes a comprehensive assessment of uncertainties. The results are robust and compelling, and I think they will be of very broad interest. For me, the paper is very exciting because it illustrates for the first time the capability to measure something new and relevant about the mass transport within the Greenland Ice Sheet, and a similar approach could be applied to data from Antarctica or other deglaciating regions. So the paper is highly original and of high significance.

Authors: Thank you very much for reviewing this paper and your constructive comments. In the light of the insightful response from you and Reviewer #2, we have made the corresponding changes, which have greatly improved the original manuscript. A detailed point-by-point response to your comments addressing all of the identified issues is listed below. We appreciate your encouraging agreement on the novelty and importance of the results.

The only substantive issue I think needs to be improved relates to Extended Data Figure 4. Because of the differing scales and the overall linear trend, it is a bit hard to visualize the relationship between the “nuisance models” and the remaining signal. I think this can be addressed easily enough by an additional figure or additional panels to the existing figure. In particular, I recommend adding the detrended original and residual time series, which will have a scale that may allow visual correlation with the

nuisance models, and a panel that compares the data with the sum of the models (this likely also needs to be detrended for reasons of vertical scale).

Authors: Thank you for excellent suggestion. In light with the suggestions, two more panel have been added to Extended Data. Fig. 4. Panel (h) is added to better visualize the relationship between the “nuisance models” and the remaining signal. Panel (i) is included to show a comparison of the original data with the sum of the models. For your convenience, the revised Extended Fig. 4 is shown below. Please see lines 846-853 in the revised manuscript.

Fig.A1 the revised ED Fig. 4.

Otherwise, I have only a few very minor wording suggestions.

Line 63. Change “along the transportation route” to “along its transportation route”

Authors: Thank you very much for the comment. Corrected.

Line 85. Change “does not allow one to answer one” to “does not provide an answer to one”

Authors: Corrected. Please see line 77 in the revised manuscript.

Line 188. Change “data is used” to “data are used”

Authors: Corrected. Please see line 169 in the revised manuscript.

Line 195. Insert space before the left parenthesis “factors(” to “factors (”

Authors: Corrected. Please see line 176 in the revised manuscript.

Figure 2. If possible, add to the caption something like “Extended Data Fig. 4 shows an example of how the residual vertical displacement is computed.”

Authors: Corrected. Please see lines 241-242 in the revised manuscript.

Line 355. Change “in-house” to “in-house at DTU-Space” or otherwise indicate where “in-house” means.

Authors: Corrected. Please see lines 398-399 in the revised manuscript.

Thank you again for your excellent suggestions which have resulted in an improved manuscript. We are very grateful.

Referee #2 (Remarks to the Author):

The subject of this paper, seasonal storage and release of liquid water in the Greenland ice sheet, is a key component of many unsolved science problems related to the ice sheet's mass balance and ice flow dynamics. Because meltwater cannot exit the ice sheet instantaneously, the time/space evolution of hydrologic systems revolves around water storage. Storage is, therefore, a key state variable of all supraglacial, englacial, and subglacial hydrologic processes and models thereof. Water storage at the bed is a critical aspect of sliding mechanics and geochemical processes. Yet, the processes are too complex, and our understanding is too limited to explicitly include the many hydrologic processes (e.g., the seasonal evolution of a subglacial drainage system) in any large-scale SMB/runoff model.

Different observational methods, such as radar, GPS signals, GRACE, remote sensing imagery, radar and radiometry, have been used to determine water storage in various compartments. Each method presents resolution and interpretation challenges, and several offer only snapshots rather than continuous time series. Breakthroughs with direct measurements of storage thus have the potential to make a strong impact on many different aspects of Greenland hydrology.

Authors: Thank you very much for your time to review this paper and your excellent comments. We agree with your emphasis on the importance of studies on seasonal storage and release of liquid water in the Greenland ice sheet. We are very grateful for your valuable comments and have addressed them one-by-one as follows.

Concerns:

1. Hydrologic interpretation.

Throughout the paper, the description of the water storage mechanism is quite puzzling. Examples include:

-- Figure 1 depicts englacial lakes (and subglacial lakes that are actually shown as englacial) — I know of no evidence for lakes existing inside the ice.

Authors: Thank you very much for the comment. Sorry for the misunderstanding. Figure 1 is a *schematic* figure to show the processes related to the liquid water evolution beneath the ice sheet surface and the associated changes of vertical bedrock. The distance of englacial lakes from the ice sheet surface is schematic, just indicating that some water is buffered beneath the ice sheet surface, and does not mean the actual location of englacial lakes is inside the ice sheet. However, based on the above comments, we have improved Fig. 1, to avoid any misunderstanding. For your convenience, the revised Fig. 1 is shown below. Please see lines 218-232 in the revised manuscript.

Fig. A2 The revised Fig. 1.

-- The text concerning loading is confusing because a phase change is not a mass change, so water mass loading must be due to the redistribution of water from one place to another (not stated).

Authors: Thanks for this thoughtful comment. The “phase change” refers to melting and refreezing processes. In order to avoid potential confusion, we have added the following sentence in the introductory section (Lines 84-85):

“Note that bedrock displacement time-series describe the mass of buffered water rather than the total water mass, since surface loading does not change if water re-freezes in-place.”

-- The manuscript intro/discussion is structured around water storage in the firn compartment, but why this is driving the loading signal versus other compartments is not clear. Melt across most of the firn area will result in a short-distance mass redistribution, vertically within the upper 10 m. Only a relatively narrow reach of the far lower accumulation area has substantial horizontal flow and accumulation of water, but is there evidence that this is enough to drive the loading signal? Regardless, all runoff is accumulated toward the margin. Whereas a SMB model may simulate a large amount of meltwater generation across the accumulation zone, the subglacial drainage system, not the firn compartment, has the most accumulating water mass and is therefore the most likely candidate for loading. Water at the bed cannot be dismissed by invoking fast conduit drainage. A large body of literature on subglacial hydrology debates issues such as a) how far inward on the ice sheet fast-draining conduits can actually develop (e.g., Werder et al., 2013); and, b) how much area of the bed sits between conduits and does not drain quickly (e.g., Hoffman et al., 2016).

Authors: We agree that the introductory section was too much focused on processes in the firn layer, which resulted in some mismatch between that section and the rest of the manuscript. To solve this issue, we have re-organized the introductory section to put more emphasis on water at the bed (Lines 65-76). The firn layer is mentioned in the Discussion Section only briefly, and primarily in the context of our interpretation of the scaling factors to be applied to runoff estimates provided by the SMB model (Lines 189-199).

As for “A large body of literature on subglacial hydrology debates issues such as a) how far inward on the ice sheet fast-draining conduits can actually develop (e.g., Werder et al., 2013); and, b) how much area of the bed sits between conduits and does not drain quickly (e.g., Hoffman et al., 2016).”, the loading observed using GNSS verticals is the bedrock response to integrated mass changes. We have removed SMB and other non-ice sources, attributing the remaining changes to temporally buffered meltwater. But it is difficult to further differentiate the pathways or storages of the

buffered meltwater by using GNSS data alone. We hope this study will stimulate further research to address ice sheet hydrology from a new perspective like this.

2. Signal robustness.

Because the water storage signal is a small residual that emerges after many large corrections to the original data, the robustness of the signal is dependent on the confidence of the removal of so-called nuisance signals. Perhaps some of the following issues are partly or solely responsible for the spatial variability of the results.

Authors: We agree that this signal is relatively small, but it is clearly visible (see, e.g., panel (h) in Extended Data Fig. 4 in the updated manuscript, please see Lines 851-853) and reliably estimated in this study. To ensure the robustness of our estimates, we conducted a thoughtful analysis by evaluating all corrections with the State-of-the-Art models or data in the community. Please see revised ED Fig. 5 (Lines 854-863) and the following text. It is worth to note that we also compared the GNSS-based water storage signal with that based on independent satellite gravimetry data (Fig. 3, see Lines 245-251). This comparison confirmed that this signal is real and not a data artefact.

-- Prior work by Lui et al. (2017) demonstrated the difficulty of SMB corrections in Greenland due to far-field SMB influences. In other words, where does the SMB signal come from? Lui et al. (2017) argue that SMB from glaciers 1200 km away can impact Greenland GNSS signals. In fact, they found that correcting for SMB adversely inflated the residual. The manuscript does not make clear how this problem was handled nor is the Lui et al. (2017) paper cited.

Authors: Thank you very much for the comment. Liu et al. (2017) was the previous study by one of the co-authors. Liu et al. (2017) studied Upernavik glacier using two GNSS stations located in the nearby bedrock area. We agree with Liu et al. (2017) that far-field SMB signals must be taken into account when interpreting the annual

amplitude of the GNSS verticals. In this study we use an ice-sheet-wide SMB model (RACMO) rather than a local or regional model and therefore correct for the far-field signal. We now state in line 376 that the SMB model covers the entire GrIS (and not just a small area).

In addition, we made a sensitivity study (Lines 657-670) to clarify how mass changes over the entire ice sheet, including glaciers 1,200 km away, may affect the observed mass loading signal. By taking the detrended SMB in July 2012 (Fig. A3 in the rebuttal) as the ice sheet mass change signal, we analyze the sensitivity of GNSS loading observed at KAGA as an example. The processing strategy is to take the KAGA station as the center, create ring-shaped zones outward with a step width of 50 km, calculate the vertical displacement caused by SMB inside each ring, and then normalized with the entire Greenland. The results show that cumulative contribution to the detrended SMB loading reaches approximately 80% when the distance from KAGA station is 200 km. The sensitivity curve (Fig. A3) reveals a significant contribution from near field: SMB changes within 200 km from KAGA contribute by about 80% to the detrend SMB loading displacements. There is little sensitivity to SMB beyond 500 km from KAGA.

Fig. A3 (a) Detrended SMB mass anomalies in July 2012. (b) Cumulative contribution to the detrended uplift at KAGA due to SMB loading, depending on the radius of the buffer zone around that station. All the numbers are normalized with the loading uplift caused by the SMB signal over entire Greenland. The X axis shows a step of 50 km in the distance range of 0-1,500 km. This figure is included as ED Fig. 14 in the revised manuscript.

-- The analysis does not appear to consider the seasonal loading signal caused by ice flow across the ablation zone (c.f., unloading by calving). The speed of the ice sheet in the ablation zone increases during summer, typically by a factor of 2-3 (see Zwally et al. (2002) for the seminal paper, with much confirming work published since then). Speed typically peaks mid-summer, similar to the GNSS loading signal. Ice acceleration causes horizontal advection of ice thickness gradients — more ice is moved low that must be slowly beaten back by ablation. How much of the residual could be due to this effect?

Authors: We agree with the reviewer that the seasonal loading signal might be caused by the ice flows. In this study, we have considered this effect. Please see the effect of seasonal variations in ice speed analyzed in the Section “Contribution of ice discharge”, which is a part of Methods (see lines 416-429). To obtain an upper bound of the contribution of seasonal ice discharge variations to the observed vertical displacements, we use Jakobshavn Isbræ (JI) as an example because it shows the largest ice flow velocity seasonality amongst GrIS outlet glaciers. By investigating the GNSS loading signal observed at the KAGA GPS station, which is located ~5 km from the JI terminus, we conclude that this effect (i.e., $< \sim 1\text{mm}$) is not significant.

-- The product used for LWS corrections is a global product and is not necessarily well-suited for Arctic regions dominated by permafrost hydrology. Some areas have many lakes fluctuating lake levels. Further, the product has no treatment of groundwater, whereas Liljedahl et al., (2021) show very large head swings in the groundwater system along the ice margin. LWS is a big correction and uncertainties will vary substantially around the ice sheet.

Authors: Thanks for this comment. We agree that lakes and groundwater play important roles in Arctic regions dominated by permafrost hydrology. Indeed, the GLDAS model we considered in the previous variant of the manuscript lacks both water storage components. To solve this problem, we also add two hydrological

models (see Extended Data Fig. 5c in Line 854), i.e., WaterGAP2 Global Hydrology Model (WGHM) and PCRaster Global Water Balance (PCR-GLOBWB), both of which take groundwater into account. Furthermore, we explicitly consider water mass variations of lakes in Arctic regions (Lines 431-475).

1) Three State-of-The-Art models are utilized in this study in total: GLDAS, PCR-GLOBWB and WGHM. Even though these models are not specific for Arctic regions, we believe they are accurate enough at least for the first-order magnitude. Note that WGHM is tailored in Arctic region by considering the existence of permafrost. A comparison with other models showed that WGHM agrees best with independent in-situ observations collected in Arctic region (Gädeke et al., 2020). In addition, as we have shown with our sensitivity analysis of GNSS-based loading signal presented above (Fig. A3), this contribution is mostly (~80%) limited to the radius of ~200 km. Most parts of the Arctic regions are quite far (> 1000 km) from the GNSS stations in Greenland, and thereby the mass variations over there are little impact the GNSS based water storage signals observed in this study.

Taking the KAGA as an example: the vertical displacement at KAGA in response to global LWS loading derived from the 3 hydrological models, excluding Greenland, during the study period (2009-2015) ranges from -1.0 to ~1.5 mm (Fig. A4). This magnitude is sufficiently small to allow for neglecting the potential influence of LWS on the Greenland ice sheet meltwater storage.

Fig. A4 Time series of the vertical displacements at KAGA in response to LWS loading excluding Greenland.

References:

- GLDAS: Li, B., M. Rodell, S. Kumar, H. Beaudoin, A. Getirana, B. F. Zaitchik, et al. (2019) Global GRACE data assimilation for groundwater and drought monitoring: Advances and challenges. *Water Resources Research*, 55, 7564-7586. doi:10.1029/2018wr024618
- PCR-GLOWB: Sutanudjaja, E. H., Van Beek, R., Wanders, N., Wada, Y., Bosmans, J. H., Drost, N., ... & Bierkens, M. F. (2018). PCR-GLOBWB 2: a 5 arcmin global hydrological and water resources model. *Geoscientific Model Development*, 11(6), 2429-2453.
- WGHM: Müller Schmied, H., Cáceres, D., Eisner, S., Flörke, M., Herbert, C., Niemann, C., ... & Döll, P. (2021). The global water resources and use model WaterGAP v2. 2d: Model description and evaluation. *Geoscientific Model Development*, 14(2), 1037-1079.
- Gädeke, A., Krysanova, V., Aryal, A. et al. (2020). Performance evaluation of global hydrological models in six large Pan-Arctic watersheds. *Climatic Change* 163, 1329–1351, <https://doi.org/10.1007/s10584-020-02892-2>

2) As for “Some areas have many lakes fluctuating lake levels.”, indeed, there are many lakes in the Arctic region. To further quantify the potential water mass impact of lakes onto GNSS loading signal in Greenland, we consider three types of lakes (Lines 431-475): lakes in Pan-Arctic region (north of 60 °N) in general; supraglacial lakes (SGL) on GrIS; and proglacial lakes in the coastal part of Greenland.

Fig. A5 Distribution of lakes in Pan-Arctic region. This figure is included as ED Fig. 7 in the revised manuscript.

Lakes in Pan-Arctic region: Specifically, the monthly lake water storage data of some large lakes were directly downloaded from a recently published dataset by Yao et al., (2023). In addition, we accessed the intra-annual lake level datasets from three portals, i.e., Hydroweb (<https://hydroweb.theia-land.fr/>), Global Reservoirs and Lakes Monitor (G-REALM, https://ipad.fas.usda.gov/cropexplorer/global_reservoir/), and Database for Hydrological Time Series of Inland Waters (DAHITI, <https://dahiti.dgfi.tum.de/en/>), respectively (Cretaux et al., 2011; Birkett et al., 2009;

Schwatke et al., 2015). For months without lake area data in the dataset provided by Yao et al., (2023), we follow the empirical relationship between lake water level and lake surface area from a similar previous study(Song et al., 2013), by interpolating the missing lake area data with ampler level data and converting the lake area and level data to water storage changes.

It is worth mentioning that the recorded lakes (38 in total, indicated by blue circles in Fig. A5) are typically large, with 33 of them having a surface area greater than 500 km², accounting for 61.6% of the total surface area of Arctic lakes extracted by Yao et al. (2023). Note that the reference surface area here is derived from the HydroLAKES polygon (Messenger et al., 2016). For the remaining smaller inland lakes (red circles in Fig. A5), the area data are very limited. Therefore, we employ a scale-up strategy to estimate their impact on mass loading (Song et al., 2013).

Taking KAGA station as an example, the vertical displacement induced by the load of Arctic lake water storage during the study period (2009-2015) ranges from -0.1 to 0.04 mm (Fig. A6). This magnitude is sufficiently small to allow for neglecting the potential influence of these lakes in the Arctic region.

Fig. A6 Vertical displacements of lake loading at KAGA station. This figure is included as ED Fig. 8 in the revised manuscript.

SGL on the GrIS: In addition, we assessed the vertical displacement caused by water mass loading from SGL on the GrIS. Firstly, we divided the GrIS into 10-by-10 km equal-area cells and identified all the cells with SGL using ~ 300,000 high spatiotemporal resolution optical images from Sentinel-2 and Landsat 8/9 over 2017-2022, as shown in Fig. A7. Then, the monthly SGL area changes during the melting season (i.e., May-September) were derived. The SGL area changes data are distributed via <https://doi.org/10.5281/zenodo.10398558>. Note that it is problematic to obtain the monthly SGL area change for entire GrIS before the launch of Sentinel-2B satellites in 2017 due to a high cloud contamination. We assumed, however, that the SGL area changes over 2009-2015 interval was similar to those over 2017-2022. When converting the SGL area changes to mass changes, we assumed that the maximum depth of GrIS supraglacial lakes is around 8.5 metres (Fair et al., 2020; Xiao et al., 2023), which allowed us to estimate the upper limit of water mass changes Fig. A8. The result shows that even for the maximal depth of 8.5 m, the magnitude of loading signal caused by SGL mass changes is ~0.3 mm, which is negligible, compared with the BWS signal.

Proglacial lakes in Greenland: Similar to the SGL, we also consider the impact of proglacial lakes in Greenland using Hydrolake database (Fig. A9). In total, 2 687 proglacial lakes are taken into account. Most of the proglacial lakes are smaller than 5 km², while the largest one could reach ~100 km². The loading signal caused by proglacial lakes is small. At KAGA station, for instance, it is of the order of only 0.02 mm (Fig. A10).

Fig. A7 Spatial distribution of cells with SGLs. The grey cells indicate that there were no SGL over 2017-2022 from optical images. This figure is included as ED Fig. 9 in the revised manuscript.

Fig.

A8 The vertical displacements at KAGA station caused by SGL mass changes. This figure is included as ED Fig. 10 in the revised manuscript.

Fig. A9 Spatial distribution of proglacial lakes in Greenland. This figure is included as ED Fig. 11 in the revised manuscript.

Fig. A10 | The vertical displacements at KAGA station caused by proglacial lake mass changes over entire Greenland. This figure is included as ED Fig. 12 in the revised manuscript.

References:

- Cretaux J-F., Arsen A., Calmant S., et al., 2011. SOLS: A lake database to monitor in the Near Real Time water level and storage variations from remote sensing data, *Advances in space Research*, 47, 1497-1507.
- Birkett, C.M., Reynolds, C., Beckley, B., et al., 2009. From Research to Operations: The USDA Global Reservoir and Lake Monitor, chapter 2 in 'Coastal Altimetry', Springer Publications, eds. S. Vignudelli, A.G. Kostianoy, P. Cipollini and J. Benveniste, ISBN 978-3-642-12795-3, 2010.
- Schwatke C, Dettmering D, Bosch W, et al., 2015. DAHITI—an innovative approach for estimating water level time series over inland waters using multi-mission satellite altimetry, *Hydrology and Earth System Sciences*, 19(10), 4345-4364.
- Yao F, Livneh B, Rajagopalan B, et al., 2023. Satellites reveal widespread decline in global lake water storage, *Science*, 380(6646): 743-749.
- Song, C., Huang, B., Ke, L., 2013. Modeling and analysis of lake water storage changes on the Tibetan Plateau using multi-mission satellite data. *Remote Sensing of Environment*. 135. 25-35.
- Messenger, M.L., Lehner, B., Grill, G., et al., 2016. Estimating the volume and age of water stored in global lakes using a geo-statistical approach. *Nature Communications*, 7: 13603.
- Fair, Z., Flanner, M., Brunt, K. M., et al., 2020. Using ICESat-2 and Operation IceBridge altimetry for supraglacial lake depth retrievals, *The Cryosphere*, 14(11): 4253-4263.
- Xiao, W., Hui, F., Cheng, X., et al., 2023. An automated algorithm to retrieve the location and depth of supraglacial lakes from ICESat-2 ATL03 data, *Remote Sensing of Environment*, 298: 113730.

4) As far as the study by Liljedahl et al. (2021) is concerned, we would like to stress that it reveals variations in water pressure in a hydrological well below an ice sheet margin at the depth of at least 400 m. At the seasonal time scale, these variations are indeed significant (the peak-to-peak amplitude is about 15 m in terms of water head). However, it is important to understand that there is no one-to-one relationship between those pressure variations and variations in water mass. In opposite, those pressure variations systematically show a minimum in late autumn, and then steadily increase in the course of winter and spring, reaching the maximum in early summer. Therefore, as it has been stated by Liljedahl et al. (2021), that the pressure variations cannot be explained by an accumulation of meltwater, since water mass does not increase in the course of winter season. Liljedahl et al. (2021) attribute the revealed variability to “fluid pressure changes at the ice/earth boundary”. Unfortunately, the mechanism of those changes at the ice/earth boundary is not addressed explicitly. In any case, it would be fair to conclude that Liljedahl et al. (2021) show no evidences of a significant impact of groundwater on rock displacements observed at the surface.

As for “Further, the product has no treatment of groundwater”, we calculated the loadings of the PCRaster Global Water Balance (PCR-GLOBWB) and the WaterGAP2 Global Hydrology Model (WGHM) groundwater components (Lines 477-483) in Greenland at the KAGA site (Sutanudjaja et al., 2018; Müller et al., 2021). The vertical displacement caused by groundwater loads at KAGA station is less than 1 mm (see Fig. A11) and thereby, the effect on GNSS is small, less than 5% of the GNSS signal.

Fig. A11 The loadings of PCR-GLOBWB (a) and WGHM (b) groundwater components at the KAGA site. This figure is included as ED Fig. 13 in the revised manuscript.

Reference:

Sutanudjaja, E. H., Van Beek, R., Wanders, N., Wada, Y., Bosmans, J. H., Drost, N., ... & Bierkens, M. F. (2018). PCR-GLOBWB 2: a 5 arcmin global hydrological and water resources model. *Geoscientific Model Development*, 11(6), 2429-2453.

Müller Schmied, H., Cáceres, D., Eisner, S., Flörke, M., Herbert, C., Niemann, C., ... & Döll, P. (2021). The global water resources and use model WaterGAP v2. 2d: Model description and evaluation. *Geoscientific Model Development*, 14(2), 1037-1079.

Originality:

Storage—

Seasonal storage of meltwater is well-established. For example, prior work has measured time series of ‘bed separation,’ the local water storage in the subglacial drainage system (e.g., Andrews et al. 2014). These studies have the advantage of pinpointing the storage location but exist for only a few points and time periods. Other work has used radar to identify the spatial variability of water storage at the bed over 10s to 100s of km length scales (Chu et al., 2016). Ran et al. (2018) used GRACE data to show a time delay, peaking in midsummer, between RACMO’s meltwater generation and total mass loss (i.e., water storage). The finding is required since RACMO, MAR, and other regional climate models do not simulate water transport through hydrologic pathways and so they instantly remove water from the ice sheet. Thus, the disconnect peaks midsummer when meltwater generation is at maximum.

Authors: Previous studies focused on estimating “local water storage” with in-situ measurements (Andrews et al. 2014), extent of bed water with radar data (Chu et al., 2016), and large-scale water mass variations with GRACE data (Ran et al., 2018) are acknowledged in the new variant of the introductory section (Lines 65-76). It is worth to note that Ran et al. (2018) is our previous paper.

The current paper is the first one to demonstrate that GNSS-based vertical bedrock displacement can be used for observing and understanding hydrological processes within the GrIS (and, therefore, within other ice bodies on Earth). Furthermore, our study provides, for the first time, quantitative estimates of buffered water accumulation and release. Finally, our results have major implications for modelling the Surface Mass Balance (SMB) of the GrIS. We show that current SMB model runoff may contain systematic errors, requiring additional scaling of up to ~20% for the warmest years, in particular for the northern and northeastern GrIS. An adequate behavior of SMB models in warm years is particularly important for making accurate projections of the future behavior of the GrIS, when the climate is warmer than today. With the GrIS acting as the major contributor to future sea level rise, improving these projections has vast geographic, economic, and community implications.

The fact that regional climate models do not simulate water transport and, therefore, do not contain the signal of “buffered water” is at the core of our study. Among other, this fact is the starting point of our discussion in the Section “Analytic model of the buffered water signal in GNSS elastic loading data”.

>This paper presents time series of storage/release at 22 regions around the ice sheet. However, since the compartment/precise area driving the signal is not clear, it is also not clear how the results can be used to advance understanding of hydrologic systems.

Authors: The loading signals depend on the mass of the load and its distance from the GNSS station (e.g., see Wahr et al., 2013, in doi:10.1002/jgrb.50104). It is always possible to quantify the contribution of a particular part of the GrIS to the surface loading signal at a given location. As we have shown with our sensitivity analysis presented above (Fig. A3), this contribution is mostly limited to the radius of ~200 km (Lines 657-670). In view of that, each GPS station on bedrock can contribute to understanding buffered water storage processed around that station.

Scaling factor--

Large melt biases and uncertainties in RACMO/MAR due to factors such as clouds, albedo, and other is well established (e.g., Fettweis et al., 2020).

>This paper demonstrates that a scaling factor is required to force RCM melt to better match GNSS residuals. The scaling factors are highly variable, centered around 1 (Fig. 4), and has a relatively weak correlation with melt amount. How to incorporate the result into hydrologic studies is unclear, and so the results would seem to be most relevant to RCM testing and validation studies.

Authors: Thank you very much for the comments. We admit that the estimated scaling factors suffer from uncertainties, which may reach 10-20% (see Extended Data Table 2). However, when one looks at the scaling factors for warm, ordinary and cold years (Lines 254-257), it is quite clear to see the difference (see Fig. A12). At the same time, an application of these scaling factors is an intrinsic part of our methodology, which allows for a more accurate estimation of signal triggered by buffered water (see Extended Data Fig. 1). Furthermore, they offer a novel way to calibrate SMB models, so that they can better estimate the future behavior of the ice sheet in view of projected Arctic warming (see Sect. "Discussion"). Taking into account all these considerations, we decided to keep the estimation of the aforementioned scaling factors in the paper. In addition, Fettweis et al. (2020) was a previous study by some of our co-authors.

As for the remark "How to incorporate the result into hydrologic studies is unclear, and so the results would seem to be most relevant to RCM testing and validation studies", we would like to respond as follows. GNET data offer a novel and unique way to improve GrIS SMB estimates from regional climate models. Those models are currently the tool of choice for estimating GrIS-integrated surface melt rate. Despite their generally good and consistent performance, considerable uncertainties remain in the modelled melt products³⁸. Having independent estimates of adjustments required to improve/calibrate the melt products from regional climate models, as provided in this study, is therefore highly valuable for the Greenland mass balance research community. Among other, regional climate models require adjustments in this way for abnormally warm summers. This is particularly relevant in view of projected Arctic warming¹⁻⁵. Extremely high summer temperatures today may become normal in the foreseeable future. Thus, good model performance for warmer years is critical to project ice sheet behaviour and associated sea level rise across coming decades.

We have included this text in the Lines 189-199 in the "Discussion" section.

Fig. A12 The scaling factors for cold, ordinary, and warm years. This figure is the updated Fig. 4 in the revised manuscript.

=====

Werder MA, Hewitt IJ, Schoof CG and Flowers GE (2013) Modeling channelized and distributed subglacial drainage in two dimensions. *Journal of Geophysical Research: Earth Surface* 118(4), 2140–2158. doi:10.1002/jgrf.20146.

Hoffman, M.J., Andrews, L.C., Price, S.A., Catania, G.A., Neumann, T.A., Lüthi, M.P., Gulley, J.D., Ryser, C., Hawley, R.L., & Morriss, B.F. (2016). Greenland subglacial drainage evolution regulated by weakly connected regions of the bed. *Nature Communications*, 7.

Liu, L., Khan, S.A., Dam, T.V., Ma, J.H., & Bevis, M.G. (2017). Annual variations in GPS-measured vertical displacements near Upernavik Isstrøm (Greenland) and contributions from surface mass loading. *Journal of Geophysical Research: Solid Earth*, 122, 677 - 691.

Zwally, H.J., Abdalati, W., Herring, T.A., Larson, K.M., Saba, J.L., & Steffen, K. (2002). Surface Melt-Induced Acceleration of Greenland Ice-Sheet Flow. *Science*, 297, 218 - 222.

Liljedahl, L.C., Meierbachtol, T.W., Harper, J.T., van As, D., Näslund, J., Selroos, J., Saito, J., Follin, S., Ruskeeniemi, T., Kontula, A., & Humphrey, N.F. (2021). Rapid and sensitive response of Greenland's groundwater system to ice sheet change. *Nature Geoscience*, 14, 751 - 755.

Andrews, L.C., Catania, G.A., Hoffman, M.J., Gulley, J.D., Lüthi, M.P., Ryser, C., Hawley, R.L., & Neumann, T.A. (2014). Direct observations of evolving subglacial drainage beneath the Greenland Ice Sheet. *Nature*, 514, 80-83.

Li, W., Shum, C.K., Li, F., Zhang, S., Ming, F., Chen, W., Zhang, B., Lei, J., & Zhang, Q. (2020). Contributions of Greenland GPS Observed Deformation From

Multisource Mass Loading Induced Seasonal and Transient Signals. *Geophysical Research Letters*, 47.

Chu, W., Schroeder, D.M., Seroussi, H., Creyts, T.T., Palmer, S.J., & Bell, R.E. (2016). Extensive winter subglacial water storage beneath the Greenland Ice Sheet. *Geophysical Research Letters*, 43, 12,484 - 12,492.

Ran, J., Vizcaíno, M., Ditmar, P., van den Broeke, M.R., Moon, T.A., Steger, C.R., Enderlin, E.M., Wouters, B., Noël, B.P., Reijmer, C.H., Klees, R., Zhong, M., Liu, L., & Fettweis, X. (2018). Seasonal mass variations show timing and magnitude of meltwater storage in the Greenland Ice Sheet. *The Cryosphere*.

Fettweis, X., et al., (2020). GrSMBMIP: intercomparison of the modelled 1980–2012 surface mass balance over the Greenland Ice Sheet. *The Cryosphere*.

Reviewer Reports on the First Revision:

Referee #1 (Remarks to the Author):

I apologize for the delay in returning my review. Some significant real-world events got in the way of completing this.

The authors have addressed my specific comments from the first version well, and I think the same is true of the comments from the other reviewer. I read over all of the revisions and I am satisfied with the changes made by the authors. I look forward to publication of this very interesting paper.

Minor corrections

Line 434. Add “the” before “Pan-Arctic”.

Line 435. Add “the” before “GrIS”

Referee #1 (Remarks on code availability):

I did only a very quick scan of the code -- I did not test it.

Referee #3 (Remarks to the Author):

Referee #2 (Remarks to the Author):

Note: Additional referee evaluation of author responses shown in red. The authors made the changes clear and easy to follow. Nicely done!

The subject of this paper, seasonal storage and release of liquid water in the Greenland ice sheet, is a key component of many unsolved science problems related to the ice sheet's mass balance and ice flow dynamics. Because meltwater cannot exit the ice sheet instantaneously, the time/space evolution of hydrologic systems revolves around water storage. Storage is, therefore, a key state variable of all supraglacial, englacial, and subglacial hydrologic processes and models thereof. Water storage at the bed is a critical aspect of sliding mechanics and geochemical processes. Yet, the processes are too complex, and our understanding is too limited to explicitly include the many hydrologic processes (e.g., the seasonal evolution of a subglacial drainage system) in any large-scale SMB/runoff model.

Different observational methods, such as radar, GPS signals, GRACE, remote sensing imagery, radar and radiometry, have been used to determine water storage in various compartments. Each method presents resolution and interpretation challenges, and several offer only snapshots rather than continuous time series. Breakthroughs with direct measurements of storage thus have the potential to make a strong impact on many different aspects of Greenland hydrology.

Authors: Thank you very much for your time to review this paper and your excellent comments. We agree with your emphasis on the importance of studies on seasonal storage and release of liquid water in the Greenland ice sheet. We are very grateful for your valuable comments and have addressed them one-by-one as follows.

Concerns:

1. Hydrologic interpretation.

Throughout the paper, the description of the water storage mechanism is quite puzzling. Examples include:

-- Figure 1 depicts englacial lakes (and subglacial lakes that are actually shown as englacial) — I know of no evidence for lakes existing inside the ice.

Authors: Thank you very much for the comment. Sorry for the misunderstanding. Figure 1 is a *schematic* figure to show the processes related to the liquid water evolution beneath the ice sheet surface and the associated changes of vertical bedrock. The distance of englacial lakes from the ice sheet surface is schematic, just indicating that some water is buffered beneath the ice sheet surface, and does not mean the actual location of englacial lakes is inside the ice sheet. However, based on the above comments, we have improved Fig. 1, to avoid any misunderstanding. For your convenience, the revised Fig. 1 is shown below. Please see lines 218-232 in the revised manuscript.

The representation of relevant hydrological features in the new Figure 1 seems fine to me, with the exception that differences in the relationship between the white dashed line and the bedrock mound on which the GNSS station is mounted are hard to discern in the 3 panels.

Fig. A2 The revised Fig. 1.

-- The text concerning loading is confusing because a phase change is not a mass change, so water mass loading must be due to the redistribution of water from one place to another (not stated).

Authors: Thanks for this thoughtful comment. The "phase change" refers to melting

and refreezing processes. In order to avoid potential confusion, we have added the following sentence in the introductory section (Lines 84-85):

“Note that bedrock displacement time-series describe the mass of buffered water rather than the total water mass, since surface loading does not change if water re-freezes in-place.”

This clarification is helpful, but I also struggled with the lack of clarity around water mass change: the mass needs to move in order to elicit elastic deformation. Throughout the text, it almost seemed like the GNSS signals were being related to water mass, rather than a change in water mass (see also my independent comments).

-- The manuscript intro/discussion is structured around water storage in the firn compartment, but why this is driving the loading signal versus other compartments is not clear. Melt across most of the firn area will result in a short-distance mass redistribution, vertically within the upper 10 m. Only a relatively narrow reach of the far lower accumulation area has substantial horizontal flow and accumulation of water, but is there evidence that this is enough to drive the loading signal? Regardless, all runoff is accumulated toward the margin. Whereas a SMB model may simulate a large amount of meltwater generation across the accumulation zone, the subglacial drainage system, not the firn compartment, has the most accumulating water mass and is therefore the most likely candidate for loading. Water at the bed cannot be dismissed by invoking fast conduit drainage. A large body of literature on subglacial hydrology debates issues such as a) how far inward on the ice sheet fast-draining conduits can actually develop (e.g., Werder et al., 2013); and, b) how much area of the bed sits between conduits and does not drain quickly (e.g., Hoffman et al., 2016).

Authors: We agree that the introductory section was too much focused on processes in the firn layer, which resulted in some mismatch between that section and the rest of the manuscript. To solve this issue, we have re-organized the introductory section to put more emphasis on water at the bed (Lines 65-76). The firn layer is mentioned in the Discussion Section only briefly, and primarily in the context of our interpretation of the scaling factors to be applied to runoff estimates provided by the SMB model (Lines 189-199).

Firn processes did not seem over-emphasized in the revised manuscript. This point, however, also relates to the lengthscales over which mass must move to create detectable deformation.

As for “A large body of literature on subglacial hydrology debates issues such as a) how far inward on the ice sheet fast-draining conduits can actually develop (e.g., Werder et al., 2013); and, b) how much area of the bed sits between conduits and does not drain quickly (e.g., Hoffman et al., 2016).”, the loading observed using GNSS verticals is the bedrock response to integrated mass changes. We have removed SMB and other non-ice sources, attributing the remaining changes to temporally buffered

meltwater. But it is difficult to further differentiate the pathways or storages of the buffered meltwater by using GNSS data alone. We hope this study will stimulate further research to address ice sheet hydrology from a new perspective like this.

I don't think the reviewer is asking the authors to attempt to extract more detail from the GNSS data related to the style of subglacial drainage or the architecture of the system. I think the point here was to say that water in the subglacial drainage system can have a long residence time even though channels may form, i.e., "Water at the bed cannot be dismissed by invoking fast conduit drainage". I'm not sure any further action is needed, but readers would like to know that the authors have thoroughly considered the plausibility of the mechanisms proposed to explain the GNSS data.

2. Signal robustness.

Because the water storage signal is a small residual that emerges after many large corrections to the original data, the robustness of the signal is dependent on the confidence of the removal of so-called nuisance signals. Perhaps some of the following issues are partly or solely responsible for the spatial variability of the results.

Authors: We agree that this signal is relatively small, but it is clearly visible (see, e.g., panel (h) in Extended Data Fig. 4 in the updated manuscript, please see Lines 851-853) and reliably estimated in this study. To ensure the robustness of our estimates, we conducted a thoughtful analysis by evaluating all corrections with the State-of-the-Art models or data in the community. Please see revised ED Fig. 5 (Lines 854-863) and the following text. It is worth to note that we also compared the GNSS-based water storage signal with that based on independent satellite gravimetry data (Fig. 3, see Lines 245-251). This comparison confirmed that this signal is real and not a data artefact.

I'm going to assume this response will be evaluated by someone with GNSS signal processing expertise.

-- Prior work by Lui et al. (2017) demonstrated the difficulty of SMB corrections in Greenland due to far-field SMB influences. In other words, where does the SMB signal come from? Lui et al. (2017) argue that SMB from glaciers 1200 km away can impact Greenland GNSS signals. In fact, they found that correcting for SMB adversely inflated the residual. The manuscript does not make clear how this problem was handled nor is the Lui et al. (2017) paper cited.

Authors: Thank you very much for the comment. Liu et al. (2017) was the previous study by one of the co-authors. Liu et al. (2017) studied Upernavik glacier using two GNSS stations located in the nearby bedrock area. We agree with Liu et al. (2017) that far-field SMB signals must be taken into account when interpreting the annual amplitude of the GNSS verticals. In this study we use an ice-sheet-wide SMB model (RACMO) rather than a local or regional model and therefore correct for the far-field signal. We now state in line 376 that the SMB model covers the entire GrIS (and not just a small area).

In addition, we made a sensitivity study (Lines 657-670) to clarify how mass changes over the entire ice sheet, including glaciers 1,200 km away, may affect the observed mass loading signal. By taking the detrended SMB in July 2012 (Fig. A3 in the rebuttal) as the ice sheet mass change signal, we analyze the sensitivity of GNSS loading observed at KAGA as an example. The processing strategy is to take the KAGA station as the center, create ring-shaped zones outward with a step width of 50 km, calculate the vertical displacement caused by SMB inside each ring, and then normalized with the entire Greenland. The results show that cumulative contribution to the detrended SMB loading reaches approximately 80% when the distance from KAGA station is 200 km. The sensitivity curve (Fig. A3) reveals a significant contribution from near field: SMB changes within 200 km from KAGA contribute by about 80% to the detrend SMB loading displacements. There is little sensitivity to SMB beyond 500 km from KAGA.

Fig. A3 (a) Detrended SMB mass anomalies in July 2012. (b) Cumulative contribution to the detrended uplift at KAGA due to SMB loading, depending on the radius of the buffer zone around that station. All the numbers are normalized with the loading uplift caused by the SMB signal over entire Greenland. The X axis shows a step of 50 km in the distance range of 0-1,500 km. This figure is included as ED Fig. 14 in the revised manuscript.

I don't have much expertise in this area, but the response and added analysis seem reasonable. Length scales of influence were one of my big questions, so I would like to see some related information in the intro to the main text.

-- The analysis does not appear to consider the seasonal loading signal caused by ice flow across the ablation zone (c.f., unloading by calving). The speed of the ice sheet in the ablation zone increases during summer, typically by a factor of 2-3 (see Zwally et al. (2002) for the seminal paper, with much confirming work published since then). Speed typically peaks mid-summer, similar to the GNSS loading signal. Ice acceleration causes horizontal advection of ice thickness gradients — more ice is

moved low that must be slowly beaten back by ablation. How much of the residual could be due to this effect?

Authors: We agree with the reviewer that the seasonal loading signal might be caused by the ice flows. In this study, we have considered this effect. Please see the effect of seasonal variations in ice speed analyzed in the Section “Contribution of ice discharge”, which is a part of Methods (see lines 416-429). To obtain an upper bound of the contribution of seasonal ice discharge variations to the observed vertical displacements, we use Jakobshavn Isbræ (JI) as an example because it shows the largest ice flow velocity seasonality amongst GrIS outlet glaciers. By investigating the GNSS loading signal observed at the KAGA GPS station, which is located ~5 km from the JI terminus, we conclude that this effect (i.e., <~1mm) is not significant.

Seems reasonable.

-- The product used for LWS corrections is a global product and is not necessarily well-suited for Arctic regions dominated by permafrost hydrology. Some areas have many lakes fluctuating lake levels. Further, the product has no treatment of groundwater, whereas Liljedahl et al., (2021) show very large head swings in the groundwater system along the ice margin. LWS is a big correction and uncertainties will vary substantially around the ice sheet.

I share the concerns about groundwater.

Authors: Thanks for this comment. We agree that lakes and groundwater play important roles in Arctic regions dominated by permafrost hydrology. Indeed, the GLDAS model we considered in the previous variant of the manuscript lacks both water storage components. To solve this problem, we also add two hydrological models (see Extended Data Fig. 5c in Line 854), i.e., WaterGAP2 Global Hydrology Model (WGHM) and PCRaster Global Water Balance (PCR-GLOBWB), both of which take groundwater into account. Furthermore, we explicitly consider water mass variations of lakes in Arctic regions (Lines 431-475).

1) Three State-of-The-Art models are utilized in this study in total: GLDAS, PCR-GLOBWB and WGHM. Even though these models are not specific for Arctic regions, we believe they are accurate enough at least for the first-order magnitude **as evidenced by?**. Note that WGHM is tailored in Arctic region by considering the existence of permafrost. A comparison with other models showed that WGHM agrees best with independent in-situ observations collected in Arctic region (Gädeke et al., 2020). In addition, as we have shown with our sensitivity analysis of GNSS-based loading signal presented above (Fig. A3), this contribution is mostly (~80%) limited to the radius of ~200 km. Most parts of the Arctic regions are quite far (> 1000 km) from the GNSS stations in Greenland, and thereby the mass variations over there are little impact the GNSS based water storage signals observed in this study.

The highlighted text above is a good argument for the lakes, but not for groundwater around Greenland.

Taking the KAGA as an example: the vertical displacement at KAGA in response to global LWS loading derived from the 3 hydrological models, excluding Greenland, during the study period (2009-2015) ranges from -1.0 to ~1.5 mm (Fig. A4). This magnitude is sufficiently small to allow for neglecting the potential influence of LWS on the Greenland ice sheet meltwater storage.

What exactly do these models estimate as loading: water mass on the land surface? Is it possible that any of the elastic deformation could be attributable to seasonal recharge/discharge of the aquifer? Do the models include estimates of aquifer compressibility (assuming fractured rock aquifers) such that the vertical changes associated with possible compaction/expansion due to seasonal changes in hydraulic head?

I have trouble agreeing with the final highlighted statement if I am interpreting the data correctly: Fig A4 below shows variations due to LWS of +/-1 mm that were neglected in the corrections? +/-1mm does not seem small compared to the signals in Figure 2, many of which are substantially less than +/- 5mm. They also appear to be out of phase so would make the signals in Fig 3 less pronounced if used in the correction.

Without knowing enough to have confidence in the models' treatment of groundwater around Greenland, I am hesitant to dismiss this correction as negligible. Depending on what the models are doing, one solution may be not to attempt to correct for groundwater or claim it is negligible, but rather include it in what is being measured. If most of the groundwater recharge is coming from the ice sheet, then this is a source of water that would ideally be accounted for anyway.

Fig. A4 Time series of the vertical displacements at KAGA in response to LWS loading

excluding Greenland.

References:

GLDAS: Li, B., M. Rodell, S. Kumar, H. Beaudoin, A. Getirana, B. F. Zaitchik, et al. (2019) Global GRACE data assimilation for groundwater and drought monitoring: Advances and challenges. *Water Resources Research*, 55, 7564-7586. doi:10.1029/2018wr024618

PCR-GLOWB: Sutanudjaja, E. H., Van Beek, R., Wanders, N., Wada, Y., Bosmans, J. H., Drost, N., ... & Bierkens, M. F. (2018). PCR-GLOBWB 2: a 5 arcmin global hydrological and water resources model. *Geoscientific Model Development*, 11(6), 2429-2453.

WGHM: Müller Schmied, H., Cáceres, D., Eisner, S., Flörke, M., Herbert, C., Niemann, C., ... & Döll, P. (2021). The global water resources and use model WaterGAP v2. 2d: Model description and evaluation. *Geoscientific Model Development*, 14(2), 1037-1079.

Gädeke, A., Krysanova, V., Aryal, A. et al. (2020). Performance evaluation of global hydrological models in six large Pan-Arctic watersheds. *Climatic Change* 163, 1329–1351, <https://doi.org/10.1007/s10584-020-02892-2>

2) As for “Some areas have many lakes fluctuating lake levels.”, indeed, there are many lakes in the Arctic region. To further quantify the potential water mass impact of lakes onto GNSS loading signal in Greenland, we consider three types of lakes (Lines 431475): lakes in Pan-Arctic region (north of 60 °N) in general; supraglacial lakes (SGL) on GrIS; and proglacial lakes in the coastal part of Greenland.

Fig. A5 Distribution of lakes in Pan-Arctic region. This figure is included as ED Fig. 7 in the revised manuscript.

Lakes in Pan-Arctic region: Specifically, the monthly lake water storage data of some large lakes were directly downloaded from a recently published dataset by Yao et al., (2023). In addition, we accessed the intra-annual lake level datasets from three portals, i.e., Hydroweb (<https://hydroweb.theia-land.fr/>), Global Reservoirs and Lakes Monitor (G-REALM, https://ipad.fas.usda.gov/cropexplorer/global_reservoir/), and Database for Hydrological Time Series of Inland Waters (DAHITI, <https://dahiti.dgfi.tum.de/en/>), respectively (Cretaux et al., 2011; Birkett et al., 2009; Schwatke et al., 2015). For months without lake area data in the dataset provided by Yao et al., (2023), we follow the empirical relationship between lake water level and lake surface area from a similar previous study (Song et al., 2013), by interpolating the missing lake area data with ampler level data and converting the lake area and level data to water storage changes.

It is worth mentioning that the recorded lakes (38 in total, indicated by blue circles in Fig. A5) are typically large, with 33 of them having a surface area greater than 500 km², accounting for 61.6% of the total surface area of Arctic lakes extracted by Yao et al. (2023). Note that the reference surface area here is derived from the HydroLAKES polygon (Messenger et al., 2016). For the remaining smaller inland lakes (red circles in Fig. A5), the area data are very limited. Therefore, we employ a scale-up strategy to estimate their impact on mass loading (Song et al., 2013).

Taking KAGA station as an example, the vertical displacement induced by the load of Arctic lake water storage during the study period (2009-2015) ranges from -0.1 to 0.04 mm (Fig. A6). This magnitude is sufficiently small to allow for neglecting the potential influence of these lakes in the Arctic region.

Fig. A6 Vertical displacements of lake loading at KAGA station. This figure is included as ED Fig. 8 in the revised manuscript.

I had trouble grasping how lake-level variations in distal locations like Russia would influence the GNSS signals around Greenland. It would seem necessary to know where

mass lost/gained from the lakes goes in order to correctly account for this signal. It does not seem sufficiently important to pursue further.

SGL on the GrIS: In addition, we assessed the vertical displacement caused by water mass loading from SGL on the GrIS. Firstly, we divided the GrIS into 10-by-10 km equal-area cells and identified all the cells with SGL using ~ 300,000 high spatiotemporal resolution optical images from Sentinel-2 and Landsat 8/9 over 2017-2022, as shown in Fig. A7. Then, the monthly SGL area changes during the melting season (i.e., May-September) were derived. The SGL area changes data are distributed via <https://doi.org/10.5281/zenodo.10398558>. Note that it is problematic to obtain the monthly SGL area change for entire GrIS before the launch of Sentinel-2B satellites in 2017 due to a high cloud contamination. We assumed, however, that the SGL area changes over 2009-2015 interval was similar to those over 2017-2022. When converting the SGL area changes to mass changes, we assumed that the maximum depth of GrIS supraglacial lakes is around 8.5 metres (Fair et al., 2020; Xiao et al., 2023), which allowed us to estimate the upper limit of water mass changes Fig. A8. The result shows that even for the maximal depth of 8.5 m, the magnitude of loading signal caused by SGL mass changes is ~0.3 mm, which is negligible, compared with the BWS signal.

I had trouble grasping the assumptions in this analysis. If a supraglacial lake drains vertically downward, and the water is retained locally at the bed, there would presumably be no deformation measured at the GNSS station, right? Are the authors assuming that loss of lake mass instantaneously contributes to englacial/subglacial water mass somewhere significantly downstream? This would not be unreasonable, but I just don't know. Similarly, what is assumed when lakes increase in size? My assumption would be no deformation since the meltwater is likely to be (relatively) locally sourced and does not, therefore, represent a substantial lateral redistribution of mass. Here again, relating the processes envisioned to the measured signals would help a broad audience understand the assumptions and better appreciate this work.

Proglacial lakes in Greenland: Similar to the SGL, we also consider the impact of proglacial lakes in Greenland using Hydrolake database (Fig. A9). In total, 2 687 proglacial lakes are taken into account. Most of the proglacial lakes are smaller than 5 km², while the largest one could reach ~100 km². The loading signal caused by proglacial lakes is small. At KAGA station, for instance, it is of the order of only 0.02 mm (Fig. A10).

This signal size seems suitably small. I think I can imagine the assumptions being made to relate the change in lake size with station elevation, but it would not hurt to state them explicitly.

Fig. A7 Spatial distribution of cells with SGLs. The grey cells indicate that there were no SGL over 2017-2022 from optical images. This figure is included as ED Fig. 9 in the revised manuscript.

Fig. A8 The vertical displacements at KAGA station caused by SGL mass changes. This figure is included as ED Fig. 10 in the revised manuscript.

Fig. A9 Spatial distribution of proglacial lakes in Greenland. This figure is included as ED Fig. 11 in the revised manuscript.

Fig. A10 | The vertical displacements at KAGA station caused by proglacial lake mass changes over entire Greenland. This figure is included as ED Fig. 12 in the revised manuscript.

References:

Cretaux J-F., Arsen A., Calmant S., et al., 2011. SOLS: A lake database to monitor in

the Near Real Time water level and storage variations from remote sensing data, *Advances in space Research*, 47, 1497-1507.

Birkett, C.M., Reynolds, C., Beckley, B., et al., 2009. From Research to Operations: The USDA Global Reservoir and Lake Monitor, chapter 2 in 'Coastal Altimetry', Springer Publications, eds. S. Vignudelli, A.G. Kostianoy, P. Cipollini and J. Benveniste, ISBN 978-3-642-12795-3, 2010.

Schwatke C, Dettmering D, Bosch W, et al., 2015. DAHITI—an innovative approach for estimating water level time series over inland waters using multi-mission satellite altimetry, *Hydrology and Earth System Sciences*, 19(10), 4345-4364.

Yao F, Livneh B, Rajagopalan B, et al., 2023. Satellites reveal widespread decline in global lake water storage, *Science*, 380(6646): 743-749.

Song, C., Huang, B., Ke, L., 2013. Modeling and analysis of lake water storage changes on the Tibetan Plateau using multi-mission satellite data. *Remote Sensing of Environment*. 135. 25-35.

Messenger, M.L., Lehner, B., Grill, G., et al., 2016. Estimating the volume and age of water stored in global lakes using a geo-statistical approach. *Nature Communications*, 7: 13603.

Fair, Z., Flanner, M., Brunt, K. M., et al., 2020. Using ICESat-2 and Operation IceBridge altimetry for supraglacial lake depth retrievals, *The Cryosphere*, 14(11): 4253-4263.

Xiao, W., Hui, F., Cheng, X., et al., 2023. An automated algorithm to retrieve the location and depth of supraglacial lakes from ICESat-2 ATL03 data, *Remote Sensing of Environment*, 298: 113730.

4) As far as the study by Liljedahl et al. (2021) is concerned, we would like to stress that it reveals variations in water pressure in a hydrological well below an ice sheet margin at the depth of at least 400 m. At the seasonal time scale, these variations are indeed significant (the peak-to-peak amplitude is about 15 m in terms of water head). However, it is important to understand that there is no one-to-one relationship between those pressure variations and variations in water mass. In opposite, those pressure variations systematically show a minimum in late autumn, and then steadily increase in the course of winter and spring, reaching the maximum in early summer. Therefore, as it has been stated by Liljedahl et al. (2021), that the pressure variations cannot be explained by an accumulation of meltwater, since water mass does not increase in the course of winter season. Liljedahl et al. (2021) attribute the revealed variability to "fluid pressure changes at the ice/earth boundary". Unfortunately, the mechanism of those changes at the ice/earth boundary is not addressed explicitly. In any case, it would be fair to conclude that Liljedahl et al. (2021) show no evidences of a significant impact of groundwater on rock displacements observed at the surface.

This paper may plausibly be all the authors have to go on, but the lag times for groundwater transport can mean that hydraulic head variations can be out of phase with recharge from distal locations. The phase would also vary with depth due to the variation in travel pathlength

and residence time with depth. Closer to the surface than the borehole described above, phase lags between recharge and hydraulic head could be much shorter. This makes me skeptical about dismissing groundwater as a signal source based on the arguments above.

As for “Further, the product has no treatment of groundwater”, we calculated the loadings of the PCRaster Global Water Balance (PCR-GLOBWB) and the WaterGAP2 Global Hydrology Model (WGHM) groundwater components (Lines 477-483) in Greenland at the KAGA site (Sutanudjaja et al., 2018; Müller et al., 2021). The vertical displacement caused by groundwater loads at KAGA station is less than 1 mm (see Fig. A11) and thereby, the effect on GNSS is small, less than 5% of the GNSS signal.

5% of the original signal prior to correction? It is non-negligible compared to the final displacements in Fig 2.

Fig. A11 The loadings of PCR-GLOBWB (a) and WGHM (b) groundwater components at the KAGA site. This figure is included as ED Fig. 13 in the revised manuscript.

Reference:

Sutanudjaja, E. H., Van Beek, R., Wanders, N., Wada, Y., Bosmans, J. H., Drost, N., ... & Bierkens, M. F. (2018). PCR-GLOBWB 2: a 5 arcmin global hydrological and water resources model. *Geoscientific Model Development*, 11(6), 2429-2453.

Müller Schmied, H., Cáceres, D., Eisner, S., Flörke, M., Herbert, C., Niemann, C., ... & Döll, P. (2021). The global water resources and use model WaterGAP v2. 2d: Model description and evaluation. *Geoscientific Model Development*, 14(2), 10371079.

I don't have additional specific analyses to suggest to deal with groundwater contributions to the signal.

Originality:

Storage—

Seasonal storage of meltwater is well-established. For example, prior work has measured time series of ‘bed separation,’ the local water storage in the subglacial drainage system (e.g., Andrews et al. 2014). These studies have the advantage of pinpointing the storage location but exist for only a few points and time periods. Other

work has used radar to identify the spatial variability of water storage at the bed over 10s to 100s of km length scales (Chu et al., 2016). Ran et al. (2018) used GRACE data to show a time delay, peaking in midsummer, between RACMO's meltwater generation and total mass loss (i.e., water storage). The finding is required since RACMO, MAR, and other regional climate models do not simulate water transport through hydrologic pathways and so they instantly remove water from the ice sheet. Thus, the disconnect peaks midsummer when meltwater generation is at maximum.

Authors: Previous studies focused on estimating "local water storage" with in-situ measurements (Andrews et al. 2014), extent of bed water with radar data (Chu et al., 2016), and large-scale water mass variations with GRACE data (Ran et al., 2018) are acknowledged in the new variant of the introductory section (Lines 65-76). It is worth to note that Ran et al. (2018) is our previous paper.

The current paper is the first one to demonstrate that GNSS-based vertical bedrock displacement can be used for observing and understanding hydrological processes within the GrIS (and, therefore, within other ice bodies on Earth). Furthermore, our study provides, for the first time, quantitative estimates of buffered water accumulation and release. Finally, our results have major implications for modelling the Surface Mass Balance (SMB) of the GrIS. We show that current SMB model runoff may contain systematic errors, requiring additional scaling of up to ~20% for the warmest years, in particular for the northern and northeastern GrIS. An adequate behavior of SMB models in warm years is particularly important for making accurate projections of the future behavior of the GrIS, when the climate is warmer than today. With the GrIS acting as the major contributor to future sea level rise, improving these projections has vast geographic, economic, and community implications.

The fact that regional climate models do not simulate water transport and, therefore, do not contain the signal of "buffered water" is at the core of our study. Among other, this fact is the starting point of our discussion in the Section "Analytic model of the buffered water signal in GNSS elastic loading data".

I think the study is novel in using surface deformation to make inferences about seasonal changes in water-mass distribution, provided sufficient confidence can be established in the non-hydrological corrections to the signal. The revised manuscript text and responses to the reviewer still leave some mechanistic gaps for me: what presumably matters for vertical elastic deformation measurable at the GNSS stations is sufficient lateral mass movement within the radius of sensitivity. How each of the appearances and disappearances of "observable" mass translate into measurable redistributions of mass that contribute to the GNSS signal is not always clear. The paper would benefit from a clearer and more fulsome articulation of the physical processes involved, with attention to distinguishing between processes that would and would not contribute to the GNSS signal, including an explanation for the sign of the contribution.

>This paper presents time series of storage/release at 22 regions around the ice sheet. However, since the compartment/precise area driving the signal is not clear, it is also not clear how the results can be used to advance understanding of hydrologic systems.

Authors: The loading signals depend on the mass of the load and its distance from the GNSS station (e.g., see Wahr et al., 2013, in doi:10.1002/jgrb.50104). It is always possible to quantify the contribution of a particular part of the GrIS to the surface loading signal at a given location. As we have shown with our sensitivity analysis presented above (Fig. A3), this contribution is mostly limited to the radius of ~200 km (Lines 657-670). In view of that, each GPS station on bedrock can contribute to understanding buffered water storage processed around that station.

I don't think the authors have described this well enough in a manuscript intended for a journal of interest to all scientists. The details are buried in an extended figure and the basic physics not explained in the introduction. The radially symmetric influence is what I initially assumed, leading to my confusion about why the mass-change accounting was done on the basis of irregularly shaped hydrological catchments (see independent comments). A short paragraph is needed in the manuscript introduction that clearly states the premise of the analysis in sufficient detail that it also makes sense to glaciologists familiar with the mass-balance and hydrological regimes of the Greenland ice sheet.

Scaling factor--

Large melt biases and uncertainties in RACMO/MAR due to factors such as clouds, albedo, and other is well established (e.g., Fettweis et al., 2020).

>This paper demonstrates that a scaling factor is required to force RCM melt to better match GNSS residuals. The scaling factors are highly variable, centered around 1 (Fig. 4), and has a relatively weak correlation with melt amount. How to incorporate the result into hydrologic studies is unclear, and so the results would seem to be most relevant to RCM testing and validation studies.

Authors: Thank you very much for the comments. We admit that the estimated scaling factors suffer from uncertainties, which may reach 10-20% (see Extended Data Table 2). However, when one looks at the scaling factors for warm, ordinary and cold years(Lines 254-257), it is quite clear to see the difference (see Fig. A12). At the same time, an application of these scaling factors is an intrinsic part our methodology, which allows for a more accurate estimation of signal triggered by buffered water (see Extended Data Fig. 1). Furthermore, they offer a novel way to calibrate SMB models , so that they can better estimate the future behavior of the ice sheet in view of projected Arctic warming (see Sect. "Discussion"). Taking into account all these considerations, we decided to keep the estimation of the aforementioned scaling factors in the paper. In addition, Fettweis et al. (2020) was a previous study by some of our co-authors.

I would suggest labelling the result "tuned" rather than "calibrated". I would also suggest dialing back the presentation of the scaling of the model as a robust result,

without a bit of text reassuring the reader that there is no circularity in the use of the SMB.

As for the remark “How to incorporate the result into hydrologic studies is unclear, and so the results would seem to be most relevant to RCM testing and validation studies”, we would like to respond as follows. GNET data offer a novel and unique way to improve GrIS SMB estimates from regional climate models. Those models are currently the tool of choice for estimating GrIS-integrated surface melt rate. Despite their generally good and consistent performance, considerable uncertainties remain in the modelled melt products³⁸. Having independent estimates of adjustments required to improve/calibrate the melt products from regional climate models, as provided in this study, is therefore highly valuable for the Greenland mass balance research community. Among other, regional climate models require adjustments in this way for abnormally warm summers. This is particularly relevant in view of projected Arctic warming¹⁻⁵. Extremely high summer temperatures today may become normal in the foreseeable future. Thus, good model performance for warmer years is critical to project ice sheet behaviour and associated sea level rise across coming decades.

We have included this text in the Lines 189-199 in the “Discussion” section.

I think the main value added here is having an independent means---albeit an indirect one that requires corrections larger than the signal of interest (not unlike GRACE)---to infer hydrological quantities.

Fig. A12 The scaling factors for cold, ordinary, and warm years. This figure is the updated Fig. 4 in the revised manuscript.

I see the point of the above figure, but is there any more sensible way to organize the x axis? Is it already organized by latitude or another physical variable? Perhaps using dots (or box-whisker) instead of lines would be better since there is no meaning in the lines connecting the dots.

====

Werder MA, Hewitt IJ, Schoof CG and Flowers GE (2013) Modeling channelized and distributed subglacial drainage in two dimensions. *Journal of Geophysical Research: Earth Surface* 118(4), 2140–2158. doi:10.1002/jgrf.20146.

Hoffman, M.J., Andrews, L.C., Price, S.A., Catania, G.A., Neumann, T.A., Lüthi, M.P., Gulley, J.D., Ryser, C., Hawley, R.L., & Morriss, B.F. (2016). Greenland subglacial drainage evolution regulated by weakly connected regions of the bed. *Nature Communications*, 7.

Liu, L., Khan, S.A., Dam, T.V., Ma, J.H., & Bevis, M.G. (2017). Annual variations in GPS-measured vertical displacements near Upernavik Isstrøm (Greenland) and contributions from surface mass loading. *Journal of Geophysical Research: Solid Earth*, 122, 677 - 691.

Zwally, H.J., Abdalati, W., Herring, T.A., Larson, K.M., Saba, J.L., & Steffen, K. (2002). Surface Melt-Induced Acceleration of Greenland Ice-Sheet Flow. *Science*, 297, 218 - 222.

Liljedahl, L.C., Meierbachtol, T.W., Harper, J.T., van As, D., Näslund, J., Selroos, J., Saito, J., Follin, S., Ruskeeniemä, T., Kontula, A., & Humphrey, N.F. (2021). Rapid and sensitive response of Greenland's groundwater system to ice sheet change. *Nature Geoscience*, 14, 751 - 755.

Andrews, L.C., Catania, G.A., Hoffman, M.J., Gulley, J.D., Lüthi, M.P., Ryser, C., Hawley, R.L., & Neumann, T.A. (2014). Direct observations of evolving subglacial drainage beneath the Greenland Ice Sheet. *Nature*, 514, 80-83.

Li, W., Shum, C.K., Li, F., Zhang, S., Ming, F., Chen, W., Zhang, B., Lei, J., & Zhang, Q. (2020). Contributions of Greenland GPS Observed Deformation From Multisource Mass Loading Induced Seasonal and Transient Signals. *Geophysical Research Letters*, 47.

Chu, W., Schroeder, D.M., Seroussi, H., Creyts, T.T., Palmer, S.J., & Bell, R.E. (2016). Extensive winter subglacial water storage beneath the Greenland Ice Sheet. *Geophysical Research Letters*, 43, 12,484 - 12,492.

Ran, J., Vizcaíno, M., Ditmar, P., van den Broeke, M.R., Moon, T.A., Steger, C.R., Enderlin, E.M., Wouters, B., Noël, B.P., Reijmer, C.H., Klees, R., Zhong, M., Liu, L., & Fettweis, X. (2018). Seasonal mass variations show timing and magnitude of meltwater storage in the Greenland Ice Sheet. *The Cryosphere*.

Fettweis, X., et al., (2020). GrSMBMIP: intercomparison of the modelled 1980–2012 surface mass balance over the Greenland Ice Sheet. *The Cryosphere*.

This review focuses primarily on the authors' responses to Reviewer #2.

I have made comments directly in the response-to-reviewer file (in red) to address the authors' responses. Below I have made a few comments of my own while I read the revised manuscript, prior to reading the reviews.

I was surprised that in the opening comments, a (subglacial) groundwater system is neglected as a possible pathway for water export that originates as surface melt. Supraglacial, englacial, subglacial (specified to be ice-bed interface) pathways are mentioned, but even in a bedrock substrate there must be a non-zero component of subglacial groundwater recharge and transport where the ice-bed interface is temperate. Groundwater seems important to acknowledge in a study focused on water.

Line 66 onward: I personally find the use of “hydrostatic pressure” for water pressure confusing, as some might use “hydrostatic pressure” to refer to the ice overburden pressure, which is not what the authors mean. Suggest using instead “basal water pressure” or “basal effective pressure”.

Lines 69-70: “However, ice sheet regions with a channelised drainage system may alternate with regions lacking an efficient drainage, the latter retaining high hydrostatic pressure.” “Alternate” is not quite the right notion. The glacier bed and basal drainage system are highly heterogeneous in space, so even regions where channels are present could be dominated by high basal water pressure if the drainage system is not hydraulically well connected to the channels.

Line 74: “An ice penetrating radar can detect basal water”. Sorta, but this vastly oversimplifies. IPR can detect frozen/thawed bed and various other properties from which something about water might be inferred. Adjust wording slightly to improve precision, e.g., even “can detect properties indicative of basal water” would make this more palatable.

Lines 80-82: “...bedrock's elastic response to total mass variations in the vicinity of a given GNSS station”: This is a critical summary of the method and, in my view, should make the mechanism that relates the process to the observable clear for a broad readership: is GNSS mounted on bedrock outside of the ice sheet registering crustal flexure due to variable transient amounts of water mass being stored in the vicinity (all around the GNSS station, including on/in/under the ice)? Reporting the length-scale of influence/sensitivity would also be used here: over what radial distance/area is a given station GNSS sensitive? By what mechanisms do the different sources of GNSS signal contribute? For example, how extensive must the water redistribution be to be measurable? Is it correct that only horizontal redistribution matters? Some related information is presented deep within the extended

figures, but a brief synopsis would help readers here. I felt there was a mechanistic gap throughout the paper, as it was assumed the reader would immediately understand that GNSS-measured bedrock elevation changes could be equated to the mass of water leaving the ice sheet. It would be appreciated if the authors would connect those dots more clearly.

Line 95 cf. 88: Somewhat confusing at this point in the manuscript that GNSS data appear to be used to test SMB models (question iii, line 88), but that model output is being used to correct the GNSS signal (line 95). Can a sentence of clarification be introduced to dispel concern that there is circularity in the process? The scaling is reported as a major result in the opening paragraph of the paper, so this seems important.

Lines 97-102: It would help if stated explicitly that downward motion equates to accumulation of stored water and vice versa, presumably due to proximity of GNSS to locations of accumulation. I'm thinking about mass changes in general and how station position relative to the mass change might lead to downward (proximal for mass accumulation) vs upward (distal from mass accumulation) responses.

Line 124: "tuned" may be better than "calibrated" since this is an empirical scaling
Line 128-129: "total BWS". Doesn't the mass need to move substantially from its place of origin to cause crustal elevation change (i.e., meltwater that accumulates locally in a supraglacial lake would not change the mass distribution substantially enough to be detected in the form of bedrock elevation changes outside the ice sheet)?

Line 547: "relatively" => "relative"

Eqn (A9): somewhere around here, shouldn't there be a reference to a horizontal length scale? A station represents a point location that is sensitive to mass change within some region. Eqn (A8) expresses mass change at an individual (mass balance) grid cell which is assumed linearly related to elastic deformation in (A9), but then the spatial relationship between contributing (ice-sheet) grid cells and the (off-ice) station location disappears. In the equations that follow it appears that the BWS inferred at any station is being related to catchment-wide quantities. If this interpretation is correct---that each GNSS station is assumed sensitive only to mass changes that occur in its hydrological catchment (the furthest points of which may be further away than adjacent catchments)--- it would help readers like me if this were stated early in the main manuscript. It would also be useful to depict the catchments associated with each GNSS station in one of the map figures. There is some delineation of basins in the context of supraglacial lakes, but not a map that shows the catchment basins used in calculations for each GNSS station.

Lines 657-670: I see some new spatial analysis was added here. This radially symmetric approach relates to the end of the above comment. Ext Fig 14 on page 58 partially answers some of the above questions about length scale (the radius of influence), but it would still be of use to put a sentence or two in the Intro of the main text.

I still have questions about the treatment of groundwater in the paper, though the authors have done what seems to be an extensive analysis with multiple models. Thematically consistent with the above, it would be useful to hear something about physical processes and

what anticipated groundwater processes would contribute to GNSS signals, as well as a bit more detail on the possible limitations of the treatment of groundwater in the two models used to estimate the contributions to deformation.

Referee #4 (Remarks to the Author):

Given that I'm stepping in at this point following the completion of the initial round of reviews, I'll keep my comments brief.

The present paper introduces a novel method for investigating water storage in Greenland, which I believe could significantly contribute to enhancing our understanding of the notoriously hard to observe hydrology system. While there have been some studies of water transport in limited areas, large-scale models often lack constraints for key parameters, such as water storage. This is where I think this work can really advance our understanding by providing previously unavailable constraints for models of subglacial hydrology.

The authors have done a commendable job at explaining the various nuisance signals and they have presented their approach in a clear and concise way. The only part where I encountered some difficulty was in understanding the effect of ice discharge. Considering that the total change in mass of the Greenland ice sheet is roughly attributed equally to SMB and ice discharge, and acknowledging that ice velocities (and hence discharge) can vary significantly during a season, I initially found it hard to accept that ice discharge has only a negligible effect on the displacements. The discussion for the KAGA station left me skeptical until I realized that while the total change of mass due to discharge is comparable to that due to SMB, its seasonal variability is much smaller (I found an enlightening figure in Mankoff et al. 2021). Perhaps a brief comment highlighting this point at the beginning of the corresponding paragraph would help in clarifying this aspect.

My only real concern lies with the paragraph discussing the contribution of groundwater. I would argue that the groundwater storage is part of the BWS, and therefore not a nuisance signal. Every component of the hydrological system at the base of the ice receives input from meltwater and is thus part of the system being analyzed.

Additionally, I question whether the global groundwater models used are well-suited for Greenland. As highlighted in the introduction, the subglacial hydrological system is not well understood and different models adopt vastly different approaches (see e.g. de Fleurian et al. 2018). The present work can hopefully be used in the improvement of these models.

In the GLOBWB2 paper you cite (Sutanudjaja et al. 2018) it is mentioned that its computational grid does not cover Greenland (section 2.1, first paragraph) and in the WaterGAP paper Greenland is excluded from the analysis, if I see it correctly. Therefore, I am a bit confused about the data in figure 13.

Small stuff:

Figure 4: You might want to consider truncating the y-axis for a) and b), and/or put less labels on the axis to improve readability. Additionally, removing the boxes surrounding the legends could enhance visual appeal, though this is subjective. Furthermore, a different color scheme (also for extended data figure 3) could aid in distinguishing between lines more effectively.

L67. ...formation of a sub-glacial channelized drainage system...

I would remove “channelized”, because as you point out a sentence later, some regions don’t form a channelized system.

De Fleurian, B, Werder, M, et al. SHMIP The subglacial hydrology model intercomparison Project. *Journal of Glaciology*. 2018;64(248):897-916. doi:10.1017/jog.2018.78

Mankoff, K. D., Fettweis, X., Langen, P. L., Stendel, M., Kjeldsen, K. K., Karlsson, N. B., Noël, B., van den Broeke, M. R., Solgaard, A., Colgan, W., Box, J. E., Simonsen, S. B., King, M. D., Ahlstrøm, A. P., Andersen, S. B., and Fausto, R. S.: Greenland ice sheet mass balance from 1840 through next week, *Earth Syst. Sci. Data*, 13, 5001–5025, <https://doi.org/10.5194/essd-13-5001-2021>, 2021.

Referee #4 (Remarks on code availability):

I only glanced at the code, because I have little experience in matlab. It looks like it should in principle be possible to reproduce the results from the paper, but the figures in the output directory seem to be from a previous version and not the final ones that are shown in the current paper. I would appreciate an update.

Author Rebuttals to First Revision:

Referee #1 (Remarks to the Author):

I apologize for the delay in returning my review. Some significant real-world events got in the way of completing this.

The authors have addressed my specific comments from the first version well, and I think the same is true of the comments from the other reviewer. I read over all of the revisions and I am satisfied with the changes made by the authors. I look forward to publication of this very interesting paper.

Authors: Thank you very much for reviewing this paper and your constructive comments. Our response is shown below in light blue.

Minor corrections

Line 434. Add “the” before “Pan-Arctic”.

Authors: Corrected. Please see line 457 in the revised manuscript.

Line 435. Add “the” before “GrIS”

Authors: Corrected. Please see line 458 in the revised manuscript.

Referee #1 (Remarks on code availability):

I did only a very quick scan of the code -- I did not test it.

Authors: We have included more code to make sure readers can reproduce the results in this study.

Thank you again for your excellent suggestions which have resulted in an improved manuscript. We are very grateful.

Referee #3 (Remarks to the Author):

See also attached PDF

This review focuses primarily on the authors' responses to Reviewer #2.

I have made comments directly in the response-to-reviewer file (in red) to address the authors' responses. Below I have made a few comments of my own while I read the revised manuscript, prior to reading the reviews.

Authors: Thank you very much for your time to review this paper and your excellent comments. We are very grateful for your valuable comments and have addressed them one-by-one. Our responses are shown below in light blue.

In addition, we have incorporated our responses to the additional comments of reviewer 3 to the response to reviewer 2 from the last round in the end of this rebuttal.

I was surprised that in the opening comments, a (subglacial) groundwater system is neglected as a possible pathway for water export that originates as surface melt. Supraglacial, englacial, subglacial (specified to be ice-bed interface) pathways are mentioned, but even in a bedrock substrate there must be a non-zero component of subglacial groundwater recharge and transport where the ice-bed interface is temperate. Groundwater seems important to acknowledge in a study focused on water.

Authors: Thank you very much for the comment. We agree with the reviewer that groundwater is a possible pathway. In line with the comments from you and reviewer 4, we acknowledge groundwater is a part of the buffered water storage signal observed in this study and include the related words in the main text (Please see Lines 501-516).

Line 66 onward: I personally find the use of “hydrostatic pressure” for water pressure confusing, as some might use “hydrostatic pressure” to refer to the ice overburden pressure, which is not what the authors mean. Suggest using instead “basal water pressure” or “basal effective pressure”.

Authors: Thank you very much for the comment. We agree with the reviewer. They are changed as basal water pressure in lines 67, 69 and 71.

Lines 69-70: “However, ice sheet regions with a channelised drainage system may alternate with regions lacking an efficient drainage, the latter retaining high hydrostatic pressure.” “Alternate” is not quite the right notion. The glacier bed and basal drainage system are highly heterogeneous in space, so even regions where channels are present could be dominated by high basal water pressure if the drainage system is not hydraulically well connected to the channels.

Authors: Thank you very much for the comment. The statement is modified as follows: “Importantly, such drainage systems as well as the glacier bed are highly heterogeneous, so that even regions where channels are present could be dominated by high basal water pressure if the drainage system is hydraulically poorly connected to the channels.” Please see Lines 69-72 in the revised manuscript.

Line 74: “An ice penetrating radar can detect basal water”. Sorta, but this vastly oversimplifies. IPR can detect frozen/thawed bed and various other properties from which something about water might be inferred. Adjust wording slightly to improve precision, e.g., even “can detect properties indicative of basal water” would make this more palatable.

Authors: Corrected. Thank you very much for the comment. Please see Lines 75-76 in the main text.

Lines 80-82: "...bedrock's elastic response to total mass variations in the vicinity of a given GNSS station": This is a critical summary of the method and, in my view, should make the mechanism that relates the process to the observable clear for a broad readership: is GNSS mounted on bedrock outside of the ice sheet registering crustal flexure due to variable transient amounts of water mass being stored in the vicinity (all around the GNSS station, including on/in/under the ice)? Reporting the length-scale of influence/sensitivity would also be used here: over what radial distance/area is a given station GNSS sensitive? By what mechanisms do the different sources of GNSS signal contribute? For example, how extensive must the water redistribution be to be measurable? Is it correct that only horizontal redistribution matters? Some related information is presented deep within the extended figures, but a brief synopsis would help readers here. I felt there was a mechanistic gap throughout the paper, as it was assumed the reader would immediately understand that GNSS-measured bedrock elevation changes could be equated to the mass of water leaving the ice sheet. It would be appreciated if the authors would connect those dots more clearly.

Authors: Thank you very much for the comment.

As for the question "is GNSS mounted on bedrock outside of the ice sheet registering crustal flexure due to variable transient amounts of water mass being stored in the vicinity (all around the GNSS station, including on/in/under the ice)?", the answer is yes. The GNET stations were installed on the bedrock around the GrIS periphery. It is sensitive to all the processes which cause mass changes on/in/under the ice close to the station. We have included a statement to reflect this in the main text (see new Lines 374-375).

As for the question "Reporting the length-scale of influence/sensitivity would also be used here: over what radial distance/area is a given station GNSS sensitive?", based on the sensitivity test in Lines 707-712, it reveals a significant contribution from the near

field: mass changes within 200 km from the station contribute about 80% to the total loading displacements. There is little sensitivity to mass changes beyond 500 km from the station.

As for the question “By what mechanisms do the different sources of GNSS signal contribute? For example, how extensive must the water redistribution be to be measurable? Is it correct that only horizontal redistribution matters?”, the GNSS loading is based on pure elastic theory. The measurability depends on the magnitude and radial distance of vertically-integrated fluid masses to the GNSS station. Thereby, only horizontal redistribution of mass matters.

As for the remark “Some related information is presented deep within the extended figures, but a brief synopsis would help readers here. I felt there was a mechanistic gap throughout the paper, as it was assumed the reader would immediately understand that GNSS-measured bedrock elevation changes could be equated to the mass of water leaving the ice sheet. It would be appreciated if the authors would connect those dots more clearly. ”, we have added the following brief synopsis to the manuscript: “Elastic deformation of the Earth's crust occurs when stress is applied by a mass, causing the crust to alter its shape. Once the stress is removed, the crust returns to its original form. For instance, changes in the distribution of ice, snow, water, or atmospheric mass result in regional elastic deformation of the crust. By continuously and precisely monitoring the position and movement of the Earth's surface, GNSS provides quantitative information on regional mass changes in glaciers, ice sheets, and water storage systems. In general, the accumulation of mass at the surface leads to subsidence of the crust, while the removal of mass causes crustal uplift. This elastic deformation happens instantaneously when mass is redistributed.” Please see Lines 82-90 in the revised manuscript.

Line 95 cf. 88: Somewhat confusing at this point in the manuscript that GNSS data appear to be used to test SMB models (question iii, line 88), but that model output is being used to correct the GNSS signal (line 95). Can a sentence of clarification be introduced to dispel concern that there is circularity in the process? The scaling is reported as a major result in the opening paragraph of the paper, so this seems important.

Authors: Thank you for this valuable comment. Because the scaling factors applied to runoff estimates from SMB models are estimated simultaneously with parameters describing the buffered water storage, circularity is not an issue. To make this more clear in the text, we reformulated the third study objective as follows: “(iii) is it possible to better explain the BWS signal in GNSS data by scaling the runoff from SMB models?” (lines 99-100).

Lines 97-102: It would help if stated explicitly that downward motion equates to accumulation of stored water and vice versa, presumably due to proximity of GNSS to locations of accumulation. I’m thinking about mass changes in general and how station position relative to the mass change might lead to downward (proximal for mass accumulation) vs upward (distal from mass accumulation) responses.

Authors: Thank you very much for the comment. In the revised version, we now state this explicitly in the lines 111-113 as “In general, downward motion corresponds to accumulation of stored water and vice versa.”

For elastic loading, the bulge (representing the transition from downward to upward motion) occurs when the mass is located 40 degrees to 100 degrees (~4,000 km to 10,000 km) radial distance from the GNSS site. In the case of GNET, all mass changes on the GrIS are far away from the bulge and the general directional principle holds. Moreover, the GNET stations are mostly sensitive to near-field mass changes, as indicated by the sensitivity test discussed in Lines 693-705.

Reference:

Farrell, W. E. (1972), Deformation of the Earth by surface loads, *Rev. Geophys.*, 10(3), 761–797, doi:10.1029/RG010i003p00761.

Line 124: “tuned” may be better than “calibrated” since this is an empirical scaling

Authors: Thank you very much for the comment. Based on the in situ GNSS data, we evaluate the runoff modelled by RACMO 2.3p2. We contend that the calculated factors reveal an actual deficiency of the modelled runoff. Applying these factors make the modelled runoff more realistic (see Lines 152-169 in the main text). Therefore, in this case we prefer to use the term “calibrated”. If the reviewer believes that “tuned” is a better word, we are fully open and happy to take this comment.

Line 128-129: “total BWS”. Doesn’t the mass need to move substantially from its place of origin to cause crustal elevation change (i.e., meltwater that accumulates locally in a supraglacial lake would not change the mass distribution substantially enough to be detected in the form of bedrock elevation changes outside the ice sheet)?

Authors: Thank you very much for the comment. If meltwater is stored locally (near the location where it was produced, e.g., meltwater that accumulates locally in a supraglacial lake), this, most probably, will not affect the GNSS signal. We designed a test and find that horizontal mass displacement must be in the order of tens of km to emerge in elastic loading data. For more details, we refer to our answer to additional comments of Reviewer 3 to our response to Reviewer 2 from the previous round (please see Fig. S2 in P24 in the last part of this rebuttal letter).

Line 547: “relatively” => “relative”

Authors: Corrected. Thank you very much for the comment. Please see Line 579 in the main text.

Eqn (A9): somewhere around here, shouldn't there be a reference to a horizontal length scale? A station represents a point location that is sensitive to mass change within some region. Eqn (A8) expresses mass change at an individual (mass balance) grid cell which is assumed linearly related to elastic deformation in (A9), but then the spatial relationship between contributing (ice-sheet) grid cells and the (off-ice) station location disappears. In the equations that follow it appears that the BWS inferred at any station is being related to catchment-wide quantities. If this interpretation is correct---that each GNSS station is assumed sensitive only to mass changes that occur in its hydrological catchment (the furthest points of which may be further away than adjacent catchments)--- it would help readers like me if this were stated early in the main manuscript. It would also be useful to depict the catchments associated with each GNSS station in one of the map figures. There is some delineation of basins in the context of supraglacial lakes, but not a map that shows the catchment basins used in calculations for each GNSS station.

Authors: Thank you very much for the comment. We agree that it would be more fair to assume that GNSS station is sensitive to mass changes that occur in all surrounding hydrological catchment. We have adjusted the text as follows: "Let us consider the total BWS $S(t)$ in the drainage basins located around the current GNSS station (more specifically, in the drainage basins that substantially affect the elastic deformations at the current GNSS station)" (lines 607-609). For more information about the spatial sensitivity of GNSS loading data, we kindly refer to responses to the additional comments of Reviewer 3 to the response to Reviewer 2 from the last round at the end of this rebuttal.

Lines 657-670: I see some new spatial analysis was added here. This radially symmetric approach relates to the end of the above comment. Ext Fig 14 on page 58

partially answers some of the above questions about length scale (the radius of influence), but it would still be of use to put a sentence or two in the Intro of the main text.

Authors: We added the following remark (lines 92-94: “(we estimate the sensitivity radius of GNSS data as approximately 200 km, see *Spatial sensitivity of GNSS loading data*)”. We also included a more extended discussion about the GNSS loading mechanism in the introductory section see Lines 82-90.

I still have questions about the treatment of groundwater in the paper, though the authors have done what seems to be an extensive analysis with multiple models. Thematically consistent with the above, it would be useful to hear something about physical processes and what anticipated groundwater processes would contribute to GNSS signals, as well as a bit more detail on the possible limitations of the treatment of groundwater in the two models used to estimate the contributions to deformation.

Authors: Thank you very much for the comment. In this study, two types of groundwater can be identified:

1) Shallow groundwater in tundra areas, which results from snow melting and rainfalls there. In principle, this is an ordinary component of the Terrestrial Water Storage, which is described by various hydrological models, including those addressed in the manuscript (see Extended Data Fig. 13 modelled by PCR-GLOBWB and WGHM). Based on the personal communication with Prof. Mark Bierkens, who is a developer of PCR-GLOBWB, it is very difficult to model the groundwater accurately in those regions, but the model outcome is enough for a first-order estimate of its magnitude. In this study, we analyze the uncertainty of groundwater estimates in Sect. *Uncertainties of the mean annual cycle of residual vertical displacements* (see Lines 422-428 in the manuscript). We take KAGA as a representative station, and quantify the difference between mean annual vertical displacements per calendar month with and without groundwater signal subtracted from GNSS data using PCR-GLOBWB and WGHM models. This is basically the modelled groundwater signal

itself. It shows that the RMS signal from PCR-GLOBWB and WGHM model is 0.25 mm and 0.02 mm, respectively (see Fig. S1), which is small, as compared to the BWS signal (see Extended Data Fig. 5). We have included this information in Lines 422-428 of the revised manuscript.

2) Deep groundwater below the ice sheet (its presence was detected by a hydrological well down to the depth of hundreds of meters⁶²). It is a product of ice sheet melting (both at the surface and the ice sheet base). To the best of our knowledge, little is known about variations in the deep groundwater mass. Here, we consider the signal from deep groundwater as a part of the total BWS signal we detect in GNSS data. Unfortunately, elastic loading data do not allow the deep groundwater to be separated from the rest of the BWS.

Fig. S1 The mean annual vertical displacements per calendar month of modelled groundwater signal from PCR-GLOBWB (left) and WGHM (right) models, which are shown in monthly time-series in Extended Data Fig. 13.

References

- Müller Schmied, H., Cáceres, D., Eisner, S., Flörke, M., Herbert, C., Niemann, C., ... & Döll, P. The global water resources and use model WaterGAP v2.2d: Model description and evaluation. *Geoscientific Model Development* 14, 1037-1079 (2021).
- Sutanudjaja, E. H., Van Beek, R., Wanders, N., Wada, Y., Bosmans, J. H., Drost, N., ... & Bierkens, M. F. PCR-GLOBWB 2: a 5 arcmin global hydrological and water resources model. *Geoscientific Model Development* 11, 2429-2453 (2018).

Referee #4 (Remarks to the Author):

Given that I'm stepping in at this point following the completion of the initial round of reviews, I'll keep my comments brief.

The present paper introduces a novel method for investigating water storage in Greenland, which I believe could significantly contribute to enhancing our understanding of the notoriously hard to observe hydrology system. While there have been some studies of water transport in limited areas, large-scale models often lack constraints for key parameters, such as water storage. This is where I think this work can really advance our understanding by providing previously unavailable constraints for models of subglacial hydrology.

Authors: Thank you very much for reviewing this paper and your constructive comments. In the light of the insightful response from you, we have made the corresponding changes, which have greatly improved the original manuscript. A detailed point-by-point response to your comments addressing all of the identified issues is listed below. We appreciate your encouraging agreement on the novelty and importance of the results.

The authors have done a commendable job at explaining the various nuisance signals and they have presented their approach in a clear and concise way. The only part where I encountered some difficulty was in understanding the effect of ice discharge.

Considering that the total change in mass of the Greenland ice sheet is roughly attributed equally to SMB and ice discharge, and acknowledging that ice velocities (and hence discharge) can vary significantly during a season, I initially found it hard to accept that ice discharge has only a negligible effect on the displacements. The discussion for the KAGA station left me skeptical until I realized that while the total change of mass due to discharge is comparable to that due to SMB, its seasonal variability is much smaller (I found an enlightening figure in Mankoff et al. 2021).

Perhaps a brief comment highlighting this point at the beginning of the corresponding paragraph would help in clarifying this aspect.

Authors: Thank you very much for the comment. We have made this clear in Line 440-442 in the main text of the revised MS.

My only real concern lies with the paragraph discussing the contribution of groundwater. I would argue that the groundwater storage is part of the BWS, and therefore not a nuisance signal. Every component of the hydrological system at the base of the ice receives input from meltwater and is thus part of the system being analyzed.

Authors: Thank you very much for the constructive comment. We fully agree with the reviewer that the groundwater at the base of the ice sheet is a part of the BWS analyzed in this study. We distinguish it from “shallow” groundwater in tundra, which is considered as a nuisance signal. A similar comment is made by Reviewer 3, please see our response to this reviewer or the Lines 501-516 in the main text.

Additionally, I question whether the global groundwater models used are well-suited for Greenland. As highlighted in the introduction, the subglacial hydrological system is not well understood and different models adopt vastly different approaches (see e.g. de Fleurian et al. 2018). The present work can hopefully be used in the improvement of these models.

In the GLOBWB2 paper you cite (Sutanudjaja et al. 2018) it is mentioned that its computational grid does not cover Greenland (section 2.1, first paragraph) and in the WaterGAP paper Greenland is excluded from the analysis, if I see it correctly. Therefore, I am a bit confused about the data in figure 13.

Authors: Thank you very much for the comment. We agree with the reviewer that the understanding of subglacial hydrological processes within the Greenland ice sheet is limited. Our study may indeed contribute to the investigation of those processes. As for the two global models, i.e., PCR-GLOBWB and WaterGAP, we find that our versions of the two models include the tundra areas of Greenland. Based on our personal communication with Prof. Mark Bierkens, who is a developer of PCR-GLOBWB, it is very difficult to model the groundwater accurately in those regions, but the model outcome is enough for a first-order estimate of its magnitude. Therefore, in this study, we utilized their output of groundwater to make the sensitivity analysis (see Lines 422-428 in the main text), to provide a first-order estimate of the amplitude of the groundwater-related signal from tundra areas.

Small stuff:

Figure 4: You might want to consider truncating the y-axis for a) and b), and/or put less labels on the axis to improve readability. Additionally, removing the boxes surrounding the legends could enhance visual appeal, though this is subjective. Furthermore, a different color scheme (also for extended data figure 3) could aid in distinguishing between lines more effectively.

Authors: Thank you very much for the comment. We have truncated the y-axis of Figure 4 to improve the readability. The boxes surrounding the legends are removed. The color scheme is changed in Fig. 4 and Extended Data Fig. 3.

L67. ...formation of a sub-glacial channelized drainage system...

I would remove “channelized”, because as you point out a sentence later, some regions don’t form a channelized system.

Authors: Corrected. Thank you very much for the comment.

De Fleurian, B, Werder, M, et al. SHMIP The subglacial hydrology model intercomparison Project. *Journal of Glaciology*. 2018;64(248):897-916.
doi:10.1017/jog.2018.78

Mankoff, K. D., Fettweis, X., Langen, P. L., Stendel, M., Kjeldsen, K. K., Karlsson, N. B., Noël, B., van den Broeke, M. R., Solgaard, A., Colgan, W., Box, J. E., Simonsen, S. B., King, M. D., Ahlstrøm, A. P., Andersen, S. B., and Fausto, R. S.: Greenland ice sheet mass balance from 1840 through next week, *Earth Syst. Sci. Data*, 13, 5001–5025, <https://doi.org/10.5194/essd-13-5001-2021>, 2021.

Referee #4 (Remarks on code availability):

I only glanced at the code, because I have little experience in matlab. It looks like it should in principle be possible to reproduce the results from the paper, but the figures in the output directory seem to be from a previous version and not the final ones that are shown in the current paper. I would appreciate an update.

Authors: Thank you very much for the comment. Some of the figures in the main text are made using other softwares (i.e., Origin and ArcGIS), and not Matlab. Here, the matlab code is utilized for producing the data. In addition, we have updated the codes to process GRACE data via <https://github.com/JiangjunRAN/ANGELS-Mascon>.

Below is our response (shown in light blue) to the comments of Reviewer #3 (shown in red), which addressed our response (shown in blue) to the concerns of Reviewer #2 (shown in black).

Reviewer Comments & Author Rebuttals

Referee #2 (Remarks to the Author):

Note: Additional referee evaluation of author responses shown in red. The authors made the changes clear and easy to follow. Nicely done!

Authors: Thank you again for your excellent suggestions which have significantly improved the manuscript. We are very grateful.

The subject of this paper, seasonal storage and release of liquid water in the Greenland ice sheet, is a key component of many unsolved science problems related to the ice sheet's mass balance and ice flow dynamics. Because meltwater cannot exit the ice sheet instantaneously, the time/space evolution of hydrologic systems revolves around water storage. Storage is, therefore, a key state variable of all supraglacial, englacial, and subglacial hydrologic processes and models thereof. Water storage at the bed is a critical aspect of sliding mechanics and geochemical processes. Yet, the processes are too complex, and our understanding is too limited to explicitly include the many hydrologic processes (e.g., the seasonal evolution of a subglacial drainage system) in any large-scale SMB/runoff model.

Different observational methods, such as radar, GPS signals, GRACE, remote sensing imagery, radar and radiometry, have been used to determine water storage in various compartments. Each method presents resolution and interpretation challenges, and several offer only snapshots rather than continuous time series. Breakthroughs with direct measurements of storage thus have the potential to make a strong impact on many different aspects of Greenland hydrology.

Authors: Thank you very much for your time to review this paper and your excellent comments. We agree with your emphasis on the importance of studies on seasonal

storage and release of liquid water in the Greenland ice sheet. We are very grateful for your valuable comments and have addressed them one-by-one as follows.

Concerns:

1. Hydrologic interpretation.

Throughout the paper, the description of the water storage mechanism is quite puzzling. Examples include:

-- Figure 1 depicts englacial lakes (and subglacial lakes that are actually shown as englacial) — I know of no evidence for lakes existing inside the ice.

Authors: Thank you very much for the comment. Sorry for the misunderstanding. Figure 1 is a *schematic* figure to show the processes related to the liquid water evolution beneath the ice sheet surface and the associated changes of vertical bedrock. The distance of englacial lakes from the ice sheet surface is schematic, just indicating that some water is buffered beneath the ice sheet surface, and does not mean the actual location of englacial lakes is inside the ice sheet. However, based on the above comments, we have improved Fig. 1, to avoid any misunderstanding. For your convenience, the revised Fig. 1 is shown below. Please see lines 218-232 in the revised manuscript.

The representation of relevant hydrological features in the new Figure 1 seems fine to me, with the exception that differences in the relationship between the white dashed line and the bedrock mound on which the GNSS station is mounted are hard to discern in the 3 panels.

Authors: Thank you very much for your agreement on the revised Fig. 1. In the first panel, the dashed line is the same as the boundary of ice-bedrock interface, because the vertical deformation does not start. However, the deformation is non-zero in panels b and c. In line with the reviewer's comment, the difference between the white dashed line and ice-bedrock interface is increased ~1.5 times (see Fig. 1), as compared to the previous version of the plot.

Fig. S1 The revised Fig. 1.

-- The text concerning loading is confusing because a phase change is not a mass change, so water mass loading must be due to the redistribution of water from one place to another (not stated).

Authors: Thanks for this thoughtful comment. The “phase change” refers to melting and refreezing processes. In order to avoid potential confusion, we have added the following sentence in the introductory section (Lines 84-85):

“Note that bedrock displacement time-series describe the mass of buffered water rather than the total water mass, since surface loading does not change if water re-freezes in-place.”

This clarification is helpful, but I also struggled with the lack of clarity around water mass change: the mass needs to move in order to elicit elastic deformation. Throughout the text, it almost seemed like the GNSS signals were being related to water mass, rather than a change in water mass (see also my independent comments).

Authors: Thank you very much for the comment. The reviewer is correct that the GNSS measures the elastic deformation caused by the change in water mass. This is fully consistent with the manuscript. We clean GNSS data from nuisance signals to obtain residual deformations that are referred to *the Buffered Water Storage (BWS)*. The latter is defined as “the water mass temporarily buffered within the GrIS, being mostly released into the ocean before the onset of next melt season” (lines 64-65). To make the relationship between water mass and vertical deformations even more clear for readers, we have extended our clarification cited above (in blue) as follows: “Note that resulting bedrock displacements peak during the melt season and gradually reduce thereafter. This is because they describe BWS rather than the total water mass (surface loading does not change if water re-freezes in-place)” (lines 95-98).

-- The manuscript intro/discussion is structured around water storage in the firn compartment, but why this is driving the loading signal versus other compartments is not clear. Melt across most of the firn area will result in a short-distance mass redistribution, vertically within the upper 10 m. Only a relatively narrow reach of the far lower accumulation area has substantial horizontal flow and accumulation of water,

but is there evidence that this is enough to drive the loading signal? Regardless, all runoff is accumulated toward the margin. Whereas a SMB model may simulate a large amount of meltwater generation across the accumulation zone, the subglacial drainage system, not the firn compartment, has the most accumulating water mass and is therefore the most likely candidate for loading. Water at the bed cannot be dismissed by invoking fast conduit drainage. A large body of literature on subglacial hydrology debates issues such as a) how far inward on the ice sheet fast-draining conduits can actually develop (e.g., Werder et al., 2013); and, b) how much area of the bed sits between conduits and does not drain quickly (e.g., Hoffman et al., 2016).

Authors: We agree that the introductory section was too much focused on processes in the firn layer, which resulted in some mismatch between that section and the rest of the manuscript. To solve this issue, we have re-organized the introductory section to put more emphasis on water at the bed (Lines 65-76). The firn layer is mentioned in the Discussion Section only briefly, and primarily in the context of our interpretation of the scaling factors to be applied to runoff estimates provided by the SMB model (Lines 189-199).

Firn processes did not seem over-emphasized in the revised manuscript. This point, however, also relates to the lengthscales over which mass must move to create detectable deformation.

Authors: We are pleased to see that we reached a consensus concerning the text about the firn processes. For the issue of the length scale over which mass must move to create detectable deformation, we designed a sensitivity analysis which is similar to Fig.1 in Wahr et al., (2013) and Fig. 6 in Bevis et al., (2016). By considering a mass formed by a 1-m water layer over a disk of 20 km radius, we calculated the deformation at different distances from the disk centre (Fig. S2). This plot clearly shows that horizontal displacement of a mass must be of the order of at least tens of km to be seen in elastic loading data. Thus, a local re-distribution of meltwater within the firn layer can be safely ignored. We have included the clarification in Lines 707-712.

Fig. S2 Deformation caused by a load of 1-m water disk of 20 km radius at different distances from the disk centre. This result is consistent with Figure 1 by Wahr et al. (2013).

Reference

Wahr, J., S. A. Khan, T. van Dam, L. Liu, J. H. van Angelen, M. R. van den Broeke, and C. M. Meertens (2013), The use of GPS horizontals for loading studies, with applications to northern California and southeast Greenland, *J. Geophys. Res. Solid Earth*, 118, doi:10.1002/jgrb.50104.

As for “A large body of literature on subglacial hydrology debates issues such as a) how far inward on the ice sheet fast-draining conduits can actually develop (e.g., Werder et al., 2013); and, b) how much area of the bed sits between conduits and does not drain quickly (e.g., Hoffman et al., 2016).”, the loading observed using GNSS verticals is the bedrock response to integrated mass changes. We have removed SMB and other non-ice sources, attributing the remaining changes to temporally buffered meltwater. But it is difficult to further differentiate the pathways or storages of the buffered meltwater by using GNSS data alone. We hope this study will stimulate further research to address ice sheet hydrology from a new perspective like this.

I don't think the reviewer is asking the authors to attempt to extract more detail from the GNSS data related to the style of subglacial drainage or the architecture of the system. I

think the point here was to say that water in the subglacial drainage system can have a long residence time even though channels may form, i.e., “Water at the bed cannot be dismissed by invoking fast conduit drainage”. I’m not sure any further action is needed, but readers would like to know that the authors have thoroughly considered the plausibility of the mechanisms proposed to explain the GNSS data.

Authors: Thank you very much for the excellent comment. We agree that “water in the subglacial drainage system can have a long residence time even though channels may form”. This is acknowledged, among others, in the introductory section. It contains now the following text, which also considers the independent recommendations of Reviewer 3: “Ultimately, the accumulation of water results in a formation of a subglacial drainage system, which facilitates a water release into the ocean, so that the basal water pressure is reduced. Importantly, such drainage systems as well as the glacier bed are highly heterogeneous, so that even regions where channels are present could be dominated by high basal water pressure if the drainage system is hydraulically poorly connected to the channels” (lines 67-72).

2. Signal robustness.

Because the water storage signal is a small residual that emerges after many large corrections to the original data, the robustness of the signal is dependent on the confidence of the removal of so-called nuisance signals. Perhaps some of the following issues are partly or solely responsible for the spatial variability of the results.

Authors: We agree that this signal is relatively small, but it is clearly visible (see, e.g., panel (h) in Extended Data Fig. 4 in the updated manuscript, please see Lines 851-853) and reliably estimated in this study. To ensure the robustness of our estimates, we conducted a thoughtful analysis by evaluating all corrections with the State-of-the-Art models or data in the community. Please see revised ED Fig. 5 (Lines 854-863) and the following text. It is worth to note that we also compared the GNSS-based water storage signal with that based on independent satellite gravimetry data (Fig. 3, see

Lines 245-251). This comparison confirmed that this signal is real and not a data artefact.

I'm going to assume this response will be evaluated by someone with GNSS signal processing expertise.

Authors: Thank you very much for the comment. We believe Reviewer 1 is a GNSS expert.

-- Prior work by Lui et al. (2017) demonstrated the difficulty of SMB corrections in Greenland due to far-field SMB influences. In other words, where does the SMB signal come from? Lui et al. (2017) argue that SMB from glaciers 1200 km away can impact Greenland GNSS signals. In fact, they found that correcting for SMB adversely inflated the residual. The manuscript does not make clear how this problem was handled nor is the Lui et al. (2017) paper cited.

Authors: Thank you very much for the comment. Liu et al. (2017) was the previous study by one of the co-authors. Liu et al. (2017) studied Upernavik glacier using two GNSS stations located in the nearby bedrock area. We agree with Liu et al. (2017) that far-field SMB signals must be taken into account when interpreting the annual amplitude of the GNSS verticals. In this study we use an ice-sheet-wide SMB model (RACMO) rather than a local or regional model and therefore correct for the far-field signal. We now state in line 376 that the SMB model covers the entire GrIS (and not just a small area).

In addition, we made a sensitivity study (Lines 657-670) to clarify how mass changes over the entire ice sheet, including glaciers 1,200 km away, may affect the observed mass loading signal. By taking the detrended SMB in July 2012 (Fig. A3 in the rebuttal) as the ice sheet mass change signal, we analyze the sensitivity of GNSS loading observed at KAGA as an example. The processing strategy is to take the KAGA station as the center, create ring-shaped zones outward with a step width of 50 km, calculate the vertical displacement caused by SMB inside each ring, and then normalized with the entire Greenland. The results show that cumulative contribution to the detrended SMB loading reaches approximately 80% when the distance from

KAGA station is 200 km. The sensitivity curve (Fig. A3) reveals a significant contribution from near field: SMB changes within 200 km from KAGA contribute by about 80% to the detrend SMB loading displacements. There is little sensitivity to SMB beyond 500 km from KAGA.

Fig. A3 (a) Detrended SMB mass anomalies in July 2012. (b) Cumulative contribution to the detrended uplift at KAGA due to SMB loading, depending on the radius of the buffer zone around that station. All the numbers are normalized with the loading uplift caused by the SMB signal over entire Greenland. The X axis shows a step of 50 km in the distance range of 0-1,500 km. This figure is included as ED Fig. 14 in the revised manuscript.

I don't have much expertise in this area, but the response and added analysis seem reasonable. Length scales of influence were one of my big questions, so I would like to see some related information in the intro to the main text.

Authors: Thank you very much for the comment. We have included a sensitivity analysis above, i.e., Fig. S2. We have also added the following remark to the introductory section of the main text (lines 92-94): "(we estimate the sensitivity radius of GNSS data as approximately 200 km, see Spatial sensitivity of GNSS loading data)".

We designed a sensitivity test to clarify the length scale over which mass must move to create detectable deformation. It is found that horizontal displacement of a mass must be of the order of at least tens of km to be seen in elastic loading data (see Fig. S2 above). We have included the text in Lines 707-712 in the main text.

-- The analysis does not appear to consider the seasonal loading signal caused by ice flow across the ablation zone (c.f., unloading by calving). The speed of the ice sheet in the ablation zone increases during summer, typically by a factor of 2-3 (see Zwally et al. (2002) for the seminal paper, with much confirming work published since then). Speed typically peaks mid-summer, similar to the GNSS loading signal. Ice acceleration causes horizontal advection of ice thickness gradients — more ice is moved low that must be slowly beaten back by ablation. How much of the residual could be due to this effect?

Authors: We agree with the reviewer that the seasonal loading signal might be caused by the ice flows. In this study, we have considered this effect. Please see the effect of seasonal variations in ice speed analyzed in the Section “Contribution of ice discharge”, which is a part of Methods (see lines 416-429). To obtain an upper bound of the contribution of seasonal ice discharge variations to the observed vertical displacements, we use Jakobshavn Isbræ (JI) as an example because it shows the largest ice flow velocity seasonality amongst GrIS outlet glaciers. By investigating the GNSS loading signal observed at the KAGA GPS station, which is located ~5 km from the JI terminus, we conclude that this effect (i.e., $< \sim 1\text{mm}$) is not significant.

Seems reasonable.

Authors: Thank you very much for your positive reaction.

-- The product used for LWS corrections is a global product and is not necessarily well-suited for Arctic regions dominated by permafrost hydrology. Some areas have many lakes fluctuating lake levels. Further, the product has no treatment of

groundwater, whereas Liljedahl et al., (2021) show very large head swings in the groundwater system along the ice margin. LWS is a big correction and uncertainties will vary substantially around the ice sheet.

I share the concerns about groundwater.

Authors: Thank you very much for the comment. We have clarified this issue in response to your independent comments in the rebuttal and modified the paper (see Lines 501-516), by taking the comments from Reviewers 3 and 4.

Authors: Thanks for this comment. We agree that lakes and groundwater play important roles in Arctic regions dominated by permafrost hydrology. Indeed, the GLDAS model we considered in the previous variant of the manuscript lacks both water storage components. To solve this problem, we also add two hydrological models (see Extended Data Fig. 5c in Line 854), i.e., WaterGAP2 Global Hydrology Model (WGHM) and PCRaster Global Water Balance (PCR-GLOBWB), both of which take groundwater into account. Furthermore, we explicitly consider water mass variations of lakes in Arctic regions (Lines 431-475).

1) Three State-of-The-Art models are utilized in this study in total: GLDAS, PCR-GLOBWB and WGHM. Even though these models are not specific for Arctic regions, we believe they are accurate enough at least for the first-order magnitude as evidenced by?. Note that WGHM is tailored in Arctic region by considering the existence of permafrost. A comparison with other models showed that WGHM agrees best with independent in-situ observations collected in Arctic region (Gädeke et al., 2020). In addition, as we have shown with our sensitivity analysis of GNSS-based loading signal presented above (Fig. A3), this contribution is mostly (~80%) limited to the radius of ~200 km. Most parts of the Arctic regions are quite far (> 1000 km) from the GNSS stations in Greenland, and thereby the mass variations over there are little impact the GNSS based water storage signals observed in this study.

The highlighted text above is a good argument for the lakes, but not for groundwater around Greenland.

Authors: Thank you very much for the comment. We agree that some mass re-distribution of hydrological origin occurs also at smaller distances from Greenland than 1000 km. However, the groundwater output from the state-of-the-art models, i.e., PCR-GLOBWB and WGHM, covers the all the tundra area around Greenland. From the personal communication with Prof. Mark Bierkens, a developer of PCR-GLOBWB, little is known to model the groundwater accurately in those regions, but may be fair to give a first-order estimate of its magnitude. We have clarified this issue and please kindly refer to those in P10-11 above in the rebuttal.

Taking the KAGA as an example: the vertical displacement at KAGA in response to global LWS loading derived from the 3 hydrological models, excluding Greenland, during the study period (2009-2015) ranges from -1.0 to ~1.5 mm (Fig. A4). This magnitude is sufficiently small to allow for neglecting the potential influence of LWS on the Greenland ice sheet meltwater storage.

What exactly do these models estimate as loading: water mass on the land surface? Is it possible that any of the elastic deformation could be attributable to seasonal recharge/discharge of the aquifer? Do the models include estimates of aquifer compressibility (assuming fractured rock aquifers) such that the vertical changes associated with possible compaction/expansion due to seasonal changes in hydraulic head?

Authors: Thank you very much for the comment. First of all, as for “What exactly do these models estimate as loading: water mass on the land surface?”, PCR-GLOBWB and WGHM models estimate the Total Water Storage (TWS) variations of the hydrological components, e.g., including canopy water storage, groundwater storage, lake/reservoir storage, snow water storage, soil water storage. Note that GLDAS model lacks the groundwater and lake components (unlike PCR-GLOBWB and WGHM).

As for “Is it possible that any of the elastic deformation could be attributable to seasonal recharge/discharge of the aquifer? Do the models include estimates of aquifer compressibility (assuming fractured rock aquifers) such that the vertical changes associated with possible compaction/expansion due to seasonal changes in hydraulic head?”, we could not find any information about the possible presence of underground aquifers below GNET stations. Most probably, nobody has investigated that issue yet. It is known, however, that poroelastic deformations observed at the Earth’s surface due to a compressibility of underlying aquifers show a totally different behavior, as compared to elastic deformations, which are our primary focus. For instance, recharge of an underlying compressible aquifer results in an uplift, not in a subsidence of the Earth’s surface (Larochelle et al, 2022). We have carefully checked the GNSS time-series for all stations in Greenland and did not find evidences of such a signal. From this, we conclude that recharge/discharge of compressible aquifers below GNET stations likely provides a negligible contribution to the observed residual displacements, if at all.

I have trouble agreeing with the final highlighted statement if I am interpreting the data correctly: Fig A4 below shows variations due to LWS of +/-1 mm that were neglected in the corrections? +/-1mm does not seem small compared to the signals in Figure 2, many of which are substantially less than +/- 5mm. They also appear to be out of phase so would make the signals in Fig 3 less pronounced if used in the correction.

Authors: Thank you very much for the comment. There is some misunderstanding here. Fig. A4 shows the *total* Land Water Storage (LWS) signal caused by hydrological processes world-wide excluding Greenland. Since our target is to analyze the Buffered Water Storage (BWS) in GrIS, thereby, the LWS signal shown in Fig. A4 is considered as a *nuisance* signal. Therefore, even though the LWS signal shown Fig. A4 may reach +/-1 mm in total, it has to be subtracted. What we care about are uncertainties of the estimated LWS signal. In Sect. *Uncertainties of the mean annual cycle of residual vertical displacements*, we take KAGA as a representative station to investigate, among others, those uncertainties in the context of the mean annual cycles

of residual vertical displacements (Fig. 2). We find that those uncertainties are small, i.e., around 0.21 mm (see Extended Data Fig. 5).

Without knowing enough to have confidence in the models' treatment of groundwater around Greenland, I am hesitant to dismiss this correction as negligible. Depending on what the models are doing, one solution may be not to attempt to correct for groundwater or claim it is negligible, but rather include it in what is being measured. If most of the groundwater recharge is coming from the ice sheet, then this is a source of water that would ideally be accounted for anyway.

Authors: Thank you very much for the comment. We fully agree with the reviewer that the groundwater coming from the melt of ice sheet should be a part of the BWS signal considered in this study. Similar suggestion is also made by Reviewer 4. We have changed the main text (see Lines 501-516) to clarify this aspect. Importantly, the discussion above is limited to groundwater variability estimated by hydrological models: we refer there to shallow groundwater originated from precipitation outside the ice sheet.

Fig. A4 Time series of the vertical displacements at KAGA in response to LWS loading excluding Greenland.

References:

GLDAS: Li, B., M. Rodell, S. Kumar, H. Beaudoin, A. Getirana, B. F. Zaitchik, et al. (2019) Global GRACE data assimilation for groundwater and drought monitoring: Advances and challenges. *Water Resources Research*, 55, 7564-7586. doi:10.1029/2018wr024618

PCR-GLOWB: Sutanudjaja, E. H., Van Beek, R., Wanders, N., Wada, Y., Bosmans, J. H., Drost, N., ... & Bierkens, M. F. (2018). PCR-GLOBWB 2: a 5 arcmin global hydrological and water resources model. *Geoscientific Model Development*, 11(6), 2429-2453.

WGHM: Müller Schmied, H., Cáceres, D., Eisner, S., Flörke, M., Herbert, C., Niemann, C., ... & Döll, P. (2021). The global water resources and use model WaterGAP v2. 2d: Model description and evaluation. *Geoscientific Model Development*, 14(2), 1037-1079.

Gädeke, A., Krysanova, V., Aryal, A. et al. (2020). Performance evaluation of global hydrological models in six large Pan-Arctic watersheds. *Climatic Change* 163, 1329–1351, <https://doi.org/10.1007/s10584-020-02892-2>

Larochelle, S., Chanard, K., Fleitout, L., Fortin, J., Gualandi, A., Longuevergne, L., et al. (2022). Understanding the geodetic signature of large aquifer systems: Example of the Ozark Plateaus in central United States. *Journal of Geophysical Research: Solid Earth*, 127, e2021JB023097. <https://doi.org/10.1029/2021JB023097>

2) As for “Some areas have many lakes fluctuating lake levels.”, indeed, there are many lakes in the Arctic region. To further quantify the potential water mass impact of lakes onto GNSS loading signal in Greenland, we consider three types of lakes (Lines 431-475): lakes in Pan-Arctic region (north of 60 °N) in general; supraglacial lakes (SGL) on GrIS; and proglacial lakes in the coastal part of Greenland.

Fig. A5 Distribution of lakes in Pan-Arctic region. This figure is included as ED Fig. 7 in the revised manuscript.

Lakes in Pan-Arctic region: Specifically, the monthly lake water storage data of some large lakes were directly downloaded from a recently published dataset by Yao et al., (2023). In addition, we accessed the intra-annual lake level datasets from three portals, i.e., Hydroweb (<https://hydroweb.theia-land.fr/>), Global Reservoirs and Lakes Monitor (G-REALM, https://ipad.fas.usda.gov/cropexplorer/global_reservoir/), and Database for Hydrological Time Series of Inland Waters (DAHITI, <https://dahiti.dgfi.tum.de/en/>), respectively (Cretaux et al., 2011; Birkett et al., 2009; Schwatke et al., 2015). For months without lake area data in the dataset provided by Yao et al., (2023), we follow the empirical relationship between lake water level and lake surface area from a similar previous study (Song et al., 2013), by interpolating the

missing lake area data with ampler level data and converting the lake area and level data to water storage changes.

It is worth mentioning that the recorded lakes (38 in total, indicated by blue circles in Fig. A5) are typically large, with 33 of them having a surface area greater than 500 km², accounting for 61.6% of the total surface area of Arctic lakes extracted by Yao et al. (2023). Note that the reference surface area here is derived from the HydroLAKES polygon (Messenger et al., 2016). For the remaining smaller inland lakes (red circles in Fig. A5), the area data are very limited. Therefore, we employ a scale-up strategy to estimate their impact on mass loading (Song et al., 2013).

Taking KAGA station as an example, the vertical displacement induced by the load of Arctic lake water storage during the study period (2009-2015) ranges from -0.1 to 0.04 mm (Fig. A6). This magnitude is sufficiently small to allow for neglecting the potential influence of these lakes in the Arctic region.

Fig. A6 Vertical displacements of lake loading at KAGA station. This figure is included as ED Fig. 8 in the revised manuscript.

I had trouble grasping how lake-level variations in distal locations like Russia would influence the GNSS signals around Greenland. It would seem necessary to know where mass lost/gained from the lakes goes in order to correctly account for this signal. It does not seem sufficiently important to pursue further.

Authors: Thank you very much for the comment. As it is written above, PCR-GLOBWB and WGHM models estimate the Total Water Storage (TWS) variations of almost all hydrological components, e.g., canopy water storage, groundwater storage, lake/reservoir storage, snow water storage, soil water storage (Müller Schmied et al., 2021; Sutanudjaja et al., 2018). Note that GLDAS model lacks the lake components (unlike PCR-GLOBWB and WGHM). We considered the potential impact of lakes in Arctic region separately since the GLDAS model is utilized from the very beginning of the study.

SGL on the GrIS: In addition, we assessed the vertical displacement caused by water mass loading from SGL on the GrIS. Firstly, we divided the GrIS into 10-by-10 km equal-area cells and identified all the cells with SGL using ~ 300,000 high spatiotemporal resolution optical images from Sentinel-2 and Landsat 8/9 over 2017-2022, as shown in Fig. A7. Then, the monthly SGL area changes during the melting season (i.e., May-September) were derived. The SGL area changes data are distributed via <https://doi.org/10.5281/zenodo.10398558>. Note that it is problematic to obtain the monthly SGL area change for entire GrIS before the launch of Sentinel-2B satellites in 2017 due to a high cloud contamination. We assumed, however, that the SGL area changes over 2009-2015 interval was similar to those over 2017-2022. When converting the SGL area changes to mass changes, we assumed that the maximum depth of GrIS supraglacial lakes is around 8.5 metres (Fair et al., 2020; Xiao et al., 2023), which allowed us to estimate the upper limit of water mass changes Fig. A8. The result shows that even for the maximal depth of 8.5 m, the magnitude of loading signal caused by SGL mass changes is ~0.3 mm, which is negligible, compared with the BWS signal.

I had trouble grasping the assumptions in this analysis. If a supraglacial lake drains vertically downward, and the water is retained locally at the bed, there would presumably

be no deformation measured at the GNSS station, right? Are the authors assuming that loss of lake mass instantaneously contributes to englacial/subglacial water mass somewhere significantly downstream? This would not be unreasonable, but I just don't know. Similarly, what is assumed when lakes increase in size? My assumption would be no deformation since the meltwater is likely to be (relatively) locally sourced and does not, therefore, represent a substantial lateral redistribution of mass. Here again, relating the processes envisioned to the measured signals would help a broad audience understand the assumptions and better appreciate this work.

Authors: Thank you very much for the comment. We believe there is some misunderstanding here. The purpose of the analysis of SGL here was to demonstrate that SGL provide a minor contribution to the total buffered water storage signal observed, in order to respond the concern of reviewer 2.

As for “If a supraglacial lake drains vertically downward, and the water is retained locally at the bed, there would presumably be no deformation measured at the GNSS station, right?”, yes, if a supraglacial lake drains vertically downward, and the water is retained locally at the bed, being re-frozen there, it is not a part of BWS and is not sensed by our analysis.

As for “Are the authors assuming that loss of lake mass instantaneously contributes to englacial/subglacial water mass somewhere significantly downstream? This would not be unreasonable, but I just don't know. ”, our analysis is limited to the BWS (that is, the water mass that ultimately discharges into the ocean). A horizontal displacement of water mass with a subsequent re-freezing is beyond the scope of our analysis. This is in line with the fact that only very significant horizontal displacements of mass (tens of km and more) can affect elastic loading data (see Fig. S2 above and the associated discussion there).

As for “Similarly, what is assumed when lakes increase in size? My assumption would be no deformation since the meltwater is likely to be (relatively) locally sourced and does not, therefore, represent a substantial lateral redistribution of mass.”, our input is *residual* elastic loading data, which are obtained by subtracting all the nuisance signals, including that from the Surface Mass Balance (SMB). However, models of SMB do not take into account the fact that it takes meltwater some time to leave the ice sheet: they assume that the mass associated with meltwater runoff disappears at the moment of meltwater production. Thus, an increase in a lake size, may, in principle, be visible in residual loading data as a temporal increase in mass (i.e., as a subsidence signal). Of course, in practice this signal is likely not visible: our analysis has shown that it is too small.

Proglacial lakes in Greenland: Similar to the SGL, we also consider the impact of proglacial lakes in Greenland using Hydrolake database (Fig. A9). In total, 2 687 proglacial lakes are taken into account. Most of the proglacial lakes are smaller than 5 km², while the largest one could reach ~100 km². The loading signal caused by proglacial lakes is small. At KAGA station, for instance, it is of the order of only 0.02 mm (Fig. A10).

This signal size seems suitably small. I think I can imagine the assumptions being made to relate the change in lake size with station elevation, but it would not hurt to state them explicitly.

Authors: Thank you very much for your agreement. Because of the very limited monthly lake area and lake level data availability of proglacial lakes in Greenland, we assume similar seasonality of lake areas as that of SGL. After that, we apply method by Song et al. (2013) to estimate lake volumes/masses.

Fig. A7 Spatial distribution of cells with SGLs. The grey cells indicate that there were no SGL over 2017-2022 from optical images. This figure is included as ED Fig. 9 in the revised manuscript.

Fig.

A8 The vertical displacements at KAGA station caused by SGL mass changes. This figure is included as ED Fig. 10 in the revised manuscript.

Fig. A9 Spatial distribution of proglacial lakes in Greenland. This figure is included as ED Fig. 11 in the revised manuscript.

Fig. A10 | The vertical displacements at KAGA station caused by proglacial lake mass changes over entire Greenland. This figure is included as ED Fig. 12 in the revised manuscript.

References:

- Cretaux J-F., Arsen A., Calmant S., et al., 2011. SOLS: A lake database to monitor in the Near Real Time water level and storage variations from remote sensing data, *Advances in space Research*, 47, 1497-1507.
- Birkett, C.M., Reynolds, C., Beckley, B., et al., 2009. From Research to Operations: The USDA Global Reservoir and Lake Monitor, chapter 2 in 'Coastal Altimetry', Springer Publications, eds. S. Vignudelli, A.G. Kostianoy, P. Cipollini and J. Benveniste, ISBN 978-3-642-12795-3, 2010.
- Schwatke C, Dettmering D, Bosch W, et al., 2015. DAHITI—an innovative approach for estimating water level time series over inland waters using multi-mission satellite altimetry, *Hydrology and Earth System Sciences*, 19(10), 4345-4364.
- Yao F, Livneh B, Rajagopalan B, et al., 2023. Satellites reveal widespread decline in global lake water storage, *Science*, 380(6646): 743-749.
- Song, C., Huang, B., Ke, L., 2013. Modeling and analysis of lake water storage changes on the Tibetan Plateau using multi-mission satellite data. *Remote Sensing of Environment*. 135. 25-35.
- Messenger, M.L., Lehner, B., Grill, G., et al., 2016. Estimating the volume and age of water stored in global lakes using a geo-statistical approach. *Nature Communications*, 7: 13603.
- Fair, Z., Flanner, M., Brunt, K. M., et al., 2020. Using ICESat-2 and Operation IceBridge altimetry for supraglacial lake depth retrievals, *The Cryosphere*, 14(11): 4253-4263.

Xiao, W., Hui, F., Cheng, X., et al., 2023. An automated algorithm to retrieve the location and depth of supraglacial lakes from ICESat-2 ATL03 data, *Remote Sensing of Environment*, 298: 113730.

4) As far as the study by Liljedahl et al. (2021) is concerned, we would like to stress that it reveals variations in water pressure in a hydrological well below an ice sheet margin at the depth of at least 400 m. At the seasonal time scale, these variations are indeed significant (the peak-to-peak amplitude is about 15 m in terms of water head). However, it is important to understand that there is no one-to-one relationship between those pressure variations and variations in water mass. In opposite, those pressure variations systematically show a minimum in late autumn, and then steadily increase in the course of winter and spring, reaching the maximum in early summer. Therefore, as it has been stated by Liljedahl et al. (2021), that the pressure variations cannot be explained by an accumulation of meltwater, since water mass does not increase in the course of winter season. Liljedahl et al. (2021) attribute the revealed variability to “fluid pressure changes at the ice/earth boundary”. Unfortunately, the mechanism of those changes at the ice/earth boundary is not addressed explicitly. In any case, it would be fair to conclude that Liljedahl et al. (2021) show no evidences of a significant impact of groundwater on rock displacements observed at the surface.

This paper may plausibly be all the authors have to go on, but the lag times for groundwater transport can mean that hydraulic head variations can be out of phase with recharge from distal locations. The phase would also vary with depth due to the variation in travel pathlength and residence time with depth. Closer to the surface than the borehole described above, phase lags between recharge and hydraulic head could be much shorter. This makes me skeptical about dismissing groundwater as a signal source based on the arguments above.

Authors: As it is written above, the groundwater originated within the ice sheet is not dismissed, it is a part of the BWS signal detected with GPS data. Unfortunately, elastic deformations sensed by GPS data reflect only variations in the total surface load around the GPS station, they do not allow individual water storage compartments to be isolated.

As for “Further, the product has no treatment of groundwater”, we calculated the loadings of the PCRaster Global Water Balance (PCR-GLOBWB) and the WaterGAP2 Global Hydrology Model (WGHM) groundwater components (Lines 477-483) in Greenland at the KAGA site (Sutanudjaja et al., 2018; Müller et al., 2021). The vertical displacement caused by groundwater loads at KAGA station is less than 1 mm (see Fig. A11) and thereby, the effect on GNSS is small, **less than 5% of the GNSS signal.**

5% of the original signal prior to correction? It is non-negligible compared to the final displacements in Fig 2.

Authors: Thank you very much for the comment. Similar to the total LWS signal shown in Fig A4 above, there is some misunderstanding here. Please kindly refer to the similar reasoning in P10-11 above in the rebuttal.

Fig. A11 The loadings of PCR-GLOBWB (a) and WGHM (b) groundwater components at the KAGA site. This figure is included as ED Fig. 13 in the revised manuscript.

Reference:

Sutanudjaja, E. H., Van Beek, R., Wanders, N., Wada, Y., Bosmans, J. H., Drost, N., ... & Bierkens, M. F. (2018). PCR-GLOBWB 2: a 5 arcmin global hydrological and water resources model. *Geoscientific Model Development*, 11(6), 2429-2453.

Müller Schmied, H., Cáceres, D., Eisner, S., Flörke, M., Herbert, C., Niemann, C., ... & Döll, P. (2021). The global water resources and use model WaterGAP v2. 2d: Model description and evaluation. *Geoscientific Model Development*, 14(2), 1037-1079.

I don't have additional specific analyses to suggest to deal with groundwater contributions to the signal.

Authors: Thank you very much for your valuable comments about the groundwater above. In line with you and reviewer 4, we clarify our treatment of groundwater in Lines 501-516.

Originality:

Storage—

Seasonal storage of meltwater is well-established. For example, prior work has measured time series of ‘bed separation,’ the local water storage in the subglacial drainage system (e.g., Andrews et al. 2014). These studies have the advantage of pinpointing the storage location but exist for only a few points and time periods. Other work has used radar to identify the spatial variability of water storage at the bed over 10s to 100s of km length scales (Chu et al., 2016). Ran et al. (2018) used GRACE data to show a time delay, peaking in midsummer, between RACMO’s meltwater generation and total mass loss (i.e., water storage). The finding is required since RACMO, MAR, and other regional climate models do not simulate water transport through hydrologic pathways and so they instantly remove water from the ice sheet. Thus, the disconnect peaks midsummer when meltwater generation is at maximum.

Authors: Previous studies focused on estimating “local water storage” with in-situ measurements (Andrews et al. 2014), extent of bed water with radar data (Chu et al., 2016), and large-scale water mass variations with GRACE data (Ran et al., 2018) are acknowledged in the new variant of the introductory section (Lines 65-76). It is worth to note that Ran et al. (2018) is our previous paper.

The current paper is the first one to demonstrate that GNSS-based vertical bedrock displacement can be used for observing and understanding hydrological processes within the GrIS (and, therefore, within other ice bodies on Earth). Furthermore, our study provides, for the first time, quantitative estimates of buffered water accumulation and release. Finally, our results have major implications for modelling the Surface Mass Balance (SMB) of the GrIS. We show that current SMB model runoff may contain systematic errors, requiring additional scaling of up to ~20% for the warmest years, in particular for the northern and northeastern GrIS. An adequate behavior of SMB models in warm years is particularly important for making accurate projections of the future behavior of the GrIS, when the climate is warmer than today. With the GrIS acting as the major contributor to future sea level rise, improving these projections has vast geographic, economic, and community implications.

The fact that regional climate models do not simulate water transport and, therefore, do not contain the signal of “buffered water” is at the core of our study. Among other, this fact is the starting point of our discussion in the Section “Analytic model of the buffered water signal in GNSS elastic loading data”.

I think the study is novel in using surface deformation to make inferences about seasonal changes in water-mass distribution, provided sufficient confidence can be established in the non-hydrological corrections to the signal. The revised manuscript text and responses to the reviewer still leave some mechanistic gaps for me: what presumably matters for vertical elastic deformation measurable at the GNSS stations is sufficient lateral mass movement within the radius of sensitivity. How each of the appearances and disappearances of “observable” mass translate into measurable redistributions of mass that contribute to the GNSS signal is not always clear. The paper would benefit from a clearer and more fulsome articulation of the physical processes involved, with attention to distinguishing between processes that would and would not contribute to the GNSS signal, including an explanation for the sign of the contribution.

Authors: Thank you very much for the comment. We have added a paragraph that describes how vertical elastic deformation measurable at the GNSS stations can be used to detect lateral mass movement. The theory behind elastic deformation is well known and published by Farrell in 1972. Many publications have used this theory to study e.g. change in the distribution of ice, snow, water, or atmospheric mass. By

monitoring the deformation with GNSS, it is possible to place constraints on the amplitude of the change in mass and the location of that mass (Barletta 2024, Wahr 2013, Hansen 2021). Here we describe the basic principle and demonstrate for the first time how GNSS can be used to place constraints on the change in mass due to englacial water accumulation, storage and ultimate release.

Hansen, K., Truffer, M., Aschwanden, A., Mankoff, K., Bevis, M., Humbert, A., et al. (2021). Estimating ice discharge at Greenland's three largest outlet glaciers using local bedrock uplift. *Geophysical Research Letters*, 48, e2021GL094252. <https://doi.org/10.1029/2021GL094252>

Wahr, J., S. A. Khan, T. van Dam, L. Liu, J. H. van Angelen, M. R. van den Broeke, and C. M. Meertens (2013), The use of GPS horizontals for loading studies, with applications to northern California and southeast Greenland, *J. Geophys. Res. Solid Earth*, 118, 1795–1806, doi:10.1002/jgrb.50104.

Barletta, V. R., Bordoni, A., & Khan, S. A. (2024). GNET derived mass balance and glacial isostatic adjustment constraints for Greenland. *Geophysical Research Letters*, 51, e2023GL106891. <https://doi.org/10.1029/2023GL106891>

>This paper presents time series of storage/release at 22 regions around the ice sheet. However, since the compartment/precise area driving the signal is not clear, it is also not clear how the results can be used to advance understanding of hydrologic systems.

Authors: The loading signals depend on the mass of the load and its distance from the GNSS station (e.g., see Wahr et al., 2013, in doi:10.1002/jgrb.50104). It is always possible to quantify the contribution of a particular part of the GrIS to the surface loading signal at a given location. As we have shown with our sensitivity analysis presented above (Fig. A3), this contribution is mostly limited to the radius of ~200 km (Lines 657-670). In view of that, each GPS station on bedrock can contribute to understanding buffered water storage processed around that station.

I don't think the authors have described this well enough in a manuscript intended for a journal of interest to all scientists. The details are buried in an extended figure and the

basic physics not explained in the introduction. The radially symmetric influence is what I initially assumed, leading to my confusion about why the mass-change accounting was done on the basis of irregularly shaped hydrological catchments (see independent comments). A short paragraph is needed in the manuscript introduction that clearly states the premise of the analysis in sufficient detail that it also makes sense to glaciologists familiar with the mass-balance and hydrological regimes of the Greenland ice sheet.

Authors: Thank you very much for the comment. We have clarified the mechanism in the introduction (see Lines 82-90) in the main text.

Scaling factor--

Large melt biases and uncertainties in RACMO/MAR due to factors such as clouds, albedo, and other is well established (e.g., Fettweis et al., 2020).

>This paper demonstrates that a scaling factor is required to force RCM melt to better match GNSS residuals. The scaling factors are highly variable, centered around 1 (Fig. 4), and has a relatively weak correlation with melt amount. How to incorporate the result into hydrologic studies is unclear, and so the results would seem to be most relevant to RCM testing and validation studies.

Authors: Thank you very much for the comments. We admit that the estimated scaling factors suffer from uncertainties, which may reach 10-20% (see Extended Data Table 2). However, when one looks at the scaling factors for warm, ordinary and cold years(Lines 254-257), it is quite clear to see the difference (see Fig. A12). At the same time, an application of these scaling factors is an intrinsic part our methodology, which allows for a more accurate estimation of signal triggered by buffered water (see Extended Data Fig. 1). Furthermore, they offer a novel way to calibrate SMB models, so that they can better estimate the future behavior of the ice sheet in view of projected Arctic warming (see Sect. "Discussion"). Taking into account all these considerations, we decided to keep the estimation of the aforementioned scaling factors in the paper. In addition, Fettweis et al. (2020) was a previous study by some of our co-authors.

I would suggest labelling the result “tuned” rather than “calibrated”. I would also suggest dialing back the presentation of the scaling of the model as a robust result, without a bit of text reassuring the reader that there is no circularity in the use of the SMB.

Authors: Thank you very much for the comment. We have clarified these issues in our response to the independent comments of reviewer 3. Please refer to that response above in P10 of the rebuttal.

As for the remark “How to incorporate the result into hydrologic studies is unclear, and so the results would seem to be most relevant to RCM testing and validation studies”, we would like to respond as follows. GNET data offer a novel and unique way to improve GrIS SMB estimates from regional climate models. Those models are currently the tool of choice for estimating GrIS-integrated surface melt rate. Despite their generally good and consistent performance, considerable uncertainties remain in the modelled melt products³⁸. Having independent estimates of adjustments required to improve/calibrate the melt products from regional climate models, as provided in this study, is therefore highly valuable for the Greenland mass balance research community. Among other, regional climate models require adjustments in this way for abnormally warm summers. This is particularly relevant in view of projected Arctic warming¹⁻⁵. Extremely high summer temperatures today may become normal in the foreseeable future. Thus, good model performance for warmer years is critical to project ice sheet behaviour and associated sea level rise across coming decades.

We have included this text in the Lines 189-199 in the “Discussion” section.

I think the main value added here is having an independent means---albeit an indirect one that requires corrections larger than the signal of interest (not unlike GRACE)---to infer hydrological quantities.

Authors: Thank you very much for your positive evaluation.

Fig. A12 The scaling factors for cold, ordinary, and warm years. This figure is the updated Fig. 4 in the revised manuscript.

I see the point of the above figure, but is there any more sensible way to organize the x axis? Is it already organized by latitude or another physical variable? Perhaps using dots (or box-whisker) instead of lines would be better since there is no meaning in the lines connecting the dots.

Authors: Thank you very much for the comment. The stations in the x axis are addressed in the clockwise direction, starting from MARG station at the northwest corner. In addition, the stations are not connected as lines. Please see the revised Fig. 4.

=====

Werder MA, Hewitt IJ, Schoof CG and Flowers GE (2013) Modeling channelized and distributed subglacial drainage in two dimensions. *Journal of Geophysical Research: Earth Surface* 118(4), 2140–2158. doi:10.1002/jgrf.20146.

Hoffman, M.J., Andrews, L.C., Price, S.A., Catania, G.A., Neumann, T.A., Lüthi, M.P., Gulley, J.D., Ryser, C., Hawley, R.L., & Morriss, B.F. (2016). Greenland subglacial drainage evolution regulated by weakly connected regions of the bed. *Nature Communications*, 7.

Liu, L., Khan, S.A., Dam, T.V., Ma, J.H., & Bevis, M.G. (2017). Annual variations in GPS-measured vertical displacements near Upernavik Isstrøm (Greenland) and contributions from surface mass loading. *Journal of Geophysical Research: Solid Earth*, 122, 677 - 691.

Zwally, H.J., Abdalati, W., Herring, T.A., Larson, K.M., Saba, J.L., & Steffen, K. (2002). Surface Melt-Induced Acceleration of Greenland Ice-Sheet Flow. *Science*, 297, 218 - 222.

Liljedahl, L.C., Meierbachtol, T.W., Harper, J.T., van As, D., Näslund, J., Selroos, J., Saito, J., Follin, S., Ruskeeniemi, T., Kontula, A., & Humphrey, N.F. (2021). Rapid and sensitive response of Greenland's groundwater system to ice sheet change. *Nature Geoscience*, 14, 751 - 755.

Andrews, L.C., Catania, G.A., Hoffman, M.J., Gulley, J.D., Lüthi, M.P., Ryser, C., Hawley, R.L., & Neumann, T.A. (2014). Direct observations of evolving subglacial drainage beneath the Greenland Ice Sheet. *Nature*, 514, 80-83.

Li, W., Shum, C.K., Li, F., Zhang, S., Ming, F., Chen, W., Zhang, B., Lei, J., & Zhang, Q. (2020). Contributions of Greenland GPS Observed Deformation From Multisource Mass Loading Induced Seasonal and Transient Signals. *Geophysical Research Letters*, 47.

Chu, W., Schroeder, D.M., Seroussi, H., Creyts, T.T., Palmer, S.J., & Bell, R.E. (2016). Extensive winter subglacial water storage beneath the Greenland Ice Sheet. *Geophysical Research Letters*, 43, 12,484 - 12,492.

Ran, J., Vizcaíno, M., Ditmar, P., van den Broeke, M.R., Moon, T.A., Steger, C.R., Enderlin, E.M., Wouters, B., Noël, B.P., Reijmer, C.H., Klees, R., Zhong, M., Liu, L., & Fettweis, X. (2018). Seasonal mass variations show timing and magnitude of meltwater storage in the Greenland Ice Sheet. *The Cryosphere*.

Fettweis, X., et al., (2020). GrSMBMIP: intercomparison of the modelled 1980–2012 surface mass balance over the Greenland Ice Sheet. *The Cryosphere*.

Reviewer Reports on the Second Revision:

Referee #4 (Remarks to the Author):

Thank you for addressing my and the other reviewers' comments in a thorough and convincing manner. I am satisfied with the changes, and with some clarifications, I consider this paper ready for publication.

I. 77

Consider rephrasing this to "poor spatial and temporal resolution".

I. 95

I suggest repositioning the sentence "Note that resulting..." and the following one before the current sentence about the correction of nuisance. This would improve the reading flow.

I. 100

I recommend reformulating the third question to: "Is it possible to use the GNSS data to improve the runoff estimates from SMB models?" The current wording is specific but somewhat difficult to understand without prior reading of the paper.

I. 110

The additions made since the last version read a bit rough. I suggest the following revision: "The pattern is similar for all stations: a slow downward motion (corresponding to the accumulation of stored water) from February to April..."

I. 403

This section could benefit from a brief general introduction explaining your method. For instance: "We identify various nuisance signals and categorize them into those that can be completely neglected (e.g., ice discharge, lakes) and those that need to be included in the uncertainty analysis..." It might also be helpful to mention and refer to the treatment of shallow-tundra groundwater and deep groundwater beneath the ice, as detailed in the 'Contribution of Groundwater Storage in Greenland' section.

I. 413

Please refer to the alternative SMB model (MAR) by name directly, so readers do not need to look it up in the table.

I. 422

This paragraph requires further clarification. What do you mean by saying that the GWS signal in tundra areas is not subtracted? Subtracted from what? For uncertainty analysis? How does this relate to the SMB model? Additionally, when you state that the RMS signal is small, do you mean the assumed error or the signal itself? Does this imply that it is neglected in the uncertainty estimation or not?

Referee #5 (Remarks to the Author):

This review focuses primarily on the authors' responses to Reviewer #3. Below are some independent comments that I made while reading the manuscript. After reading the previous reviews, I see that the major themes from my independent comments were already discussed by the reviewers. However, given that I had the same questions while reading the revised manuscript, there is still some way to go to fully address the previous reviewer's concerns.

See also comments in magenta in the attached document.

Referee #3 (Remarks to the Author):

Note: Additional evaluation of author responses by referee #5 shown in magenta. To keep the responses as concise as possible, I have not commented on responses regarding sections I have previously commented on or responses that I believe to be appropriate.

See also attached PDF

This review focuses primarily on the authors' responses to Reviewer #2.

I have made comments directly in the response-to-reviewer file (in red) to address the authors' responses. Below I have made a few comments of my own while I read the revised manuscript, prior to reading the reviews.

Authors: Thank you very much for your time to review this paper and your excellent comments. We are very grateful for your valuable comments and have addressed them one-by-one. Our responses are shown below in light blue.

In addition, we have incorporated our responses to the additional comments of reviewer 3 to the response to reviewer 2 from the last round in the end of this rebuttal.

I was surprised that in the opening comments, a (subglacial) groundwater system is neglected as a possible pathway for water export that originates as surface melt. Supraglacial, englacial, subglacial (specified to be ice-bed interface) pathways are mentioned, but even in a bedrock substrate there must be a non-zero component of subglacial groundwater recharge and transport where the ice-bed interface is temperate. Groundwater seems important to acknowledge in a study focused on water.

Authors: Thank you very much for the comment. We agree with the reviewer that groundwater is a possible pathway. In line with the comments from you and reviewer 4, we acknowledge groundwater is a part of the buffered water storage signal observed in this study and include the related words in the main text (Please see Lines 501-516).

Line 66 onward: I personally find the use of “hydrostatic pressure” for water pressure confusing, as some might use “hydrostatic pressure” to refer to the ice overburden pressure, which is not what the authors mean. Suggest using instead “basal water pressure” or “basal effective pressure”.

Authors: Thank you very much for the comment. We agree with the reviewer. They are changed as basal water pressure in lines 67, 69 and 71.

Lines 69-70: “However, ice sheet regions with a channelised drainage system may alternate with regions lacking an efficient drainage, the latter retaining high hydrostatic pressure.” “Alternate” is not quite the right notion. The glacier bed and basal drainage system are highly heterogeneous in space, so even regions where channels are present could be dominated by high basal water pressure if the drainage system is not hydraulically well connected to the channels.

Authors: Thank you very much for the comment. The statement is modified as follows: “Importantly, such drainage systems as well as the glacier bed are highly heterogeneous, so that even regions where channels are present could be dominated by high basal water pressure if the drainage system is hydraulically poorly connected to the channels.” Please see Lines 69-72 in the revised manuscript.

Referee #5: This clarification is important, but I think some additional context (i.e., one or two sentences) about the conceptual model of seasonal changes in subglacial drainage would be helpful before jumping straight to a subtle but important point about heterogeneous basal hydraulic connectivity.

Line 74: “An ice penetrating radar can detect basal water”. Sorta, but this vastly oversimplifies. IPR can detect frozen/thawed bed and various other properties from which something about water might be inferred. Adjust wording slightly to improve

precision, e.g., even “can detect properties indicative of basal water” would make this more palatable.

Authors: Corrected. Thank you very much for the comment. Please see Lines 75-76 in the main text.

Lines 80-82: “...bedrock’s elastic response to total mass variations in the vicinity of a given GNSS station”: This is a critical summary of the method and, in my view, should make the mechanism that relates the process to the observable clear for a broad readership: is GNSS mounted on bedrock outside of the ice sheet registering crustal flexure due to variable transient amounts of water mass being stored in the vicinity (all around the GNSS station, including on/in/under the ice)? Reporting the length-scale of influence/sensitivity would also be used here: over what radial distance/area is a given station GNSS sensitive? By what mechanisms do the different sources of GNSS signal contribute? For example, how extensive must the water redistribution be to be measurable? Is it correct that only horizontal redistribution matters? Some related information is presented deep within the extended figures, but a brief synopsis would help readers here. I felt there was a mechanistic gap throughout the paper, as it was assumed the reader would immediately understand that GNSS-measured bedrock elevation changes could be equated to the mass of water leaving the ice sheet. It would be appreciated if the authors would connect those dots more clearly.

Authors: Thank you very much for the comment.

As for the question “is GNSS mounted on bedrock outside of the ice sheet registering crustal flexure due to variable transient amounts of water mass being stored in the vicinity (all around the GNSS station, including on/in/under the ice)?”, the answer is yes. The GNET stations were installed on the bedrock around the GrIS periphery. It is sensitive to all the processes which cause mass changes on/in/under the ice close to the station. We have included a statement to reflect this in the main text (see new Lines 374-375).

Referee #5: This is a critical clarification. It could be integrated more smoothly into the text, e.g. “We consider records from 22 Greenland GNSS Network (GNET) stations (Fig. 2), which contain information about the bedrock’s elastic response to total mass variations within ~200 km of a given GNSS station (see Spatial sensitivity of GNSS loading data)”

As for the question “Reporting the length-scale of influence/sensitivity would also be used here: over what radial distance/area is a given station GNSS sensitive?”, based on the sensitivity test in Lines 707-712, it reveals a significant contribution from the near field: mass changes within 200 km from the station contribute about 80% to the total loading displacements. There is little sensitivity to mass changes beyond 500 km from the station.

As for the question “By what mechanisms do the different sources of GNSS signal contribute? For example, how extensive must the water redistribution be to be measurable? Is it correct that only horizontal redistribution matters?”, the GNSS loading is based on pure elastic theory. The measurability depends on the magnitude and radial distance of vertically-integrated fluid masses to the GNSS station. Thereby, only horizontal redistribution of mass matters.

Referee #5: The authors have elsewhere in this document and in the extended data shown that the horizontal mass displacement must be on the order of tens of kms. This information should be added to the main text in question to address the reviewer’s comment. This would also help to address remaining reviewer concerns.

As for the remark “Some related information is presented deep within the extended figures, but a brief synopsis would help readers here. I felt there was a mechanistic gap throughout the paper, as it was assumed the reader would immediately understand that GNSS-measured bedrock elevation changes could be equated to the mass of water leaving the ice sheet. It would be appreciated if the authors would connect those dots

more clearly.”, we have added the following brief synopsis to the manuscript: “Elastic deformation of the Earth's crust occurs when stress is applied by a mass, causing the crust to alter its shape. Once the stress is removed, the crust returns to its original form. For instance, changes in the distribution of ice, snow, water, or atmospheric mass result in regional elastic deformation of the crust. By continuously and precisely monitoring the position and movement of the Earth's surface, GNSS provides quantitative information on regional mass changes in glaciers, ice sheets, and water storage systems. In general, the accumulation of mass at the surface leads to subsidence of the crust, while the removal of mass causes crustal uplift. This elastic deformation happens instantaneously when mass is redistributed.” Please see Lines 82-90 in the revised manuscript.

Line 95 cf. 88: Somewhat confusing at this point in the manuscript that GNSS data appear to be used to test SMB models (question iii, line 88), but that model output is being used to correct the GNSS signal (line 95). Can a sentence of clarification be introduced to dispel concern that there is circularity in the process? The scaling is reported as a major result in the opening paragraph of the paper, so this seems important.

Authors: Thank you for this valuable comment. Because the scaling factors applied to runoff estimates from SMB models are estimated simultaneously with parameters describing the buffered water storage, circularity is not an issue. To make this more clear in the text, we reformulated the third study objective as follows: “(iii) is it possible to better explain the BWS signal in GNSS data by scaling the runoff from SMB models?” (lines 99-100).

Referee #5: I believe the circularity the reviewer has highlighted still exists in the manuscript. The uncalibrated RACMO runoff data is used to remove the SMB signal from the GNSS signal, then RACMO scaling factors are derived by fitting modelled uplift to the cleaned GNSS signal.

Lines 97-102: It would help if stated explicitly that downward motion equates to accumulation of stored water and vice versa, presumably due to proximity of GNSS to locations of accumulation. I'm thinking about mass changes in general and how station position relative to the mass change might lead to downward (proximal for mass accumulation) vs upward (distal from mass accumulation) responses.

Authors: Thank you very much for the comment. In the revised version, we now state this explicitly in the lines 111-113 as “In general, downward motion corresponds to accumulation of stored water and vice versa.”

Referee #5: This is a helpful clarification. However, the second sentence, “Therefore, we interpret the signal as water accumulation starting from the onset of the melt season”, could be further clarified by referencing the GNSS sensitivity footprint and horizontal mass transport. I have repeatedly suggested this change since I believe it is in the spirit of the previous reviewer's comments

For elastic loading, the bulge (representing the transition from downward to upward motion) occurs when the mass is located 40 degrees to 100 degrees (~4,000 km to 10,000 km) radial distance from the GNSS site. In the case of GNET, all mass changes on the GrIS are far away from the bulge and the general directional principle holds. Moreover, the GNET stations are mostly sensitive to near-field mass changes, as indicated by the sensitivity test discussed in Lines 693-705.

Reference:

Farrell, W. E. (1972), Deformation of the Earth by surface loads, *Rev. Geophys.*, 10(3), 761–797, doi:10.1029/RG010i003p00761.

Line 124: “tuned” may be better than “calibrated” since this is an empirical scaling

Authors: Thank you very much for the comment. Based on the in situ GNSS data,

we evaluate the runoff modelled by RACMO 2.3p2. We contend that the calculated factors reveal an actual deficiency of the modelled runoff. Applying these factors make the modelled runoff more realistic (see Lines 152-169 in the main text). Therefore, in this case we prefer to use the term “calibrated”. If the reviewer believes that “tuned” is a better word, we are fully open and happy to take this comment.

Referee #5: In line with my first major comment, I am strongly on the side of Reviewer #3 here. Without more evidence, I am not convinced the SMB scale factors represent actual deficiency of the RACMO modelled runoff, and so the wording around the scale factors should be much more careful.

Line 128-129: “total BWS”. Doesn’t the mass need to move substantially from its place of origin to cause crustal elevation change (i.e., meltwater that accumulates locally in a supraglacial lake would not change the mass distribution substantially enough to be detected in the form of bedrock elevation changes outside the ice sheet)?

Authors: Thank you very much for the comment. If meltwater is stored locally (near the location where it was produced, e.g., meltwater that accumulates locally in a supraglacial lake), this, most probably, will not affect the GNSS signal. We designed a test and find that horizontal mass displacement must be in the order of tens of km to emerge in elastic loading data. For more details, we refer to our answer to additional comments of Reviewer 3 to our response to Reviewer 2 from the previous round (please see Fig. S2 in P24 in the last part of this rebuttal letter).

Referee #5: I don’t believe this comment has been appropriately addressed in the manuscript text. At this point in the text, it seems that “BWS” is implicitly assumed to mean “Buffered water storage within ~200 km of the GNSS station that has been sourced from tens of kms away from the storage location, such that it induces a detectable GNSS signal”. However, I (and, it seems, the previous reviewers) interpret BWS to mean the total water mass transiently stored within the glacier and groundwater system, precisely as defined by Eq. A10. Without reference to horizontal transport of mass from distant to nearby locations, the Eq. A10 definition of BWS doesn’t

necessarily imply a bedrock elastic response. This needs to be clarified in the manuscript text.

Line 547: “relatively” => “relative”

Authors: Corrected. Thank you very much for the comment. Please see Line 579 in the main text.

Eqn (A9): somewhere around here, shouldn't there be a reference to a horizontal length scale? A station represents a point location that is sensitive to mass change within some region. Eqn (A8) expresses mass change at an individual (mass balance) grid cell which is assumed linearly related to elastic deformation in (A9), but then the spatial relationship between contributing (ice-sheet) grid cells and the (off-ice) station location disappears. In the equations that follow it appears that the BWS inferred at any station is being related to catchment-wide quantities. If this interpretation is correct---that each GNSS station is assumed sensitive only to mass changes that occur in its hydrological catchment (the furthest points of which may be further away than adjacent catchments)-- it would help readers like me if this were stated early in the main manuscript. It would also be useful to depict the catchments associated with each GNSS station in one of the map figures. There is some delineation of basins in the context of supraglacial lakes, but not a map that shows the catchment basins used in calculations for each GNSS station.

Authors: Thank you very much for the comment. We agree that it would be more fair to assume that GNSS station is sensitive to mass changes that occur in all surrounding hydrological catchment. We have adjusted the text as follows: “Let us consider the total BWS $S(t)$ in the drainage basins located around the current GNSS station (more specifically, in the drainage basins that substantially affect the elastic deformations at the current GNSS station)” (lines 607-609). For more information about the spatial sensitivity of GNSS loading data, we kindly refer to responses to the additional

comments of Reviewer 3 to the response to Reviewer 2 from the last round at the end of this rebuttal.

Lines 657-670: I see some new spatial analysis was added here. This radially symmetric approach relates to the end of the above comment. Ext Fig 14 on page 58 partially answers some of the above questions about length scale (the radius of influence), but it would still be of use to put a sentence or two in the Intro of the main text.

Authors: We added the following remark (lines 92-94: “(we estimate the sensitivity radius of GNSS data as approximately 200 km, see *Spatial sensitivity of GNSS loading data*)”. We also included a more extended discussion about the GNSS loading mechanism in the introductory section see Lines 82-90.

I still have questions about the treatment of groundwater in the paper, though the authors have done what seems to be an extensive analysis with multiple models. Thematically consistent with the above, it would be useful to hear something about physical processes and what anticipated groundwater processes would contribute to GNSS signals, as well as a bit more detail on the possible limitations of the treatment of groundwater in the two models used to estimate the contributions to deformation.

Authors: Thank you very much for the comment. In this study, two types of groundwater can be identified:

- 1) Shallow groundwater in tundra areas, which results from snow melting and rainfalls there. In principle, this is an ordinary component of the Terrestrial Water Storage, which is described by various hydrological models, including those addressed in the manuscript (see Extended Data Fig. 13 modelled by PCR-GLOBWB and WGHM). Based on the personal communication with Prof. Mark Bierkens, who is a developer of PCR-GLOBWB, it is very difficult to model the groundwater accurately in those regions, but the model outcome is enough for a first-order estimate of its magnitude. In this study, we analyze the uncertainty of groundwater estimates in Sect.

Uncertainties of the mean annual cycle of residual vertical displacements (see Lines 422-428 in the manuscript). We take KAGA as a representative station, and quantify the difference between mean annual vertical displacements per calendar month with and without groundwater signal subtracted from GNSS data using PCR-GLOBWB and WGHM models. This is basically the modelled groundwater signal itself. It shows that the RMS signal from PCR-GLOBWB and WGHM model is 0.25 mm and 0.02 mm, respectively (see Fig. S1), which is small, as compared to the BWS signal (see Extended Data Fig. 5). We have included this information in Lines 422-428 of the revised manuscript.

2) Deep groundwater below the ice sheet (its presence was detected by a hydrological well down to the depth of hundreds of meters⁶²). It is a product of ice sheet melting (both at the surface and the ice sheet base). To the best of our knowledge, little is known about variations in the deep groundwater mass. Here, we consider the signal from deep groundwater as a part of the total BWS signal we detect in GNSS data. Unfortunately, elastic loading data do not allow the deep groundwater to be separated from the rest of the BWS.

I am pleased to see that the authors have clarified the two categories of groundwater and that category (2) is included as a possible contributor to the BWS GNSS signal. However, given the questionable representation of all the processes related to (1) in the groundwater models, perhaps it is best to explain that any processes in (1) missing from the models would also be a part of the GNSS signal (see page 45).

Fig. S1 The mean annual vertical displacements per calendar month of modelled groundwater signal from PCR-GLOBWB (left) and WGHM (right) models, which are shown in monthly time-series in Extended Data Fig. 13.

References

Müller Schmied, H., Cáceres, D., Eisner, S., Flörke, M., Herbert, C., Niemann, C., ... & Döll, P. The global water resources and use model WaterGAP v2.2d: Model description and evaluation. *Geoscientific Model Development* 14, 1037-1079 (2021).

Sutanudjaja, E. H., Van Beek, R., Wanders, N., Wada, Y., Bosmans, J. H., Drost, N., ... & Bierkens, M. F. PCR-GLOBWB 2: a 5 arcmin global hydrological and water resources model. *Geoscientific Model Development* 11, 2429-2453 (2018).

Below is our response (shown in light blue) to the comments of Reviewer #3 (shown in red), which addressed our response (shown in blue) to the concerns of Reviewer #2 (shown in black).

Reviewer Comments & Author Rebuttals

Referee #2 (Remarks to the Author):

Note: Additional referee evaluation of author responses shown in red. The authors made the changes clear and easy to follow. Nicely done!

Authors: Thank you again for your excellent suggestions which have significantly improved the manuscript. We are very grateful.

The subject of this paper, seasonal storage and release of liquid water in the Greenland ice sheet, is a key component of many unsolved science problems related to the ice sheet's mass balance and ice flow dynamics. Because meltwater cannot exit the ice sheet instantaneously, the time/space evolution of hydrologic systems revolves around water storage. Storage is, therefore, a key state variable of all supraglacial, englacial, and subglacial hydrologic processes and models thereof. Water storage at the bed is a critical aspect of sliding mechanics and geochemical processes. Yet, the processes are too complex, and our understanding is too limited to explicitly include the many hydrologic processes (e.g., the seasonal evolution of a subglacial drainage system) in any large-scale SMB/runoff model.

Different observational methods, such as radar, GPS signals, GRACE, remote sensing imagery, radar and radiometry, have been used to determine water storage in various compartments. Each method presents resolution and interpretation challenges, and several offer only snapshots rather than continuous time series. Breakthroughs with direct measurements of storage thus have the potential to make a strong impact on many different aspects of Greenland hydrology.

Authors: Thank you very much for your time to review this paper and your excellent comments. We agree with your emphasis on the importance of studies on seasonal storage and release of liquid water in the Greenland ice sheet. We are very grateful for

your valuable comments and have addressed them one-by-one as follows.

Concerns:

1. Hydrologic interpretation.

Throughout the paper, the description of the water storage mechanism is quite puzzling.

Examples include:

-- Figure 1 depicts englacial lakes (and subglacial lakes that are actually shown as englacial) — I know of no evidence for lakes existing inside the ice.

Authors: Thank you very much for the comment. Sorry for the misunderstanding. Figure 1 is a *schematic* figure to show the processes related to the liquid water evolution beneath the ice sheet surface and the associated changes of vertical bedrock. The distance of englacial lakes from the ice sheet surface is schematic, just indicating that some water is buffered beneath the ice sheet surface, and does not mean the actual location of englacial lakes is inside the ice sheet. However, based on the above comments, we have improved Fig. 1, to avoid any misunderstanding. For your convenience, the revised Fig. 1 is shown below. Please see lines 218-232 in the revised manuscript.

The representation of relevant hydrological features in the new Figure 1 seems fine to me, with the exception that differences in the relationship between the white dashed line and the bedrock mound on which the GNSS station is mounted are hard to discern in the 3 panels.

Authors: Thank you very much for your agreement on the revised Fig. 1. In the first panel, the dashed line is the same as the boundary of ice-bedrock interface, because the vertical deformation does not start. However, the deformation is non-zero in panels b and c. In line with the reviewer's comment, the difference between the white dashed

line and ice-bedrock interface is increased ~ 1.5 times (see Fig. 1), as compared to the previous version of the plot.

Referee #5: The caption makes me expect that the separation between actual and modelled (white dashed line) bedrock should be highest in Early Summer and lower in Late Summer, but to me it looks like the separation is greatest in Late Summer? Could the 1.5x increase in separation between bedrock and dashed line be further increased to 2—3x to make it unambiguous?

Fig. S1 The revised Fig. 1.

-- The text concerning loading is confusing because a phase change is not a mass change, so water mass loading must be due to the redistribution of water from one place to another (not stated).

Authors: Thanks for this thoughtful comment. The “phase change” refers to melting and refreezing processes. In order to avoid potential confusion, we have added the following sentence in the introductory section (Lines 84-85):

“Note that bedrock displacement time-series describe the mass of buffered water rather than the total water mass, since surface loading does not change if water re-freezes in-place.”

This clarification is helpful, but I also struggled with the lack of clarity around water mass change: the mass needs to move in order to elicit elastic deformation. Throughout the text, it almost seemed like the GNSS signals were being related to water mass, rather than a change in water mass (see also my independent comments).

Authors: Thank you very much for the comment. The reviewer is correct that the GNSS measures the elastic deformation caused by the change in water mass. This is fully consistent with the manuscript. We clean GNSS data from nuisance signals to obtain residual deformations that are referred to *the Buffered Water Storage (BWS)*. The latter is defined as “the water mass temporarily buffered within the GrIS, being mostly released into the ocean before the onset of next melt season” (lines 64-65). To make the relationship between water mass and vertical deformations even more clear for readers, we have extended our clarification cited above (in blue) as follows: “Note that resulting bedrock displacements peak during the melt season and gradually reduce thereafter.

This is because they describe BWS rather than the total water mass (surface loading does not change if water re-freezes in-place)” (lines 95-98).

Referee #5: Following previous comments, I don’t believe this concern has been appropriately addressed.

Regarding the highlighted text, would it be more clear to say, “This is because bedrock displacement describe mass redistribution from distant, upstream source locations to downstream BWS locations closer to the GNSS stations”?

-- The manuscript intro/discussion is structured around water storage in the firn compartment, but why this is driving the loading signal versus other compartments is not clear. Melt across most of the firn area will result in a short-distance mass redistribution, vertically within the upper 10 m. Only a relatively narrow reach of the far lower accumulation area has substantial horizontal flow and accumulation of water, but is there evidence that this is enough to drive the loading signal? Regardless, all runoff is accumulated toward the margin. Whereas a SMB model may simulate a large amount of meltwater generation across the accumulation zone, the subglacial drainage system, not the firn compartment, has the most accumulating water mass and is therefore the most likely candidate for loading. Water at the bed cannot be dismissed by invoking fast conduit drainage. A large body of literature on subglacial hydrology debates issues such as a) how far inward on the ice sheet fast-draining conduits can actually develop (e.g., Werder et al., 2013); and, b) how much area of the bed sits between conduits and does not drain quickly (e.g., Hoffman et al., 2016).

Authors: We agree that the introductory section was too much focused on processes in the firn layer, which resulted in some mismatch between that section and the rest of the manuscript. To solve this issue, we have re-organized the introductory section to put more emphasis on water at the bed (Lines 65-76). The firn layer is mentioned in the Discussion Section only briefly, and primarily in the context of our interpretation of the scaling factors to be applied to runoff estimates provided by the SMB model (Lines 189-199).

Firn processes did not seem over-emphasized in the revised manuscript. This point, however, also relates to the lengthscales over which mass must move to create detectable deformation.

Authors: We are pleased to see that we reached a consensus concerning the text about the firm processes. For the issue of the length scale over which mass must move to create detectable deformation, we designed a sensitivity analysis which is similar to Fig.1 in Wahr et al., (2013) and Fig. 6 in Bevis et al., (2016). By considering a mass formed by a 1-m water layer over a disk of 20 km radius, we calculated the deformation at different distances from the disk centre (Fig. S2). This plot clearly shows that horizontal displacement of a mass must be of the order of at least tens of km to be seen in elastic loading data. Thus, a local re-distribution of meltwater within the firm layer can be safely ignored. We have included the clarification in Lines 707-712.

Fig. S2 Deformation caused by a load of 1-m water disk of 20 km radius at different distances from the disk centre. This result is consistent with Figure 1 by Wahr et al. (2013).

Referee #5: This sensitivity experiment helps to clarify the question of how far mass must be transported to produce a measurable vertical bedrock displacement. However, the authors have not addressed whether or not a horizontal displacement of tens of kms is reasonable for the proposed BWS components (supraglacial, englacial, subglacial, groundwater beneath the ice sheet) with reference to appropriate literature. A summary of this information should also be integrated into the main text.

Reference

Wahr, J., S. A. Khan, T. van Dam, L. Liu, J. H. van Angelen, M. R. van den Broeke,

and C. M. Meertens (2013), The use of GPS horizontals for loading studies, with applications to northern California and southeast Greenland, *J. Geophys. Res. Solid Earth*, 118, doi:10.1002/jgrb.50104.

As for “A large body of literature on subglacial hydrology debates issues such as a) how far inward on the ice sheet fast-draining conduits can actually develop (e.g., Werder et al., 2013); and, b) how much area of the bed sits between conduits and does not drain quickly (e.g., Hoffman et al., 2016).”, the loading observed using GNSS verticals is the bedrock response to integrated mass changes. We have removed SMB and other non-ice sources, attributing the remaining changes to temporally buffered meltwater. But it is difficult to further differentiate the pathways or storages of the buffered meltwater by using GNSS data alone. We hope this study will stimulate further research to address ice sheet hydrology from a new perspective like this.

I don't think the reviewer is asking the authors to attempt to extract more detail from the GNSS data related to the style of subglacial drainage or the architecture of the system. I think the point here was to say that water in the subglacial drainage system can have a long residence time even though channels may form, i.e., “Water at the bed cannot be dismissed by invoking fast conduit drainage”. I'm not sure any further action is needed, but readers would like to know that the authors have thoroughly considered the plausibility of the mechanisms proposed to explain the GNSS data.

Authors: Thank you very much for the excellent comment. We agree that “water in the subglacial drainage system can have a long residence time even though channels may form”. This is acknowledged, among others, in the introductory section. It contains now the following text, which also considers the independent recommendations of Reviewer 3: “Ultimately, the accumulation of water results in a formation of a sub-glacial drainage system, which facilitates a water release into the ocean, so that the basal water pressure is reduced. Importantly, such drainage systems as well as the glacier bed are highly heterogeneous, so that even regions where channels are present could be

dominated by high basal water pressure if the drainage system is hydraulically poorly connected to the channels” (lines 67-72).

2. Signal robustness.

Because the water storage signal is a small residual that emerges after many large corrections to the original data, the robustness of the signal is dependent on the confidence of the removal of so-called nuisance signals. Perhaps some of the following issues are partly or solely responsible for the spatial variability of the results.

Authors: We agree that this signal is relatively small, but it is clearly visible (see, e.g., panel (h) in Extended Data Fig. 4 in the updated manuscript, please see Lines 851-853) and reliably estimated in this study. To ensure the robustness of our estimates, we conducted a thoughtful analysis by evaluating all corrections with the State-of-the-Art models or data in the community. Please see revised ED Fig. 5 (Lines 854-863) and the following text. It is worth to note that we also compared the GNSS-based water storage signal with that based on independent satellite gravimetry data (Fig. 3, see Lines 245251). This comparison confirmed that this signal is real and not a data artefact.

I'm going to assume this response will be evaluated by someone with GNSS signal processing expertise.

Authors: Thank you very much for the comment. We believe Reviewer 1 is a GNSS expert.

-- Prior work by Lui et al. (2017) demonstrated the difficulty of SMB corrections in Greenland due to far-field SMB influences. In other words, where does the SMB signal come from? Lui et al. (2017) argue that SMB from glaciers 1200 km away can impact Greenland GNSS signals. In fact, they found that correcting for SMB adversely inflated the residual. The manuscript does not make clear how this problem was handled nor is the Lui et al. (2017) paper cited.

Authors: Thank you very much for the comment. Liu et al. (2017) was the previous study by one of the co-authors. Liu et al. (2017) studied Upernavik glacier using two GNSS stations located in the nearby bedrock area. We agree with Liu et al. (2017) that far-field SMB signals must be taken into account when interpreting the annual amplitude of the GNSS verticals. In this study we use an ice-sheet-wide SMB model (RACMO) rather than a local or regional model and therefore correct for the far-field signal. We now state in line 376 that the SMB model covers the entire GrIS (and not just a small area).

In addition, we made a sensitivity study (Lines 657-670) to clarify how mass changes over the entire ice sheet, including glaciers 1,200 km away, may affect the observed mass loading signal. By taking the detrended SMB in July 2012 (Fig. A3 in the rebuttal) as the ice sheet mass change signal, we analyze the sensitivity of GNSS loading observed at KAGA as an example. The processing strategy is to take the KAGA station as the center, create ring-shaped zones outward with a step width of 50 km, calculate the vertical displacement caused by SMB inside each ring, and then normalized with the entire Greenland. The results show that cumulative contribution to the detrended SMB loading reaches approximately 80% when the distance from KAGA station is 200 km. The sensitivity curve (Fig. A3) reveals a significant contribution from near field: SMB changes within 200 km from KAGA contribute by about 80% to the detrend SMB loading displacements. There is little sensitivity to SMB beyond 500 km from KAGA.

Fig. A3 (a) Detrended SMB mass anomalies in July 2012. (b) Cumulative contribution to the detrended uplift at KAGA due to SMB loading, depending on the radius of the buffer zone around that station. All the numbers are normalized with the loading uplift caused by the SMB signal over entire Greenland. The X axis shows a step of 50 km in the distance range of 0-1,500 km. This figure is included as ED Fig. 14 in the revised manuscript.

I don't have much expertise in this area, but the response and added analysis seem reasonable. Length scales of influence were one of my big questions, so I would like to see some related information in the intro to the main text.

Authors: Thank you very much for the comment. We have included a sensitivity analysis above, i.e., Fig. S2. We have also added the following remark to the introductory section of the main text (lines 92-94): “(we estimate the sensitivity radius of GNSS data as approximately 200 km, see Spatial sensitivity of GNSS loading data)”.

We designed a sensitivity test to clarify the length scale over which mass must move to create detectable deformation. It is found that horizontal displacement of a mass must be of the order of at least tens of km to be seen in elastic loading data (see Fig. S2 above). We have included the text in Lines 707-712 in the main text.

-- The analysis does not appear to consider the seasonal loading signal caused by ice flow across the ablation zone (c.f., unloading by calving). The speed of the ice sheet in the ablation zone increases during summer, typically by a factor of 2-3 (see Zwally et al. (2002) for the seminal paper, with much confirming work published since then). Speed typically peaks mid-summer, similar to the GNSS loading signal. Ice acceleration causes horizontal advection of ice thickness gradients — more ice is moved low that must be slowly beaten back by ablation. How much of the residual could be due to this effect?

Authors: We agree with the reviewer that the seasonal loading signal might be caused by the ice flows. In this study, we have considered this effect. Please see the effect of seasonal variations in ice speed analyzed in the Section “Contribution of ice discharge”, which is a part of Methods (see lines 416-429). To obtain an upper bound of the contribution of seasonal ice discharge variations to the observed vertical displacements, we use Jakobshavn Isbræ (JI) as an example because it shows the largest ice flow velocity seasonality amongst GrIS outlet glaciers. By investigating the GNSS loading signal observed at the KAGA GPS station, which is located ~5 km from the JI terminus, we conclude that this effect (i.e., $< \sim 1\text{mm}$) is not significant.

Seems reasonable.

Authors: Thank you very much for your positive reaction.

-- The product used for LWS corrections is a global product and is not necessarily well-suited for Arctic regions dominated by permafrost hydrology. Some areas have many lakes fluctuating lake levels. Further, the product has no treatment of groundwater, whereas Liljedahl et al., (2021) show very large head swings in the groundwater system along the ice margin. LWS is a big correction and uncertainties will vary substantially around the ice sheet.

I share the concerns about groundwater.

Authors: Thank you very much for the comment. We have clarified this issue in response to your independent comments in the rebuttal and modified the paper (see Lines 501-516), by taking the comments from Reviewers 3 and 4.

Authors: Thanks for this comment. We agree that lakes and groundwater play important roles in Arctic regions dominated by permafrost hydrology. Indeed, the GLDAS model we considered in the previous variant of the manuscript lacks both water storage components. To solve this problem, we also add two hydrological models (see Extended Data Fig. 5c in Line 854), i.e., WaterGAP2 Global Hydrology Model (WGHM) and PCRaster Global Water Balance (PCR-GLOBWB), both of which take groundwater into account. Furthermore, we explicitly consider water mass variations of lakes in Arctic regions (Lines 431-475).

1) Three State-of-The-Art models are utilized in this study in total: GLDAS, PCR-GLOBWB and WGHM. Even though these models are not specific for Arctic regions, we believe they are accurate enough at least for the first-order magnitude **as evidenced by?**. Note that WGHM is tailored in Arctic region by considering the existence of permafrost. A comparison with other models showed that WGHM agrees best with independent in-situ observations collected in Arctic region (Gädeke et al., 2020). In addition, as we have shown with our sensitivity analysis of GNSS-based loading signal presented above (Fig. A3), this contribution is mostly (~80%) limited to the radius of ~200 km. Most parts of the Arctic regions are quite far (> 1000 km) from the GNSS stations in Greenland, and thereby the mass variations over there are little impact the GNSS based water storage signals observed in this study.

The highlighted text above is a good argument for the lakes, but not for groundwater around Greenland.

Authors: Thank you very much for the comment. We agree that some mass redistribution of hydrological origin occurs also at smaller distances from Greenland than 1000 km. However, the groundwater output from the state-of-the-art models, i.e., PCR-GLOBWB and WGHM, covers the all the tundra area around Greenland. From the personal communication with Prof. Mark Bierkens, a developer of PCR-GLOBWB, little is known to model the groundwater accurately in those regions, but may be fair to give a first-order estimate of its magnitude. We have clarified this issue and please kindly refer to those in P10-11 above in the rebuttal.

Taking the KAGA as an example: the vertical displacement at KAGA in response to global LWS loading derived from the 3 hydrological models, excluding Greenland, during the study period (2009-2015) ranges from -1.0 to ~1.5 mm (Fig. A4). This magnitude is sufficiently small to allow for neglecting the potential influence of LWS on the Greenland ice sheet meltwater storage.

What exactly do these models estimate as loading: water mass on the land surface? Is it possible that any of the elastic deformation could be attributable to seasonal recharge/discharge of the aquifer? Do the models include estimates of aquifer compressibility (assuming fractured rock aquifers) such that the vertical changes associated with possible compaction/expansion due to seasonal changes in hydraulic head?

Authors: Thank you very much for the comment. First of all, as for “What exactly do these models estimate as loading: water mass on the land surface?”, PCR-GLOBWB and WGHM models estimate the Total Water Storage (TWS) variations of the hydrological components, e.g., including canopy water storage, groundwater storage, lake/reservoir storage, snow water storage, soil water storage. Note that GLDAS model lacks the groundwater and lake components (unlike PCR-GLOBWB and WGHM).

As for “Is it possible that any of the elastic deformation could be attributable to seasonal recharge/discharge of the aquifer? Do the models include estimates of aquifer compressibility (assuming fractured rock aquifers) such that the vertical changes associated with possible compaction/expansion due to seasonal changes in hydraulic head?”, we could not find any information about the possible presence of underground aquifers below GNET stations. Most probably, nobody has investigated that issue yet. It is known, however, that poroelastic deformations observed at the Earth’s surface due to a compressibility of underlying aquifers show a totally different behavior, as compared to elastic deformations, which are our primary focus. For instance, recharge of an underlying compressible aquifer results in an uplift, not in a subsidence of the Earth’s surface (Larochelle et al, 2022). We have carefully checked the GNSS time-series for all stations in Greenland and did not find evidences of such a signal. From this, we conclude that recharge/discharge of compressible aquifers below GNET stations likely provides a negligible contribution to the observed residual displacements, if at all.

I have trouble agreeing with the final highlighted statement if I am interpreting the data correctly: Fig A4 below shows variations due to LWS of +/-1 mm that were neglected in the corrections? +/-1mm does not seem small compared to the signals in Figure 2, many of which are substantially less than +/- 5mm. They also appear to be out of phase so would make the signals in Fig 3 less pronounced if used in the correction.

Authors: Thank you very much for the comment. There is some misunderstanding here. Fig. A4 shows the *total* Land Water Storage (LWS) signal caused by hydrological processes world-wide excluding Greenland. Since our target is to analyze the Buffered Water Storage (BWS) in GrIS, thereby, the LWS signal shown in Fig. A4 is considered as a *nuisance* signal. Therefore, even though the LWS signal shown Fig. A4 may reach +/-1 mm in total, it has to be subtracted. What we care about are uncertainties of the estimated LWS signal. In Sect. *Uncertainties of the mean annual cycle of residual vertical displacements*, we take KAGA as a representative station to investigate, among

others, those uncertainties in the context of the mean annual cycles of residual vertical displacements (Fig. 2). We find that those uncertainties are small, i.e., around 0.21 mm (see Extended Data Fig. 5).

Without knowing enough to have confidence in the models' treatment of groundwater around Greenland, I am hesitant to dismiss this correction as negligible. Depending on what the models are doing, one solution may be not to attempt to correct for groundwater or claim it is negligible, but rather include it in what is being measured. If most of the groundwater recharge is coming from the ice sheet, then this is a source of water that would ideally be accounted for anyway.

Authors: Thank you very much for the comment. We fully agree with the reviewer that the groundwater coming from the melt of ice sheet should be a part of the BWS signal considered in this study. Similar suggestion is also made by Reviewer 4. We have changed the main text (see Lines 501-516) to clarify this aspect. Importantly, the discussion above is limited to groundwater variability estimated by hydrological models: we refer there to shallow groundwater originated from precipitation outside the ice sheet.

Referee #5: I think this clarification meaningfully addresses part of the reviewer's comments here. Including groundwater beneath the GrIS as a BWS component seems important. It seems that the models and products available to evaluate LWS are not as detailed as the authors and reviewers would like, leaving some open questions (page 45).

Fig. A4 Time series of the vertical displacements at KAGA in response to LWS loading excluding Greenland.

References:

GLDAS: Li, B., M. Rodell, S. Kumar, H. Beaudoin, A. Getirana, B. F. Zaitchik, et al. (2019) Global GRACE data assimilation for groundwater and drought monitoring: Advances and challenges. *Water Resources Research*, 55, 7564-7586. doi:10.1029/2018wr024618

PCR-GLOWB: Sutanudjaja, E. H., Van Beek, R., Wanders, N., Wada, Y., Bosmans, J. H., Drost, N., ... & Bierkens, M. F. (2018). PCR-GLOBWB 2: a 5 arcmin global hydrological and water resources model. *Geoscientific Model Development*, 11(6), 2429-2453.

WGHM: Müller Schmied, H., Cáceres, D., Eisner, S., Flörke, M., Herbert, C., Niemann, C., ... & Döll, P. (2021). The global water resources and use model WaterGAP v2. 2d: Model description and evaluation. *Geoscientific Model Development*, 14(2), 1037-1079.

Gädeke, A., Krysanova, V., Aryal, A. et al. (2020). Performance evaluation of global hydrological models in six large Pan-Arctic watersheds. *Climatic Change* 163, 1329–1351, <https://doi.org/10.1007/s10584-020-02892-2>

Larochelle, S., Chanard, K., Fleitout, L., Fortin, J., Gualandi, A., Longuevergne, L., et

al. (2022). Understanding the geodetic signature of large aquifer systems: Example of the Ozark Plateaus in central United States. *Journal of Geophysical Research: Solid Earth*, 127, e2021JB023097. <https://doi.org/10.1029/2021JB023097>

2) As for “Some areas have many lakes fluctuating lake levels.”, indeed, there are many lakes in the Arctic region. To further quantify the potential water mass impact of lakes onto GNSS loading signal in Greenland, we consider three types of lakes (Lines 431-475): lakes in Pan-Arctic region (north of 60 °N) in general; supraglacial lakes (SGL) on GrIS; and proglacial lakes in the coastal part of Greenland.

Fig. A5 Distribution of lakes in Pan-Arctic region. This figure is included as ED Fig. 7 in the revised manuscript.

Lakes in Pan-Arctic region: Specifically, the monthly lake water storage data of some large lakes were directly downloaded from a recently published dataset by Yao et al., (2023). In addition, we accessed the intra-annual lake level datasets from three portals,

i.e., Hydroweb (<https://hydroweb.theia-land.fr/>), Global Reservoirs and Lakes Monitor (G-REALM, https://ipad.fas.usda.gov/cropexplorer/global_reservoir/), and Database for Hydrological Time Series of Inland Waters (DAHITI, <https://dahiti.dgfi.tum.de/en/>), respectively (Cretaux et al., 2011; Birkett et al., 2009; Schwatke et al., 2015). For months without lake area data in the dataset provided by Yao et al., (2023), we follow the empirical relationship between lake water level and lake surface area from a similar previous study (Song et al., 2013), by interpolating the missing lake area data with ampler level data and converting the lake area and level data to water storage changes.

It is worth mentioning that the recorded lakes (38 in total, indicated by blue circles in Fig. A5) are typically large, with 33 of them having a surface area greater than 500 km², accounting for 61.6% of the total surface area of Arctic lakes extracted by Yao et al. (2023). Note that the reference surface area here is derived from the HydroLAKES polygon (Messenger et al., 2016). For the remaining smaller inland lakes (red circles in Fig. A5), the area data are very limited. Therefore, we employ a scale-up strategy to estimate their impact on mass loading (Song et al., 2013).

Taking KAGA station as an example, the vertical displacement induced by the load of Arctic lake water storage during the study period (2009-2015) ranges from -0.1 to 0.04 mm (Fig. A6). This magnitude is sufficiently small to allow for neglecting the potential influence of these lakes in the Arctic region.

Fig. A6 Vertical displacements of lake loading at KAGA station. This figure is included as ED Fig. 8 in the revised manuscript.

I had trouble grasping how lake-level variations in distal locations like Russia would influence the GNSS signals around Greenland. It would seem necessary to know where mass lost/gained from the lakes goes in order to correctly account for this signal. It does not seem sufficiently important to pursue further.

Authors: Thank you very much for the comment. As it is written above, PCR-GLOBWB and WGHM models estimate the Total Water Storage (TWS) variations of almost all hydrological components, e.g., canopy water storage, groundwater storage, lake/reservoir storage, snow water storage, soil water storage (Müller Schmied et al., 2021; Sutanudjaja et al., 2018). Note that GLDAS model lacks the lake components (unlike PCR-GLOBWB and WGHM). We considered the potential impact of lakes in Arctic region separately since the GLDAS model is utilized from the very beginning of the study.

SGL on the GrIS: In addition, we assessed the vertical displacement caused by water mass loading from SGL on the GrIS. Firstly, we divided the GrIS into 10-by-10 km equal-area cells and identified all the cells with SGL using ~ 300,000 high

spatiotemporal resolution optical images from Sentinel-2 and Landsat 8/9 over 2017-2022, as shown in Fig. A7. Then, the monthly SGL area changes during the melting season (i.e., May-September) were derived. The SGL area changes data are distributed via <https://doi.org/10.5281/zenodo.10398558>. Note that it is problematic to obtain the monthly SGL area change for entire GrIS before the launch of Sentinel-2B satellites in 2017 due to a high cloud contamination. We assumed, however, that the SGL area changes over 2009-2015 interval was similar to those over 2017-2022. When converting the SGL area changes to mass changes, we assumed that the maximum depth of GrIS supraglacial lakes is around 8.5 metres (Fair et al., 2020; Xiao et al., 2023), which allowed us to estimate the upper limit of water mass changes Fig. A8. The result shows that even for the maximal depth of 8.5 m, the magnitude of loading signal caused by SGL mass changes is ~0.3 mm, which is negligible, compared with the BWS signal. I had trouble grasping the assumptions in this analysis. If a supraglacial lake drains vertically downward, and the water is retained locally at the bed, there would presumably be no deformation measured at the GNSS station, right? Are the authors assuming that loss of lake mass instantaneously contributes to englacial/subglacial water mass somewhere significantly downstream? This would not be unreasonable, but I just don't know. Similarly, what is assumed when lakes increase in size? My assumption would be no deformation since the meltwater is likely to be (relatively) locally sourced and does not, therefore, represent a substantial lateral redistribution of mass. Here again, relating the processes envisioned to the measured signals would help a broad audience understand the assumptions and better appreciate this work.

Authors: Thank you very much for the comment. We believe there is some misunderstanding here. The purpose of the analysis of SGL here was to demonstrate that SGL provide a minor contribution to the total buffered water storage signal observed, in order to respond the concern of reviewer 2.

As for “If a supraglacial lake drains vertically downward, and the water is retained locally at the bed, there would presumably be no deformation measured at the GNSS station, right?”, yes, if a supraglacial lake drains vertically downward, and the water is retained locally at the bed, being re-frozen there, it is not a part of BWS and is not sensed by our analysis.

As for “Are the authors assuming that loss of lake mass instantaneously contributes to englacial/subglacial water mass somewhere significantly downstream? This would not be unreasonable, but I just don’t know. ”, our analysis is limited to the BWS (that is, the water mass that ultimately discharges into the ocean). A horizontal displacement of water mass with a subsequent re-freezing is beyond the scope of our analysis. This is in line with the fact that only very significant horizontal displacements of mass (tens of km and more) can affect elastic loading data (see Fig. S2 above and the associated discussion there).

As for “Similarly, what is assumed when lakes increase in size? My assumption would be no deformation since the meltwater is likely to be (relatively) locally sourced and does not, therefore, represent a substantial lateral redistribution of mass.”, our input is *residual* elastic loading data, which are obtained by subtracting all the nuisance signals, including that from the Surface Mass Balance (SMB). However, models of SMB do not take into account the fact that it takes meltwater some time to leave the ice sheet: they assume that the mass associated with meltwater runoff disappears at the moment of meltwater production. Thus, an increase in a lake size, may, in principle, be visible in

residual loading data as a temporal increase in mass (i.e., as a subsidence signal). Of course, in practice this signal is likely not visible: our analysis has shown that it is too small.

Proglacial lakes in Greenland: Similar to the SGL, we also consider the impact of proglacial lakes in Greenland using Hydrolake database (Fig. A9). In total, 2 687 proglacial lakes are taken into account. Most of the proglacial lakes are smaller than 5 km², while the largest one could reach ~100 km². The loading signal caused by proglacial lakes is small. At KAGA station, for instance, it is of the order of only 0.02 mm (Fig. A10).

This signal size seems suitably small. I think I can imagine the assumptions being made to relate the change in lake size with station elevation, but it would not hurt to state them explicitly.

Authors: Thank you very much for your agreement. Because of the very limited monthly lake area and lake level data availability of proglacial lakes in Greenland, we assume similar seasonality of lake areas as that of SGL. After that, we apply method by Song et al. (2013) to estimate lake volumes/masses.

Fig. A7 Spatial distribution of cells with SGLs. The grey cells indicate that there were no SGL over 2017-2022 from optical images. This figure is included as ED Fig. 9 in the revised manuscript.

Fig. A8 The vertical displacements at KAGA station caused by SGL mass changes. This figure is included as ED Fig. 10 in the revised manuscript.

Fig. A9 Spatial distribution of proglacial lakes in Greenland. This figure is included as ED Fig. 11 in the revised manuscript.

Fig. A10 | The vertical displacements at KAGA station caused by proglacial lake mass changes over entire Greenland. This figure is included as ED Fig. 12 in the revised

manuscript.

References:

Cretaux J-F., Arsen A., Calmant S., et al., 2011. SOLS: A lake database to monitor in the Near Real Time water level and storage variations from remote sensing data, *Advances in space Research*, 47, 1497-1507.

Birkett, C.M., Reynolds, C., Beckley, B., et al., 2009. From Research to Operations: The USDA Global Reservoir and Lake Monitor, chapter 2 in 'Coastal Altimetry', Springer Publications, eds. S. Vignudelli, A.G. Kostianoy, P. Cipollini and J. Benveniste, ISBN 978-3-642-12795-3, 2010.

Schwatke C, Dettmering D, Bosch W, et al., 2015. DAHITI—an innovative approach for estimating water level time series over inland waters using multi-mission satellite altimetry, *Hydrology and Earth System Sciences*, 19(10), 4345-4364.

Yao F, Livneh B, Rajagopalan B, et al., 2023. Satellites reveal widespread decline in global lake water storage, *Science*, 380(6646): 743-749.

Song, C., Huang, B., Ke, L., 2013. Modeling and analysis of lake water storage changes on the Tibetan Plateau using multi-mission satellite data. *Remote Sensing of Environment*. 135. 25-35.

Messenger, M.L., Lehner, B., Grill, G., et al., 2016. Estimating the volume and age of water stored in global lakes using a geo-statistical approach. *Nature Communications*, 7: 13603.

Fair, Z., Flanner, M., Brunt, K. M., et al., 2020. Using ICESat-2 and Operation IceBridge altimetry for supraglacial lake depth retrievals, *The Cryosphere*, 14(11): 4253-4263.

Xiao, W., Hui, F., Cheng, X., et al., 2023. An automated algorithm to retrieve the location and depth of supraglacial lakes from ICESat-2 ATL03 data, *Remote Sensing of Environment*, 298: 113730.

4) As far as the study by Liljedahl et al. (2021) is concerned, we would like to stress that it reveals variations in water pressure in a hydrological well below an ice sheet margin at the depth of at least 400 m. At the seasonal time scale, these variations are indeed significant (the peak-to-peak amplitude is about 15 m in terms of water head). However, it is important to understand that there is no one-to-one relationship between those pressure variations and variations in water mass. In opposite, those pressure variations systematically show a minimum in late autumn, and then steadily increase in the course of winter and spring, reaching the maximum in early summer. Therefore, as it has been stated by Liljedahl et al. (2021), that the pressure variations cannot be explained by an accumulation of meltwater, since water mass does not increase in the course of winter season. Liljedahl et al. (2021) attribute the revealed variability to “fluid pressure changes at the ice/earth boundary”. Unfortunately, the mechanism of those changes at the ice/earth boundary is not addressed explicitly. In any case, it would be fair to conclude that Liljedahl et al. (2021) show no evidences of a significant impact of groundwater on rock displacements observed at the surface.

This paper may plausibly be all the authors have to go on, but the lag times for groundwater transport can mean that hydraulic head variations can be out of phase with recharge from distal locations. The phase would also vary with depth due to the variation in travel pathlength and residence time with depth. Closer to the surface than the borehole described above, phase lags between recharge and hydraulic head could be much shorter. This makes me skeptical about dismissing groundwater as a signal source based on the arguments above.

Authors: As it is written above, the groundwater originated within the ice sheet is not dismissed, it is a part of the BWS signal detected with GPS data. Unfortunately, elastic deformations sensed by GPS data reflect only variations in the total surface load around the GPS station, they do not allow individual water storage compartments to be isolated.

Referee #5: I think the reviewers are correct to suggest that the definition of groundwater that contributes to the GNSS BWS signal should be expanded. If I am interpreting correctly, all groundwater components that are not included as part of the LWS correction would contribute to the BWS signal. This should be added to Line 143 in the text. Instead of restricting to only “groundwater stored beneath the GrIS”, the

BWS variations represent all changes in groundwater storage that have not been removed by the LWS correction.

As for “Further, the product has no treatment of groundwater”, we calculated the loadings of the PCRaster Global Water Balance (PCR-GLOBWB) and the WaterGAP2 Global Hydrology Model (WGHM) groundwater components (Lines 477-483) in Greenland at the KAGA site (Sutanudjaja et al., 2018; Müller et al., 2021). The vertical displacement caused by groundwater loads at KAGA station is less than 1 mm (see Fig. A11) and thereby, the effect on GNSS is small, less than 5% of the GNSS signal.

5% of the original signal prior to correction? It is non-negligible compared to the final displacements in Fig 2.

Authors: Thank you very much for the comment. Similar to the total LWS signal shown in Fig A4 above, there is some misunderstanding here. Please kindly refer to the similar reasoning in P10-11 above in the rebuttal.

Fig. A11 The loadings of PCR-GLOBWB (a) and WGHM (b) groundwater components at the KAGA site. This figure is included as ED Fig. 13 in the revised manuscript.

Reference:

Sutanudjaja, E. H., Van Beek, R., Wanders, N., Wada, Y., Bosmans, J. H., Drost, N., ... & Bierkens, M. F. (2018). PCR-GLOBWB 2: a 5 arcmin global hydrological and water resources model. *Geoscientific Model Development*, 11(6), 2429-2453.

Müller Schmied, H., Cáceres, D., Eisner, S., Flörke, M., Herbert, C., Niemann, C., ... & Döll, P. (2021). The global water resources and use model WaterGAP v2. 2d: Model description and evaluation. *Geoscientific Model Development*, 14(2), 10371079.

I don't have additional specific analyses to suggest to deal with groundwater contributions to the signal.

Authors: Thank you very much for your valuable comments about the groundwater above. In line with you and reviewer 4, we clarify our treatment of groundwater in Lines 501-516.

Originality:

Storage—

Seasonal storage of meltwater is well-established. For example, prior work has measured time series of ‘bed separation,’ the local water storage in the subglacial drainage system (e.g., Andrews et al. 2014). These studies have the advantage of pinpointing the storage location but exist for only a few points and time periods. Other work has used radar to identify the spatial variability of water storage at the bed over 10s to 100s of km length scales (Chu et al., 2016). Ran et al. (2018) used GRACE data to show a time delay, peaking in midsummer, between RACMO’s meltwater generation and total mass loss (i.e., water storage). The finding is required since RACMO, MAR, and other regional climate models do not simulate water transport through hydrologic pathways and so they instantly remove water from the ice sheet. Thus, the disconnect peaks midsummer when meltwater generation is at maximum.

Authors: Previous studies focused on estimating “local water storage” with in-situ measurements (Andrews et al. 2014), extent of bed water with radar data (Chu et al., 2016), and large-scale water mass variations with GRACE data (Ran et al., 2018) are

acknowledged in the new variant of the introductory section (Lines 65-76). It is worth to note that Ran et al. (2018) is our previous paper.

The current paper is the first one to demonstrate that GNSS-based vertical bedrock displacement can be used for observing and understanding hydrological processes within the GrIS (and, therefore, within other ice bodies on Earth). Furthermore, our study provides, for the first time, quantitative estimates of buffered water accumulation and release. Finally, our results have major implications for modelling the Surface Mass Balance (SMB) of the GrIS. We show that current SMB model runoff may contain systematic errors, requiring additional scaling of up to ~20% for the warmest years, in particular for the northern and northeastern GrIS. An adequate behavior of SMB models in warm years is particularly important for making accurate projections of the future behavior of the GrIS, when the climate is warmer than today. With the GrIS acting as the major contributor to future sea level rise, improving these projections has vast geographic, economic, and community implications.

The fact that regional climate models do not simulate water transport and, therefore, do not contain the signal of “buffered water” is at the core of our study. Among other, this fact is the starting point of our discussion in the Section “Analytic model of the buffered water signal in GNSS elastic loading data”.

I think the study is novel in using surface deformation to make inferences about seasonal changes in water-mass distribution, provided sufficient confidence can be established in the non-hydrological corrections to the signal. The revised manuscript text and responses to the reviewer still leave some mechanistic gaps for me: what presumably matters for vertical elastic deformation measurable at the GNSS stations is sufficient lateral mass movement within the radius of sensitivity. How each of the appearances and disappearances of “observable” mass translate into measurable redistributions of mass that contribute to the GNSS signal is not always clear. The paper would benefit from a clearer and more fulsome articulation of the physical processes involved, with attention to distinguishing between

processes that would and would not contribute to the GNSS signal, including an explanation for the sign of the contribution.

Authors: Thank you very much for the comment. We have added a paragraph that describes how vertical elastic deformation measurable at the GNSS stations can be used to detect lateral mass movement. The theory behind elastic deformation is well known and published by Farrell in 1972. Many publications have used this theory to study e.g. change in the distribution of ice, snow, water, or atmospheric mass. By monitoring the deformation with GNSS, it is possible to place constraints on the amplitude of the change in mass and the location of that mass (Barletta 2024, Wahr 2013, Hansen 2021). Here we describe the basic principle and demonstrate for the first time how GNSS can be used to place constraints on the change in mass due to englacial water accumulation, storage and ultimate release.

Referee #5: The authors have made some effort to address the “mechanistic gap” explained by the reviewer but more explanations are needed. No matter how well established the theory is, the explanations in the current work need to be clear and thorough to the reader and stand alone without requiring expert knowledge of the theory. My comments throughout should highlight areas that this should be improved.

Hansen, K., Truffer, M., Aschwanden, A., Mankoff, K., Bevis, M., Humbert, A., et al. (2021). Estimating ice discharge at Greenland's three largest outlet glaciers using local bedrock uplift. *Geophysical Research Letters*, 48, e2021GL094252.

<https://doi.org/10.1029/2021GL094252>

Wahr, J., S. A. Khan, T. van Dam, L. Liu, J. H. van Angelen, M. R. van den Broeke, and C. M. Meertens (2013), The use of GPS horizontals for loading studies, with applications to northern California and southeast Greenland, *J. Geophys. Res. Solid Earth*, 118, 1795–1806, doi:10.1002/jgrb.50104.

Barletta, V. R., Bordoni, A., & Khan, S. A. (2024). GNET derived mass balance and glacial isostatic adjustment constraints for Greenland. *Geophysical Research Letters*, 51, e2023GL106891. <https://doi.org/10.1029/2023GL106891>

>This paper presents time series of storage/release at 22 regions around the ice sheet. However, since the compartment/precise area driving the signal is not clear, it is also not clear how the results can be used to advance understanding of hydrologic systems.

Authors: The loading signals depend on the mass of the load and its distance from the GNSS station (e.g., see Wahr et al., 2013, in doi:10.1002/jgrb.50104). It is always possible to quantify the contribution of a particular part of the GrIS to the surface loading signal at a given location. As we have shown with our sensitivity analysis presented above (Fig. A3), this contribution is mostly limited to the radius of ~200 km (Lines 657-670). In view of that, each GPS station on bedrock can contribute to understanding buffered water storage processed around that station.

I don't think the authors have described this well enough in a manuscript intended for a journal of interest to all scientists. The details are buried in an extended figure and the basic physics not explained in the introduction. The radially symmetric influence is what I initially assumed, leading to my confusion about why the mass-change accounting was done on the basis of irregularly shaped hydrological catchments (see independent comments). A short paragraph is needed in the manuscript introduction that clearly states the premise of the analysis in sufficient detail that it also makes sense to glaciologists familiar with the mass-balance and hydrological regimes of the Greenland ice sheet.

Authors: Thank you very much for the comment. We have clarified the mechanism in the introduction (see Lines 82-90) in the main text.

Scaling factor--

Large melt biases and uncertainties in RACMO/MAR due to factors such as clouds, albedo, and other is well established (e.g., Fettweis et al., 2020).

>This paper demonstrates that a scaling factor is required to force RCM melt to better match GNSS residuals. The scaling factors are highly variable, centered around 1 (Fig. 4), and has a relatively weak correlation with melt amount. How to incorporate the result into hydrologic studies is unclear, and so the results would seem to be most relevant to RCM testing and validation studies.

Authors: Thank you very much for the comments. We admit that the estimated scaling factors suffer from uncertainties, which may reach 10-20% (see Extended Data Table 2). However, when one looks at the scaling factors for warm, ordinary and cold years (Lines 254-257), it is quite clear to see the difference (see Fig. A12). At the same time, an application of these scaling factors is an intrinsic part of our methodology, which allows for a more accurate estimation of signal triggered by buffered water (see Extended Data Fig. 1). Furthermore, they offer a novel way to calibrate SMB models, so that they can better estimate the future behavior of the ice sheet in view of projected Arctic warming (see Sect. "Discussion"). Taking into account all these considerations, we decided to keep the estimation of the aforementioned scaling factors in the paper. In addition, Fettweis et al. (2020) was a previous study by some of our co-authors.

I would suggest labelling the result "tuned" rather than "calibrated". I would also suggest dialing back the presentation of the scaling of the model as a robust result, without a bit of text reassuring the reader that there is no circularity in the use of the SMB.

Authors: Thank you very much for the comment. We have clarified these issues in our response to the independent comments of reviewer 3. Please refer to that response above in P10 of the rebuttal.

As for the remark "How to incorporate the result into hydrologic studies is unclear, and so the results would seem to be most relevant to RCM testing and validation studies", we would like to respond as follows. GNET data offer a novel and unique way to improve GrIS SMB estimates from regional climate models. Those models are currently the tool of choice for estimating GrIS-integrated surface melt rate. Despite their generally good and consistent performance, considerable uncertainties remain in the modelled melt products³⁸. Having independent estimates of adjustments required to improve/calibrate the melt products from regional climate models, as provided in this study, is therefore highly valuable for the Greenland mass balance research community. Among other, regional climate models require adjustments in this way for abnormally

warm summers. This is particularly relevant in view of projected Arctic warming¹⁻⁵. Extremely high summer temperatures today may become normal in the foreseeable future. Thus, good model performance for warmer years is critical to project ice sheet behaviour and associated sea level rise across coming decades.

We have included this text in the Lines 189-199 in the “Discussion” section.

I think the main value added here is having an independent means---albeit an indirect one that requires corrections larger than the signal of interest (not unlike GRACE)---to infer hydrological quantities.

Authors: Thank you very much for your positive evaluation.

Fig. A12 The scaling factors for cold, ordinary, and warm years. This figure is the updated Fig. 4 in the revised manuscript.

I see the point of the above figure, but is there any more sensible way to organize the x axis? Is it already organized by latitude or another physical variable? Perhaps using dots (or box-whisker) instead of lines would be better since there is no meaning in the lines connecting the dots.

Authors: Thank you very much for the comment. The stations in the x axis are addressed in the clockwise direction, starting from MARG station at the northwest corner. In addition, the stations are not connected as lines. Please see the revised Fig. 4.

=====

Werder MA, Hewitt IJ, Schoof CG and Flowers GE (2013) Modeling channelized and distributed subglacial drainage in two dimensions. *Journal of Geophysical Research: Earth Surface* 118(4), 2140–2158. doi:10.1002/jgrf.20146.

Hoffman, M.J., Andrews, L.C., Price, S.A., Catania, G.A., Neumann, T.A., Lüthi, M.P., Gulley, J.D., Ryser, C., Hawley, R.L., & Morriss, B.F. (2016). Greenland subglacial drainage evolution regulated by weakly connected regions of the bed. *Nature Communications*, 7.

Liu, L., Khan, S.A., Dam, T.V., Ma, J.H., & Bevis, M.G. (2017). Annual variations in GPS-measured vertical displacements near Upernavik Isstrøm (Greenland) and contributions from surface mass loading. *Journal of Geophysical Research: Solid Earth*, 122, 677 - 691.

Zwally, H.J., Abdalati, W., Herring, T.A., Larson, K.M., Saba, J.L., & Steffen, K. (2002). Surface Melt-Induced Acceleration of Greenland Ice-Sheet Flow. *Science*, 297, 218 - 222.

Liljedahl, L.C., Meierbachtol, T.W., Harper, J.T., van As, D., Näslund, J., Selroos, J., Saito, J., Follin, S., Ruskeeniemi, T., Kontula, A., & Humphrey, N.F. (2021). Rapid and sensitive response of Greenland's groundwater system to ice sheet change. *Nature Geoscience*, 14, 751 - 755.

Andrews, L.C., Catania, G.A., Hoffman, M.J., Gulley, J.D., Lüthi, M.P., Ryser, C., Hawley, R.L., & Neumann, T.A. (2014). Direct observations of evolving subglacial drainage beneath the Greenland Ice Sheet. *Nature*, 514, 80-83.

Li, W., Shum, C.K., Li, F., Zhang, S., Ming, F., Chen, W., Zhang, B., Lei, J., & Zhang, Q. (2020). Contributions of Greenland GPS Observed Deformation From Multisource Mass Loading Induced Seasonal and Transient Signals. *Geophysical Research Letters*, 47.

Chu, W., Schroeder, D.M., Seroussi, H., Creyts, T.T., Palmer, S.J., & Bell, R.E. (2016). Extensive winter subglacial water storage beneath the Greenland Ice Sheet. *Geophysical Research Letters*, 43, 12,484 - 12,492.

Ran, J., Vizcaíno, M., Ditmar, P., van den Broeke, M.R., Moon, T.A., Steger, C.R., Enderlin, E.M., Wouters, B., Noël, B.P., Reijmer, C.H., Klees, R., Zhong, M., Liu, L., & Fettweis, X. (2018). Seasonal mass variations show timing and magnitude of meltwater storage in the Greenland Ice Sheet. *The Cryosphere*.

Fettweis, X., et al., (2020). GrSMBMIP: intercomparison of the modelled 1980–2012 surface mass balance over the Greenland Ice Sheet. *The Cryosphere*.

Major comment 1

Without additional evidence, I am not convinced that the SMB scaling factors are a direct result of errors in the RACMO runoff fields. While there will be errors and biases in the RACMO fields, I don't believe the methods applied here are appropriate to derive such corrections. The authors have used a simple ad-hoc relationship between the actual stored mass $S(t)$ and the BWS GNSS signal $s(t)$ (Eq. A15—A16). I can imagine several plausible physical explanations for why the $S(t) - s(t)$ relationship should differ in warm vs cold years that are independent of shortcomings in the RACMO surface runoff field: (1) Might there be differences in the horizontal transport distances from source to buffer between low/high melt years that could necessitate a year-to-year correction in this relationship? (2) Subglacial channel efficiency, the extent of weakly connected bed regions, and subglacial water storage will all depend on the intensity of surface melt. (3) Bare ice exposed at higher elevations in warmer years might change the mass redistribution pattern between warm/cold years. (4) How does groundwater storage respond to moulin inputs in warm/cold years? If you allowed θ to vary yearly, would you still need the year-to-year scaling factors ϵ_k ? If these and other physical processes can explain the necessity for year-to-year scaling factors, then the scaling factors represent a correction to the treatment of the BWS GNSS signal rather than corrections to the SMB field. As a more technical note, the authors explain that their nonlinear system of equations is not well-posed and does not have a physically meaningful unique solution without regularization. Thus, any SMB scale factors that you derive are conditioned on your regularization terms. This is another reason I'm skeptical of interpreting the SMB scaling factors as representing failures of the RACMO model, rather than necessary corrections to the methods in this paper.

Major comment 2

Throughout the main text, the physical processes being invoked need to be explained more clearly. This has been improved in the previous rounds of reviews, but I was still unsure about the physical mechanisms until reading the full methods and extended data. For example:

- The addition, "Note that resulting bedrock displacements peak during the melt season and gradually reduce thereafter. This is because they describe BWS rather than the total water mass (surface loading does not change if water re-freezes in place)", on L95-97 that was added in response to Reviewer #3, made this section more confusing to me on the first read. While it may be obvious to some readers familiar with GNSS processing, others might not be immediately able to interpret that the GNSS signals are a result of mass (mainly liquid water) moving from locations distant to the GNSS stations to downstream locations closer to the stations. I think that emphasizing this mass transfer.

- To make the processes under consideration unambiguously clear, maybe the authors could add a paragraph that explicitly discusses water transport mechanisms and how they may or may not contribute to the observed GNSS signals. For example, "We consider mass variations within ~200 km of GNSS stations arising from horizontal transport of meltwater within the glacier and groundwater systems. To produce detectable GNSS signals, we assume that water is transported tens of kms from distant, upstream source locations to downstream transient storage locations closer to GNSS stations. Transport mechanisms may include [list your hypothesized mechanisms]. [Clarify the processes, e.g. firn percolation and storage, local storage within supraglacial lake basins, etc. that likely don't include enough horizontal transport to produce detectable signals]."

- L111: "In general, downward motion corresponds to an accumulation of stored water and vice versa. Therefore, we interpret this signal as water accumulation starting from the onset of the melt

season (Fig. 1).” On first read, this was confusing: the “stored water” was always part of the GrIS mass (other than small contributions perhaps from temporary storage of rain in snow, firn, supraglacial lakes), so why should it cause downward motion of the GNSS stations? On second read, I realize that this is because the assumption is that the “stored water” has been transferred from locations farther from the GNSS station to locations closer to the station (in particular, the transport distance are far enough to result in a measurable GNSS signal). I think that emphasizing the role of mass transfer here is very important given the broad audience.

Minor comments and technical corrections

L52: I may be missing something, but it is not clear to me how runoff from rainfall increases GrIS mass loss? Rainfall increases surface runoff, but since the mass of the rain was not stored in the ice sheet, I’m not sure this should count as mass loss, which readers probably interpret as mass loss that contributes to sea-level change.

L68: Be consistent about "subglacial" vs. "sub-glacial"

L134: Considering my first major comment, I would be more comfortable seeing an explanation along the lines of, “Importantly, the introduced analytic function takes into account possible inaccuracies in the proposed relationship between BWS and GNSS displacement and errors in the runoff estimated by the adopted SMB model”.

L167—169: The weak-to-moderate correlation (correlation 0.42) should be provided here.

L374: “It is sensible” to “The GNSS stations are sensitive”

L480: Why are supraglacial lakes considered here? Aren’t they part of the BWS?

“Spatial sensitivity of GNSS loading data”: This information is key for understanding the manuscript, this should be better integrated into the main text.

Figure 2: Why do some of the signals (e.g., QAAR, KAPI) appear non-periodic? I believe this is addressed deep within the methods and extended data, but it might help other readers to acknowledge this here.

Eq. A15 and the associated explanation: Related to my second major comment, I am uncomfortable with this explanation. As non-expert in GNSS signal processing and solid earth deformation, explanations in terms of the “surface load”, rather than in terms of the horizontal redistribution of mass, are confusing to me. My concern could be removed by addressing my first major comment.

Referee #5 (Remarks on code availability):

I have reviewed but not attempted to run the provided code. The code structure is simple enough that installation should not be a concern. The code seems extremely specific to this application, so while the code may be used to verify the current results, it would be difficult to generalize the code to new analyses.

Author Rebuttals to Second Revision:

Referee #4 (Remarks to the Author):

Thank you for addressing my and the other reviewers' comments in a thorough and convincing manner. I am satisfied with the changes, and with some clarifications, I consider this paper ready for publication.

Authors: Thank you very much for taking the time to review our paper and for your excellent comments. We appreciate your encouraging feedback and are grateful for your valuable suggestions, which we have addressed one by one. Our responses are shown below in light blue.

1. 77

Consider rephrasing this to “poor spatial and temporal resolution”.

Authors: Corrected. Please refer to line 79 in the revised manuscript.

1. 95

I suggest repositioning the sentence “Note that resulting...” and the following one before the current sentence about the correction of nuisance. This would improve the reading flow.

Authors: Thank you very much for your valuable comment. We have substantially rewritten this part of the introductory section. The reading flow is now consistent with the reviewer's suggestion by rearranging the sentences.

l. 100

I recommend reformulating the third question to: “Is it possible to use the GNSS data to improve the runoff estimates from SMB models?” The current wording is specific but somewhat difficult to understand without prior reading of the paper.

Authors: Corrected. Please refer to line 111 in the revised manuscript.

l. 110

The additions made since the last version read a bit rough. I suggest the following revision: “The pattern is similar for all stations: a slow downward motion (corresponding to the accumulation of stored water) from February to April...”

Authors: Corrected. Please refer to line 121 in the revised manuscript.

l. 403

This section could benefit from a brief general introduction explaining your method. For instance: “We identify various nuisance signals and categorize them into those that can be completely neglected (e.g., ice discharge, lakes) and those that need to be included in the uncertainty analysis...” It might also be helpful to mention and refer to the treatment of shallow-tundra groundwater and deep groundwater beneath the ice, as detailed in the ‘Contribution of Groundwater Storage in Greenland’ section.

Authors: Corrected. Please refer to lines 417-421 in the revised manuscript. We have included the following text:

“We identify various nuisance signals and categorize them into those that can be completely neglected (e.g., ice discharge, lakes) and those that need to be included in the uncertainty analysis. Note that we consider separately shallow groundwater in tundra areas and deep groundwater beneath the ice sheet, as detailed in the *Contribution of Groundwater Storage in Greenland* section.”

l. 413

Please refer to the alternative SMB model (MAR) by name directly, so readers do not need to look it up in the table.

Authors: Corrected. Please refer to line 429 in the revised manuscript.

l. 422

This paragraph requires further clarification. What do you mean by saying that the GWS signal in tundra areas is not subtracted? Subtracted from what? For uncertainty analysis? How does this relate to the SMB model? Additionally, when you state that the RMS signal is small, do you mean the assumed error or the signal itself? Does this imply that it is neglected in the uncertainty estimation or not?

Authors: Thank you very much for your comments. We have further clarified this in the revised manuscript. The GWS signal in tundra areas is not part of the SMB models (SMB is only defined over glacial ice), so it is not subtracted from GNSS data when computing residual vertical displacements. To quantify the impact of this signal in the uncertainty analysis, we consider the GWS outputs from two hydrological models and quantify the RMS

signal. Compared to the BWS signal, the GWS signal is negligible. We have modified the text in the revised manuscript (see lines 438-446).

Thank you again for your excellent suggestions, which have resulted in an improved manuscript. We are very grateful.

Referee #5 (Remarks to the Author):

This review focuses primarily on the authors' responses to Reviewer #3. Below are some independent comments that I made while reading the manuscript. After reading the previous reviews, I see that the major themes from my independent comments were already discussed by the reviewers. However, given that I had the same questions while reading the revised manuscript, there is still some way to go to fully address the previous reviewer's concerns.

See also comments in magenta in the attached document.

Authors: Thank you very much for taking the time to review our paper and for your comments. Based on some comments of Reviewer 5 we feel that parts of the methodology require improved explanation, which we have included in the revised manuscript and rebuttal. We are very grateful for your valuable feedback and have addressed the reviewer's comments one by one. Our responses are shown below in light blue.

Additionally, we have incorporated our responses to the additional comments from reviewer 5 into the response to reviewer 3 from the last round, which can be found at the end of this rebuttal.

Major comment 1

Without additional evidence, I am not convinced that the SMB scaling factors are a direct result of errors in the RACMO runoff fields. While there will be errors and biases in the RACMO fields, I don't believe the methods applied here are appropriate to derive such corrections.

Authors: Thank you very much for your comment. We have modified the manuscript and the rebuttal to make the presentation of our approach more convincing and clear (Please see below).

The authors have used a simple ad-hoc relationship between the actual stored mass $S(t)$ and the BWS GNSS signal $s(t)$ (Eq. A15—A16).

Authors: We believe there is a misunderstanding here. Both $s(t)$ and $S(t)$ refer to the Buffered Water Storage (BWS). $S(t)$ is for BWS in mass, $s(t)$ is for vertical displacements caused by BWS. BWS is less than the total water mass (the latter also accounts for water that re-freezes in-place).

I can imagine several plausible physical explanations for why the $S(t) - s(t)$ relationship should differ in warm vs cold years that are independent of shortcomings in the RACMO surface runoff field: (1) Might there be differences in the horizontal transport distances from source to buffer between low/high melt years that could necessitate a year-to-year correction in this relationship? (2) Subglacial channel efficiency, the extent of weakly connected bed regions, and subglacial water storage will all depend on the intensity of surface melt. (3) Bare ice exposed at higher elevations in warmer years might change the mass redistribution pattern between warm/cold years. (4) How does groundwater storage respond to moulin inputs in warm/cold years?

Authors: Thank you very much for your comment. We further clarify the misunderstanding here. All these hypotheses address different BWS features, which play a role only on a short time scale (during the melt season and a few weeks thereafter). They have only a minor impact on the estimated scaling factors, which control the long-term behavior of the SMB model. To clarify this for the reader, we have extended Section "Analytic model of the BWS signal in GNSS elastic loading data" with the following discussion (lines 699-722):

“Importantly, the last term at the left-hand side of the functional model given by Eq. (A18) describes the accumulation and discharge of the BWS, which is a short-term process. This signal declines exponentially after the end of the melt season (i.e., after the runoff-related signal $r_0(t)$ turns to zero). This is fully consistent with the behavior of BWS, which is primarily produced as a result of ice/firn/snow melting and ends up as discharge into the ocean. This term controls, in the first instance, the estimated water storage time T_{st} and empirical coefficient θ . In contrast, the third term at the left-hand side of Eq. (A18) describes the long-term effect of inaccuracies in the runoff estimated as part of the SMB. The effect of these inaccuracies does not vanish in the course of time. The estimated corrections ϵ_k are mostly controlled by this term, whereas the impact of the fourth term onto those estimates is minor. In order to demonstrate that, we have considered a modified functional model that lacks the BWS-related signal (i.e., the aforementioned fourth term):

$$c + at - \sum_{k=1}^{K(t)} \epsilon_k \int_{t_{k_0}}^{t_{k_e}} r_0(\tau) d\tau = m(t) - \int_{t_0}^t b_0(\tau) d\tau. \quad (A20)$$

Of course, such a functional model is not applicable in the course of the melt season and immediately thereafter.

Therefore, we made an analysis which limited the input data time-series to either 6 months per year (November-April) or even to 4 months per year (December-March). This allowed us to obtain two alternative estimates of corrections ϵ_k (in addition to those based on the original functional model given by Eq. (A18)). The mean of the three estimates, as well as the associated standard deviation, is reported in terms of scaling factor $(1 + \epsilon_k)$ per station per year in Extended Data Table 4. One can see that the standard deviation in the overwhelming majority of cases is less than 0.15. Only one GNET station – GMMA – may show standard deviations larger than 0.3. This means that the obtained estimates of corrections ϵ_k are sufficiently robust; intrinsic uncertainties associated with the spatial distribution of BWS during the melt season have only a minor effect.”

Finally, it is worth mentioning that the processes addressed by the reviewer may indeed affect the GNSS signal. However, this is beyond the scope of our study. In this context, we estimate only one water storage time per GNET station, which describes the average behavior of BWS.

If you allowed θ to vary yearly, would you still need the year-to-year scaling factors ϵ_k ? If these and other physical processes can explain the necessity for year-to-year scaling factors, then the scaling factors represent a correction to the treatment of the BWS GNSS signal rather than corrections to the SMB field.

Authors: Thank you very much for your comments. We believe there is a misunderstanding, which we will further clarify here. “Corrections to the SMB field” actually refer to corrections of the SMB-based runoff. As explained above, they are associated with the third term on the left-hand side of the functional model given by Eq.(18)). The presence of this term allows the scaling factors ϵ_k to be estimated, so that the SMB-based runoff is scaled by $(1+ \epsilon_k)$. Estimating scaling factors ϵ_k on a yearly basis cannot be replaced by estimating coefficients θ on a yearly basis. This is because the additional empirical coefficient θ is only present in the fourth term on the left-hand side of our functional model (the term associated with BWS). Since BWS may indeed show different behavior in cold and warm years, it would be problematic to interpret coefficients θ_k if they are estimated per year instead of scaling factors ϵ_k . In any case, it would be unfair to interpret them as runoff corrections.

One might suggest that it would be instructive to estimate *both* ϵ_k and θ on a yearly basis. However, the inverse problem becomes poorly-conditioned in this case, resulting in unphysical outcomes.

As a more technical note, the authors explain that their nonlinear system of equations is not well-posed and does not have a physically meaningful unique solution without regularization. Thus, any SMB scale factors that you derive are conditioned on your regularization terms. This is another reason I’m skeptical of interpreting the SMB scaling factors as representing failures of the RACMO model, rather than necessary corrections to the methods in this paper.

Authors: As it is explained in the manuscript (lines 683-686), our only constraint is that the mean value of the corrections ϵ_k per GNSS station is zero. This constraint can be interpreted

as an assumption that, on average, the runoff estimates provided by the SMB model are correct over the study period, even though they still may contain errors in individual years. A difference between cold and warm years can be revealed despite this constraint.

Major comment 2

Throughout the main text, the physical processes being invoked need to be explained more clearly. This has been improved in the previous rounds of reviews, but I was still unsure about the physical mechanisms until reading the full methods and extended data. For example:

- The addition, “Note that resulting bedrock displacements peak during the melt season and gradually reduce thereafter. This is because they describe BWS rather than the total water mass (surface loading does not change if water re-freezes in place)”, on L95-97 that was added in response to Reviewer #3, made this section more confusing to me on the first read. While it may be obvious to some readers familiar with GNSS processing, others might not be immediately able to interpret that the GNSS signals are a result of mass (mainly liquid water) moving from locations distant to the GNSS stations to downstream locations closer to the stations. I think that emphasizing this mass transfer.

Authors: Thank you very much for your comments. There may be some misunderstandings, which we will clarify here. The *total* vertical GNSS signals reflect vertical displacements caused by *all* the processes affecting mass changes (e.g., solid ice discharge, liquid water change, post-glacial rebound, etc.) on or under the ice near the station. By correcting for nuisance signals, including those related to Surface Mass Balance (SMB) (see Extended Data Table 1), we obtain time-series of “*residual* vertical displacements”.

Note that SMB models account for water mass that is ultimately re-frozen in-place, but do not consider the mass of temporally buffered water that is discharged into the ocean (referred as BWS in this study). This is because RACMO, MAR, and other regional climate models do not simulate lateral water transport through hydrologic pathways and so they assume that meltwater that is not retained in the local firn layer (below the model grid cell where the meltwater was produced) is removed instantly from the ice sheet.

In reality, the part of meltwater not retained in the local firn is (gradually) transported through hydrologic pathways (i.e., the BWS in this study), temporally stored in various compartments of the ice sheet it encounters, such as remote snow/firn, moulines, lakes, basal water storage and groundwater storage below the ice sheet, and does not instantly move to the ocean. Consequently, a gradual increase in BWS at the beginning of the melt season leads to a downward signal in the *residual* bedrock displacements. The residual GNSS signal gradually reduces at the end and after the melt season, as meltwater is finally transferred into the ocean. We have included this explanation into the revised manuscript (lines 94-108).

- To make the processes under consideration unambiguously clear, maybe the authors could add a paragraph that explicitly discusses water transport mechanisms and how they may or may not contribute to the observed GNSS signals. For example, “We consider mass variations within ~200 km of GNSS stations arising from horizontal transport of meltwater within the glacier and groundwater systems. To produce detectable GNSS signals, we assume that water is transported tens of kms from distant, upstream source locations to downstream transient storage locations closer to GNSS stations. Transport mechanisms may include [list your hypothesized mechanisms]. [Clarify the processes, e.g. firn percolation and storage, local storage within supraglacial lake basins, etc. that likely don't include enough horizontal transport to produce detectable signals].”

Authors: Thank you very much for your comments. Similar to the clarification provided above (see pages 8-9 in the rebuttal), we have revised our manuscript accordingly. Please see lines 94-108 in the revised manuscript.

We briefly explain it again here. The SMB models extend downwards to the firn-ice boundary. They only account for local storage (by capillary retention, refreezing) of meltwater in the seasonal snow layer and firn column of that particular model grid cell. In the SMB models any meltwater that is not locally stored in the firn, is assumed to reach the ocean instantly. In reality, however, this meltwater (slowly) finds its way to the ocean through the ice sheet hydrological system. In the GNSS signal, this is visible first as a residual downward motion (as a result of the reduced upward motion) that slowly returns to zero when the

meltwater actually runs off. By taking the mean vertical displacements per calendar month at KAGA station as an example, we can clearly show the seasonal cycles of vertical displacements observed by a GNSS station and computed from an SMB model (see Fig. S1 below). After subtracting SMB data from GNSS data, the BWS signal is clearly visible.

Fig. S1 The mean vertical displacements per calendar month at KAGA station (after subtraction of linear trends). An SMB model assumes an instantaneous withdrawal of meltwater runoff, so that the vertical displacements modelled on its basis show an uplift starting from May (in green). In reality, meltwater transportation towards the ocean takes some time. As a result, GNSS data cleaned from nuisance signals (except for the SMB signal) show an uplift starting only from June (in red). The residual displacements computed as a difference between those two curves demonstrate a land subsidence caused by an accumulation and discharge of BWS (in cyan).

- L111: “In general, downward motion corresponds to an accumulation of stored water and vice versa. Therefore, we interpret this signal as water accumulation starting from the onset of the melt season (Fig. 1).” On first read, this was confusing: the “stored water” was always part of the GrIS mass (other than small contributions perhaps from temporary storage of rain in snow, firn, supraglacial lakes), so why should it cause downward motion of the GNSS stations?

Authors: Thank you very much for your comment. Please refer to the clarification provided above on pages 8-9 of the rebuttal. To further clarify the misunderstanding:

The BWS is indeed a component of the Greenland Ice Sheet (GrIS) mass; however, it is not accounted for by SMB models that describe variations in GrIS mass. As noted by Reviewer

#1, "RACMO, MAR, and other regional climate models do not simulate water transport through hydrologic pathways and so they instantly remove water from the ice sheet." SMB models assume that BWS is removed from the ice sheet and transported to the ocean instantly, with no delay.

In our analysis, when deriving residual vertical displacements, we correct for nuisance signals, including those related to Surface Mass Balance (SMB) (see Extended Data Table 1). This approach allows us to clearly identify the BWS signal in the residual vertical displacements (see Fig. 2). The BWS manifests itself only after model-based nuisance signals have been subtracted from the GNSS data.

We have modified the text in the revised manuscript (see Lines 122-126).

"In general, residual downward motion corresponds to an accumulation of stored water that is not local, i.e. not accounted for in the SMB models, and vice versa. Therefore, we interpret this signal as BWS accumulation within ~200 km around the GNSS station starting from the onset of the melt season (Fig. 1), which is unaccounted for in the SMB models."

On second read, I realize that this is because the assumption is that the "stored water" has been transferred from locations farther from the GNSS station to locations closer to the station (in particular, the transport distance are far enough to result in a measurable GNSS signal). I think that emphasizing the role of mass transfer here is very important given the broad audience.

Authors: Thank you very much for your comment. Please refer to the clarification provided above on pages 8-9 of the rebuttal. We have emphasized the inferred mass transfer in the introduction of the revised manuscript (see lines 94-108).

Minor comments and technical corrections

L52: I may be missing something, but it is not clear to me how runoff from rainfall increases GrIS mass loss?

Authors: Rainfall, like snowfall, is a component of Surface Mass Balance (SMB) and is included in the representation $P-SU-ER-R$, where P represents total precipitation (the sum of snowfall and rainfall), SU is sublimation, ER is the erosion of snow due to drifting snow transport divergence, and R is runoff (see lines 590-592). It is crucial to consider all these terms to accurately model variations in the Greenland Ice Sheet (GrIS) mass. Nonetheless, the contribution of rainfall to the precipitation in Greenland is minor (about 5-7%). As a large part of rainfall directly runoff over the ablation zone, having rainfall instead of snowfall does no more contribute to an accumulation of mass in this area. In the accumulation zone, rainfall is retained by the snowpack and refreezes during the following winter.

Rainfall increases surfaces runoff, but since the mass of the rain was not stored in the ice sheet, I'm not sure this should count as mass loss, which readers probably interpret as mass loss that contributes to sea-level change.

Authors: Mass of the rainfall may or may not be (temporarily) stored in the ablation zone and refreezes in the accumulation zone. It is the subject of SMB modelling to determine the fate of rainfall, depending on the location, meteorological conditions, and other factors. In the SMB modelling, rainfall on snow represents a mass gain and the vertical meltwater motion through firn is treated in the same way as meltwater. Rainfall on ice does not impact SMB in RACMO/MAR as it assumed to run off immediately while in reality this will also take time.

L68: Be consistent about "subglacial" vs. "sub-glacial"

Authors: Corrected. We consistently used "subglacial" in the revised manuscript. Please see lines 66, 70, and 257.

L134: Considering my first major comment, I would be more comfortable seeing an explanation along the lines of, “Importantly, the introduced analytic function takes into account possible inaccuracies in the proposed relationship between BWS and GNSS displacement and errors in the runoff estimated by the adopted SMB model”.

Authors: Corrected. Thank you very much for your comment. The cited statement is not present in the manuscript. The reviewer most likely refers to the statement, “Importantly, the introduced analytic function takes into account possible inaccuracies in the runoff estimated by the adopted SMB model” (lines 147-148 in the revised manuscript). We have added the clarification, “It is assumed that the true runoff is related to the SMB-modeled runoff by a factor.” Please see lines 148-149 in the revised manuscript.

In addition, interested readers are kindly referred to the further explanations in the "Method" section.

L167—169: The weak-to-moderate correlation (correlation 0.42) should be provided here.

Authors: We modified the text as follows: “We confirm the scaling factor dependence on summer temperatures by *showing a noticeable ($R=0.42$) correlation ...*” (lines 181-184).

L374: “It is sensible” to “The GNSS stations are sensitive”

Authors: Corrected. Please see line 386 in the revised manuscript.

L480: Why are supraglacial lakes considered here? Aren't they part of the BWS?

Authors: The reason to consider the supraglacial lakes is to evaluate their impact on the residual vertical displacements observed by GNSS stations. It is shown that the magnitude of loading signal caused by supraglacial lakes mass change is small, i.e., ~0.3 mm. Therefore, we conclude that this signal provides only a minor contribution to the BWS signal considered in this study. We have slightly adjusted the text in the manuscript to present our conclusion more explicitly (lines 508-511).

“Spatial sensitivity of GNSS loading data”: This information is key for understanding the manuscript, this should be better integrated into the main text.

Authors: Thank you very much. We have further clarified the GNSS signal in the introduction in the revised manuscript (please see Lines 94-108).

Figure 2: Why do some of the signals (e.g., QAAR, KAPI) appear non-periodic? I believe this is addressed deep within the methods and extended data, but it might help other readers to acknowledge this here.

Authors: Thank you very much for the comment. Apparently, the reviewer means multiple minima, which can be observed in the mean annual cycle of residual vertical displacements at some GNET stations. We also noticed them. In particular, there is a decrease or flatten pattern in January-February for QAAR, KAPI, etc. This could be caused, e.g., by wrong timing of snowfall and/or runoff in the SMB models. A further analysis of these features is beyond the scope of this study, which is focused on the signal caused by the BWS accumulation during the melt season.

Eq. A15 and the associated explanation: Related to my second major comment, I am uncomfortable with this explanation. As non-expert in GNSS signal processing and solid earth deformation, explanations in terms of the “surface load”, rather than in terms of the

horizontal redistribution of mass, are confusing to me. My concern could be removed by addressing my first major comment.

Authors: In response to these remarks, we have extended the last section of the manuscript to include an explicit relationship between the mass load and surface deformation (lines 737-751). Consequently, that section has been renamed “*Computation of elastic vertical deformations and spatial sensitivity of GNSS loading data*”. Furthermore, a reference to this relationship has been included in the section “*Analytic model of the BWS signal in GNSS elastic loading data* (lines 619-621)”:

“Technical details of the transformation of surface mass load into elastic vertical deformations can be found in Sect. *Computation of elastic vertical deformations and spatial sensitivity of GNSS loading data* below.”

This reference is included around Eq. (A9), and not around Eq. (A15) because it is Eq. (A9) where the aforementioned relationship is mentioned for the first time in Sect. *Analytic model of the BWS signal in GNSS elastic loading data*.

Referee #5 (Remarks on code availability):

I have reviewed but not attempted to run the provided code. The code structure is simple enough that installation should not be a concern. The code seems extremely specific to this application, so while the code may be used to verify the current results, it would be difficult to generalize the code to new analyses.

Authors: Thank you very much for your comments. We have included additional code pertaining to the raw input time series of GNSS, SMB, TEM, LWS, etc. Readers can investigate other ice areas by replacing these data with similar data from their GNSS stations. Please see the link: <https://github.com/RANjiangjun/data-for-processing/tree/main> .

Below is our response (shown in purple) to the comments of Reviewer #5 (shown in magenta), addressing our response (shown in light blue) to the concerns of Reviewer #3 (shown in black).

Comment 1.

Referee #3 (Remarks to the Author):

Reviewer # 5: Note: Additional evaluation of author responses by referee #5 shown in magenta. To keep the responses as concise as possible, I have not commented on responses regarding sections I have previously commented on or responses that I believe to be appropriate.

Authors: Thank you again for your suggestions. In the rebuttal below, we only address and quote the comments from reviewer #5 that are relevant and not already commented on or satisfied by the reviewer.

Comment 2.

Reviewer #3: Lines 69-70: “However, ice sheet regions with a channelised drainage system may alternate with regions lacking an efficient drainage, the latter retaining high hydrostatic pressure.” “Alternate” is not quite the right notion. The glacier bed and basal drainage system are highly heterogeneous in space, so even regions where channels are present could be dominated by high basal water pressure if the drainage system is not hydraulically well connected to the channels.

Authors: Thank you very much for your comment. The statement has been modified as follows: “Importantly, such drainage systems as well as the glacier bed are highly heterogeneous, so that even regions where channels are present could be dominated by high basal water pressure if the drainage system is hydraulically poorly connected to the channels.” Please see Lines 69-72 in the revised manuscript.

Referee #5: This clarification is important, but I think some additional context (i.e., one or two sentences) about the conceptual model of seasonal changes in subglacial drainage would be helpful before jumping straight to a subtle but important point about heterogeneous basal hydraulic connectivity.

Authors: We have extended the discussion in line with the reviewer's advice. The corresponding fragment of text now reads as follows:

“The subglacial component of BWS may affect ice flow velocities due to high basal water pressure it induces at the ice bed²⁵⁻²⁸. This creates a “lubrication effect”, i.e., a reduced friction between the ice layer and underlying rock. For this reason, a temporary increase in ice flow velocities may occur, particularly at the beginning of melt season (Zwally, 2002; Das et al, 2008; Schoof, 2010). Ultimately, the accumulation of water results in a formation of an efficient subglacial drainage system, which facilitates rapid water release into the ocean²⁹⁻³², so that the basal water pressure is reduced. Importantly, such drainage systems as well as the glacier bed are highly heterogeneous...”

Comment 3.

Reviewer #3: Lines 80-82: “...bedrock's elastic response to total mass variations in the vicinity of a given GNSS station”: This is a critical summary of the method and, in my view, should make the mechanism that relates the process to the observable clear for a broad readership: is GNSS mounted on bedrock outside of the ice sheet registering crustal flexure due to variable transient amounts of water mass being stored in the vicinity (all around the GNSS station, including on/in/under the ice)? Reporting the length-scale of influence/sensitivity would also be used here: over what radial distance/area is a given station GNSS sensitive? By what mechanisms do the different sources of GNSS signal contribute? For example, how extensive must the water redistribution be to be measurable? Is it correct that only horizontal redistribution matters? Some related information is presented deep within the extended figures, but a brief synopsis would help readers here. I felt there was a mechanistic gap throughout the paper, as it was assumed the reader would immediately understand that GNSS-measured bedrock elevation changes could be equated to the mass of water leaving the ice sheet. It would be appreciated if the authors would connect those dots more clearly.

Authors: Thank you very much for your comment.

As for the question “is GNSS mounted on bedrock outside of the ice sheet registering crustal flexure due to variable transient amounts of water mass being stored in the vicinity (all around the GNSS station, including on/in/under the ice)?”, the answer is

yes. The GNET stations were installed on the bedrock around the GrIS periphery. They are sensitive to all the processes which cause mass changes on/in/under the ice close to the station. We have included a statement to reflect this in the main text (see new Lines 374-375).

Referee #5: This is a critical clarification. It could be integrated more smoothly into the text, e.g. “We consider records from 22 Greenland GNSS Network (GNET) stations (Fig. 2), which contain information about the bedrock’s elastic response to total mass variations within ~200 km of a given GNSS station (see Spatial sensitivity of GNSS loading data)”

Authors: Corrected. Please see lines 94-97 in the revised manuscript.

Comment 4.

As for the question “Reporting the length-scale of influence/sensitivity would also be used here: over what radial distance/area is a given station GNSS sensitive?”, based on the sensitivity test in Lines 707-712, it reveals a significant contribution from the near field: mass changes within 200 km from the station contribute about 80% to the total loading displacements. There is little sensitivity to mass changes beyond 500 km from the station.

As for the question “By what mechanisms do the different sources of GNSS signal contribute? For example, how extensive must the water redistribution be to be measurable? Is it correct that only horizontal redistribution matters?”, the GNSS loading is based on pure elastic theory. The measurability depends on the magnitude and radial distance of vertically-integrated fluid masses to the GNSS station. Thereby, only horizontal redistribution of mass matters.

Referee #5: The authors have elsewhere in this document and in the extended data shown that the horizontal mass displacement must be on the order of tens of kms. This information should be added to the main text in question to address the reviewer’s comment. This would also help to address remaining reviewer concerns.

Authors: Thank you very much for your comment. We have clarified the misunderstanding in the introduction (please see Lines 94-108 in the revised manuscript). Please also refer to the responses on pages 8-9 of the rebuttal.

Comment 5.

Reviewer #: Line 95 cf. 88: Somewhat confusing at this point in the manuscript that GNSS data appear to be used to test SMB models (question iii, line 88), but that model output is being used to correct the GNSS signal (line 95). Can a sentence of clarification be introduced to dispel concern that there is circularity in the process? The scaling is reported as a major result in the opening paragraph of the paper, so this seems important.

Authors: Thank you for your valuable comment. Because the scaling factors applied to runoff estimates from SMB models are estimated simultaneously with parameters describing the buffered water storage, circularity is not an issue. To make this more clear in the text, we reformulated the third study objective as follows: “(iii) is it possible to better explain the BWS signal in GNSS data by scaling the runoff from SMB models?” (lines 99-100).

Referee #5: I believe the circularity the reviewer has highlighted still exists in the manuscript. The uncalibrated RACMO runoff data is used to remove the SMB signal from the GNSS signal, then RACMO scaling factors are derived by fitting modelled uplift to the cleaned GNSS signal.

Authors: Thank you very much for your comment. A circularity would be indeed an issue if different unknown parameters were estimated sequentially (for example, BWS-related parameters in the first instance and the scaling factors thereafter). However, the adopted data inversion scheme is different. All the unknown parameters are estimated simultaneously, not sequentially (see the functional model given by Eq. (A18)). This concerns both the corrections ϵ_k applied to scale the RACMO-based runoff estimates and the BWS-related

parameters (namely, water storage time T_{st} and empirical coefficient θ). Therefore, the adopted data inversion scheme does not have an issue of circularity.

Comment 6.

Reviewer #3: Lines 97-102: It would help if stated explicitly that downward motion equates to accumulation of stored water and vice versa, presumably due to proximity of GNSS to locations of accumulation. I'm thinking about mass changes in general and how station position relative to the mass change might lead to downward (proximal for mass accumulation) vs upward (distal from mass accumulation) responses.

Authors: Thank you very much for your comment. In the revised version, we now state this explicitly in the lines 111-113 as "In general, downward motion corresponds to accumulation of stored water and vice versa."

Referee #5: This is a helpful clarification. However, the second sentence, "Therefore, we interpret the signal as water accumulation starting from the onset of the melt season", could be further clarified by referencing the GNSS sensitivity footprint and horizontal mass transport. I have repeatedly suggested this change since I believe it is in the spirit of the previous reviewer's comments

Authors: We appreciate the comments from the reviewer. We modified the text into "Therefore, we interpret this signal as BWS accumulation within ~200 km around the GNSS station starting from the onset of the melt season (Fig. 1), which is unaccounted for in the SMB models." (please see lines 124-126 in the revised manuscript). As for the horizontal mass transport issue, please kindly referred P8-9 and P31-32 in the rebuttal.

Comment 7.

Reviewer #3: Line 124: "tuned" may be better than "calibrated" since this is an empirical scaling

Authors: Thank you very much for your comment. Based on the in situ GNSS data, we evaluate the runoff modelled by RACMO 2.3p2. We contend that the calculated factors reveal an actual deficiency in the modelled runoff. Applying these factors makes the modelled runoff more realistic (see lines 152-169 in the main text). Therefore, in this case, we prefer to use the term “calibrated”. However, if the reviewer believes that “tuned” is a better word, we are fully open and happy to adopt this suggestion.

Referee #5: In line with my first major comment, I am strongly on the side of Reviewer #3 here. Without more evidence, I am not convinced the SMB scale factors represent actual deficiency of the RACMO modelled runoff, and so the wording around the scale factors should be much more careful.

Authors: Please see our response to the first major comment of reviewer #5 on page 4-8 in the rebuttal).

Comment 8.

Reviewer #3: Line 128-129: “total BWS”. Doesn’t the mass need to move substantially from its place of origin to cause crustal elevation change (i.e., meltwater that accumulates locally in a supraglacial lake would not change the mass distribution substantially enough to be detected in the form of bedrock elevation changes outside the ice sheet)?

Authors: Thank you very much for your comment. If meltwater is stored locally (near the location where it was produced, e.g., meltwater that accumulates locally in a supraglacial lake), this, most probably, will not affect the GNSS signal. We designed a test and find that horizontal mass displacement must be in the order of tens of km to emerge in elastic loading data. For more details, we refer to our answer to additional comments of Reviewer 3 to our response to Reviewer 2 from the previous round (please see Fig. S2 in P24 in the last part of this rebuttal letter).

Referee #5: I don’t believe this comment has been appropriately addressed in the manuscript text. At this point in the text, it seems that “BWS” is implicitly assumed to

mean “Buffered water storage within ~200 km of the GNSS station that has been sourced from tens of kms away from the storage location, such that it induces a detectable GNSS signal”. However, I (and, it seems, the previous reviewers) interpret BWS to mean the total water mass transiently stored within the glacier and groundwater system, precisely as defined by Eq. A10. Without reference to horizontal transport of mass from distant to nearby locations, the Eq. A10 definition of BWS doesn’t necessarily imply a bedrock elastic response. This needs to be clarified in the manuscript text.

Authors: Thank you very much for your comment. There may be some misunderstanding here. The GNSS signal only reveals the integrated mass variations within approximately 200 km of each GNSS station. However, it cannot distinguish the spatial distribution within this sensitivity radius. The GNSS stations are installed along the coastlines of Greenland (see Fig. 2), and the mean distance of the GNSS stations to the ice sheet margin is ~25 km (see Extended Data Table 3). When the BWS discharges to the ocean, the resulting upward motion in residual vertical displacements can be possibly interpreted as the lateral BWS movement. Please refer to the response on pages 8-9 in the rebuttal.

Comment 9.

Reviewer #3: I still have questions about the treatment of groundwater in the paper, though the authors have done what seems to be an extensive analysis with multiple models. Thematically consistent with the above, it would be useful to hear something about physical processes and what anticipated groundwater processes would contribute to GNSS signals, as well as a bit more detail on the possible limitations of the treatment of groundwater in the two models used to estimate the contributions to deformation.

Authors: Thank you very much for your comment. In this study, two types of groundwater can be identified:

- 1) Shallow groundwater in tundra areas, which results from snow melting and rainfalls there. In principle, this is an ordinary component of the Terrestrial Water Storage, which is described by various hydrological models, including those addressed

in the manuscript (see Extended Data Fig. 13 modelled by PCR-GLOBWB and WGHM). Based on the personal communication with Prof. Mark Bierkens, who is a developer of PCR-GLOBWB, it is very difficult to model the groundwater accurately in those regions, but the model outcome is enough for a first-order estimate of its magnitude. In this study, we analyze the uncertainty of groundwater estimates in Sect. *Uncertainties of the mean annual cycle of residual vertical displacements* (see Lines 422-428 in the manuscript). We take KAGA as a representative station, and quantify the difference between mean annual vertical displacements per calendar month with and without groundwater signal subtracted from GNSS data using PCR-GLOBWB and WGHM models. This is basically the modelled groundwater signal itself. It shows that the RMS signal from PCR-GLOBWB and WGHM model is 0.25 mm and 0.02 mm, respectively (see Fig. S1), which is small, as compared to the BWS signal (see Extended Data Fig. 5). We have included this information in Lines 422-428 of the revised manuscript.

2) Deep groundwater below the ice sheet (its presence was detected by a hydrological well down to the depth of hundreds of meters⁶²). It is a product of ice sheet melting (both at the surface and the ice sheet base). To the best of our knowledge, little is known about variations in the deep groundwater mass. Here, we consider the signal from deep groundwater as a part of the total BWS signal we detect in GNSS data. Unfortunately, elastic loading data do not allow the deep groundwater to be separated from the rest of the BWS.

I am pleased to see that the authors have clarified the two categories of groundwater and that category (2) is included as a possible contributor to the BWS GNSS signal. However, given the questionable representation of all the processes related to (1) in the groundwater models, perhaps it is best to explain that any processes in (1) missing from the models would also be a part of the GNSS signal (see page 45).

Authors: In principle, the reviewer is correct in stating that any processes missing from the models would be part of the signal observed in residual vertical displacements. However, our

goal is to demonstrate that this signal is predominantly caused by BWS, as the contribution of other conceivable processes is minor.

Comment 10.

...

Below is our response (shown in light blue) to the comments of Reviewer #3 (shown in red), which addressed our response (shown in blue) to the concerns of Reviewer #2 (shown in black).

....

Reviewer #2: -- Figure 1 depicts englacial lakes (and subglacial lakes that are actually shown as englacial) — I know of no evidence for lakes existing inside the ice.

Authors: Thank you very much for your comment, and we apologize for the misunderstanding. Figure 1 is a schematic representation designed to illustrate the processes related to liquid water evolution beneath the ice sheet surface and the associated changes in vertical bedrock. The distance of englacial lakes from the ice sheet surface in the figure is schematic and intended to indicate that some water is buffered beneath the ice sheet surface; it does not represent the actual location of englacial lakes within the ice sheet.

In response to the feedback, we have revised Figure 1 to avoid any further misunderstandings. For your convenience, the updated Figure 1 is shown below. Please refer to lines 218-232 in the revised manuscript for details.

Reviewer #3: The representation of relevant hydrological features in the new Figure 1 seems fine to me, with the exception that differences in the relationship between the white dashed line and the bedrock mound on which the GNSS station is mounted are hard to discern in the 3 panels.

Authors: Thank you very much for your agreement on the revised Figure 1. In the first panel, the dashed line coincides with the boundary of the ice-bedrock interface, as vertical deformation has not yet started. However, in panels b and c, the deformation is

non-zero. In response to the reviewer's comment, we have increased the difference between the white dashed line and the ice-bedrock interface by approximately 1.5 times (see Fig. 1), compared to the previous version of the plot.

Referee #5: The caption makes me expect that the separation between actual and modelled (white dashed line) bedrock should be highest in Early Summer and lower in Late Summer, but to me it looks like the separation is greatest in Late Summer? Could the 1.5x increase in separation between bedrock and dashed line be further increased to 2—3x to make it unambiguous?

Authors: Thank you again for your suggestions. It reaches the maximum in Late Summer. The separation between bedrock and dashed line has been further increased to 2.5x in the revised Fig.1.

Comment 11.

Reviewer #2: -- The text concerning loading is confusing because a phase change is not a mass change, so water mass loading must be due to the redistribution of water from one place to another (not stated).

Authors: Thanks for this thoughtful comment. The “phase change” refers to melting and refreezing processes. In order to avoid potential confusion, we have added the following sentence in the introductory section (Lines 84-85):

“Note that bedrock displacement time-series describe the mass of buffered water rather than the total water mass, since surface loading does not change if water re-freezes in-place.”

Reviewer #3: This clarification is helpful, but I also struggled with the lack of clarity around water mass change: the mass needs to move in order to elicit elastic deformation. Throughout the text, it almost seemed like the GNSS signals were being related to water mass, rather than a change in water mass (see also my independent comments).

Authors: Thank you very much for your comment. The reviewer is correct that the GNSS measures the elastic deformation caused by the change in water mass. This is fully consistent with the manuscript. We clean GNSS data from nuisance signals to obtain residual deformations that are referred to *the Buffered Water Storage (BWS)*. The latter is defined as “the water mass temporarily buffered within the GrIS, being mostly released into the ocean before the onset of next melt season” (lines 64-65). To make the relationship between water mass and vertical deformations even more clear for readers, we have extended our clarification cited above (in blue) as follows: “Note that resulting bedrock displacements peak during the melt season and gradually reduce thereafter. **This is because they describe BWS rather than the total water mass (surface loading does not change if water re-freezes in-place)**” (lines 95-98).

Referee #5: Following previous comments, I don't believe this concern has been appropriately addressed.

Regarding the highlighted text, would it be more clear to say, “This is because bedrock displacement describe mass redistribution from distant, upstream source locations to downstream BWS locations closer to the GNSS stations”?

Authors: Thank you very much for your comments. We have clarified this and revised the manuscript (please see P8-9 in the rebuttal and Lines 94-108 in the manuscript).

Comment 12.

Reviewer #2: -- The manuscript intro/discussion is structured around water storage in the firn compartment, but why this is driving the loading signal versus other compartments is not clear. Melt across most of the firn area will result in a short-distance mass redistribution, vertically within the upper 10 m. Only a relatively narrow reach of the far lower accumulation area has substantial horizontal flow and accumulation of water, but is there evidence that this is enough to drive the loading signal? Regardless, all runoff is accumulated toward the margin. Whereas a SMB

model may simulate a large amount of meltwater generation across the accumulation zone, the subglacial drainage system, not the firn compartment, has the most accumulating water mass and is therefore the most likely candidate for loading. Water at the bed cannot be dismissed by invoking fast conduit drainage. A large body of literature on subglacial hydrology debates issues such as a) how far inward on the ice sheet fast-draining conduits can actually develop (e.g., Werder et al., 2013); and, b) how much area of the bed sits between conduits and does not drain quickly (e.g., Hoffman et al., 2016).

Authors: We agree that the introductory section focused too much on processes in the firn layer, which resulted in some mismatch between that section and the rest of the manuscript. To solve this issue, we have re-organized the introduction to put more emphasis on water at the bed (Lines 65-76). The firn layer is mentioned in the Discussion Section only briefly, and primarily in the context of our interpretation of the scaling factors to be applied to runoff estimates provided by the SMB model (Lines 189-199).

Reviewer #3: Firn processes did not seem over-emphasized in the revised manuscript. This point, however, also relates to the lengthscales over which mass must move to create detectable deformation.

Authors: We are pleased to see that we reached a consensus concerning the text about the firn processes. For the issue of the length scale over which mass must move to create detectable deformation, we designed a sensitivity analysis which is similar to Fig.1 in Wahr et al., (2013) and Fig. 6 in Bevis et al., (2016). By considering a mass formed by a 1-m water layer over a disk of 20 km radius, we calculated the deformation at different distances from the disk centre (Fig. S2). This plot clearly shows that horizontal displacement of a mass must be of the order of at least tens of km to be seen in elastic loading data. Thus, a local re-distribution of meltwater within the firn layer can be safely ignored. We have included the clarification in Lines 707-712.

Fig. S2 Deformation caused by a load of 1-m water disk of 20 km radius at different distances from the disk centre. This result is consistent with Figure 1 by Wahr et al. (2013).

Referee #5: This sensitivity experiment helps to clarify the question of how far mass must be transported to produce a measurable vertical bedrock displacement. However, the authors have not addressed whether or not a horizontal displacement of tens of kms is reasonable for the proposed BWS components (supraglacial, englacial, subglacial, groundwater beneath the ice sheet) with reference to appropriate literature. A summary of this information should also be integrated into the main text.

Authors: Thank you very much for your comments. Similar to the clarification in lines 94-108 in the manuscript and P8-9 in the rebuttal, we briefly explain it here. This signal is detected due to a production of BWS (the mass of which is not a part of SMB), and is not due to a horizontal transportation of water into the sensitivity radius of GNSS data. In other words, that signal is detected because the sensitivity radius of GNSS data is sufficiently large, and not because it is sufficiently small. In order to demonstrate better that that sensitivity radius is indeed sufficiently large (of the order of 200 km), we have updated the aforementioned plot:

Extended Data Fig. 15 | Deformations caused by a load of disks of several radii at different distances from the disk centre. The thickness of each disk is defined in terms of EWH such that the deformation at its center reaches 5 mm (namely, 132 cm, 53 cm, 35 cm, and 28 cm for the disk of 20-km, 80-km, 140-km, and 200-km radius, respectively). The corresponding disk masses are 1.7, 11, 22, and 35 Gt. These results are consistent with Figure 1 by Wahr et al.⁸⁴.

The associated discussion in the last section has been re-written as follows (lines 752-767):
 “To demonstrate a possible spatial variability of elastic vertical deformations, we have considered several surface mass loads, each of which is homogeneously distributed over a disk of a given radius. The thickness of each disk is defined in terms of Equivalent Water Height (EWH) such that the deformation at its center reaches 5 mm, which is similar to the magnitude observed in real data (Fig.2). One can see (Extended Data Fig. 15) that the spatial variability of the resulting deformations strongly depends on the spatial extent of the surface load. We believe that a disk of 200-km (or larger) radius, with a total mass of (at least) 35 Gt, gives the best approximation of the actual surface load distribution. This is because the total BWS mass per drainage system estimated from GRACE satellite gravimetry data is of the order of 20-40 Gt³⁴, whereas the shape of the actual surface load distribution likely resembles half-a-disk rather than a disk (most of GNSS stations are located near the coast, whereas the surface load over the ocean is nearly constant). This simple example also shows

that the position of any realistic surface load relatively to a given observation point must move laterally by at least a few tens of km in order to change the elastic vertical deformation at the observation point substantially. This means, for instance, that a local re-distribution of meltwater within the firn layer cannot affect deformations observed at a given GNSS station.”

Comment 13.

Reviewer #3: Without knowing enough to have confidence in the models' treatment of groundwater around Greenland, I am hesitant to dismiss this correction as negligible. Depending on what the models are doing, one solution may be not to attempt to correct for groundwater or claim it is negligible, but rather include it in what is being measured. If most of the groundwater recharge is coming from the ice sheet, then this is a source of water that would ideally be accounted for anyway.

Authors: Thank you very much for your comment. We fully agree with the reviewer that groundwater resulting from ice sheet melt should be considered part of the BWS signal in this study. A similar suggestion was also made by Reviewer 4. We have revised the main text to clarify this aspect (see lines 501-516). Importantly, the discussion above specifically addresses groundwater variability estimated by hydrological models, which pertains to shallow groundwater originating from precipitation outside the ice sheet..

Referee #5: I think this clarification meaningfully addresses part of the reviewer's comments here. Including groundwater beneath the GrIS as a BWS component seems important.

Authors: Yes, we agree. This was already briefly acknowledged at the beginning of the introduction and further detailed in Sect. *Contribution of groundwater storage in Greenland*. However, groundwater was not mentioned so far in the list of water storage components in Sect. *Quantification of BWS*. This has been fixed: groundwater storage below the ice sheet has been added (line 157).

It seems that the models and products available to evaluate LWS are not as detailed as the authors and reviewers would like, leaving some open questions (page 45).

Authors: Thank you very much for your comment. Indeed, the information content of models available to evaluate LWS is limited. Firstly, they do not cover groundwater below the ice sheet, so that they can only be applied to shallow groundwater in tundra areas at the Greenland coast. Secondly, the accuracy of those models is limited and hardly allows one to do more than a first-order estimate of groundwater signal magnitude (lines 525-529). Nevertheless, the output provided by those models allowed us to conclude that the contribution of groundwater in tundra to the observed signal in residual vertical displacements is minor (lines 438-446). All that was already stated in the previous variant of the manuscript.

Comment 14.

4) As far as the study by Liljedahl et al. (2021) is concerned, we would like to stress that it reveals variations in water pressure in a hydrological well below an ice sheet margin at the depth of at least 400 m. At the seasonal time scale, these variations are indeed significant (the peak-to-peak amplitude is about 15 m in terms of water head). However, it is important to understand that there is no one-to-one relationship between those pressure variations and variations in water mass. In opposite, those pressure variations systematically show a minimum in late autumn, and then steadily increase in the course of winter and spring, reaching the maximum in early summer. Therefore, as it has been stated by Liljedahl et al. (2021), that the pressure variations cannot be explained by an accumulation of meltwater, since water mass does not increase in the course of winter season. Liljedahl et al. (2021) attribute the revealed variability to “fluid pressure changes at the ice/earth boundary”. Unfortunately, the mechanism of those changes at the ice/earth boundary is not addressed explicitly. In any case, it would be fair to conclude that Liljedahl et al. (2021) show no evidences of a significant impact of groundwater on rock displacements observed at the surface.

Reviewer 3: This paper may plausibly be all the authors have to go on, but the lag times for groundwater transport can mean that hydraulic head variations can be out of phase with

recharge from distal locations. The phase would also vary with depth due to the variation in travel pathlength and residence time with depth. Closer to the surface than the borehole described above, phase lags between recharge and hydraulic head could be much shorter. This makes me skeptical about dismissing groundwater as a signal source based on the arguments above.

Authors: As stated above, groundwater originating within the ice sheet is indeed included as part of the BWS signal detected by GPS data. However, it is important to note that the elastic deformations measured by GPS data reflect variations in the total surface load around each GPS station. These measurements do not allow for the isolation of individual water storage compartments.

Referee #5: I think the reviewers are correct to suggest that the definition of groundwater that contributes to the GNSS BWS signal should be expanded. If I am interpreting correctly, all groundwater components that are not included as part of the LWS correction would contribute to the BWS signal. This should be added to Line 143 in the text. Instead of restricting to only “groundwater stored beneath the GrIS”, the BWS variations represent all changes in groundwater storage that have not been removed by the LWS correction.

Authors: Thank you very much for your comment. As it is mentioned above, the sentence addressed by the reviewer is changed into “*These estimates account for variations in the total BWS, i.e., water stored in all ice sheet compartments, including snow/firn, moulins, lakes, basal water storage, as well as groundwater storage below the ice sheet.*” Please see lines 155-157 in the revised manuscript. We suggest not to mention the LWS correction in that sentence, since the term “LWS correction” is not known to the reader at this point, and we cannot explain that term without going too far into technical details.

Comment 15.

Reviewer #2: Storage—

Seasonal storage of meltwater is well-established. For example, prior work has measured time series of ‘bed separation,’ the local water storage in the subglacial

drainage system (e.g., Andrews et al. 2014). These studies have the advantage of pinpointing the storage location but exist for only a few points and time periods. Other work has used radar to identify the spatial variability of water storage at the bed over 10s to 100s of km length scales (Chu et al., 2016). Ran et al. (2018) used GRACE data to show a time delay, peaking in midsummer, between RACMO's meltwater generation and total mass loss (i.e., water storage). The finding is required since RACMO, MAR, and other regional climate models do not simulate water transport through hydrologic pathways and so they instantly remove water from the ice sheet. Thus, the disconnect peaks midsummer when meltwater generation is at maximum.

Authors: Previous studies have estimated "local water storage" using in-situ measurements (Andrews et al., 2014), assessed the extent of bed water with radar data (Chu et al., 2016), and analyzed large-scale water mass variations with GRACE data (Ran et al., 2018), all of which are acknowledged in the revised introductory section (lines 65-76). It is noteworthy that Ran et al. (2018) is one of our previous papers.

This study is the first to demonstrate that GNSS-based vertical bedrock displacement can be used to observe and understand hydrological processes within the Greenland Ice Sheet (GrIS) and potentially other ice bodies on Earth. Furthermore, our study provides the first quantitative estimates of buffered water accumulation and release. Our results have significant implications for modeling the Surface Mass Balance (SMB) of the GrIS. We show that current SMB model runoff may contain systematic errors, requiring additional scaling of up to approximately 20% for the warmest years, particularly in the northern and northeastern GrIS regions. Accurate modeling of SMB behavior during warm years is crucial for making reliable projections of the future behavior of the GrIS, especially as the climate warms. Given that the GrIS is a major contributor to future sea level rise, improving these projections has substantial geographic, economic, and societal implications.

The fact that regional climate models do not simulate water transport and thus omit the signal of "buffered water" is central to our study. This issue is the foundation of our discussion in the section "Analytic Model of the Buffered Water Signal in GNSS Elastic Loading Data."

Reviewer #3: I think the study is novel in using surface deformation to make inferences about seasonal changes in water-mass distribution, provided sufficient confidence can be established in the non-hydrological corrections to the signal. The revised manuscript text and responses to the reviewer still leave some mechanistic gaps for me: what presumably matters for vertical elastic deformation measurable at the GNSS stations is sufficient lateral mass movement within the radius of sensitivity. How each of the appearances and disappearances of “observable” mass translate into measurable redistributions of mass that contribute to the GNSS signal is not always clear. The paper would benefit from a clearer and more fulsome articulation of the physical processes involved, with attention to distinguishing between processes that would and would not contribute to the GNSS signal, including an explanation for the sign of the contribution.

Authors: Thank you very much for your comment. We have added a paragraph that describes how vertical elastic deformation measurable at the GNSS stations can be used to detect lateral mass movement. The theory behind elastic deformation is well known and published by Farrell in 1972. Many publications have used this theory to study e.g. change in the distribution of ice, snow, water, or atmospheric mass. By monitoring the deformation with GNSS, it is possible to place constraints on the amplitude of the change in mass and the location of that mass (Barletta 2024, Wahr 2013, Hansen 2021). Here we describe the basic principle and demonstrate for the first time how GNSS can be used to place constraints on the change in mass due to englacial water accumulation, storage and ultimate release.

Referee #5: The authors have made some effort to address the “mechanistic gap” explained by the reviewer but more explanations are needed. No matter how well established the theory is, the explanations in the current work need to be clear and thorough to the reader and stand alone without requiring expert knowledge of the theory. My comments throughout should highlight areas that this should be improved.

Authors: Thank you very much for your comment. We have made numerous modifications to the manuscript based on the suggestions from Reviewer #5. Specifically:

1. **Clarification of BWS Signal Attribution:** We have improved our explanation of why we attribute the signal detected in residual vertical displacements to the accumulation of BWS and its subsequent discharge into the ocean (see lines 94-108).
2. **Extended Discussion on BWS Impact:** We expanded the discussion in the section "Analytic Model of the BWS Signal in GNSS Elastic Loading Data" to demonstrate that BWS-related features have only a minor impact on the estimated scaling of RACMO-based runoff (see lines 699-722).
3. **Enhanced Explanation of Elastic Vertical Deformations:** We renamed and extended the section "Spatial Sensitivity of GNSS Loading Data" to "Computation of Elastic Vertical Deformations and Spatial Sensitivity of GNSS Loading Data." This updated section now includes explicit mathematical expressions describing the relationship between surface load and resulting elastic vertical displacements (see lines 739-751).
4. **Updated Figures and Discussions:** We updated Extended Data Figure 15 and revised the associated discussion to better illustrate that the sensitivity radius of elastic displacements measured by GNSS stations is sufficiently large (~200 km).

These revisions address the reviewer's concerns and aim to provide a clearer and more comprehensive understanding of the study.

Reviewer Reports on the Third Revision:

Referee #5 (Remarks to the Author):

General comments

Thank you for your detailed responses to my comments on the previous version. The additional text that has been added to the introduction (L94—108) significantly clarifies the physical intuition behind the “residual” signal that you are isolating.

Previously, my main comment related to the conclusion that SMB models systematically underestimate runoff by 20% in warm years based on the annual scale factors derived by this method. In your response and the updated methods, you have clarified the meaning of each term in the forward model, but I am still not entirely convinced that the conclusions should be stated so strongly. I would be more comfortable if you could soften the conclusions by acknowledging that there must be some uncertainty in your inversions. Perhaps some of the 20% scaling could be attributed to complex hydrologic process which are, understandably, not considered in the methods here. I do agree that your results provide some evidence that SMB may be underestimated in warm years, and this should be investigated in more detail in future studies.

Statistics and uncertainties

- Fig. 2: Can you briefly (in a few words) elaborate on “The red shadowing depicts the one-sigma uncertainty”? I.e., something along the lines of “The red shadowing depicts the one-sigma uncertainty from 100 Monte Carlo runs”? The same shading should be defined for Extended Data Fig. 6.
- Fig. 4: “Uncertainties from 100 Monte Carlo runs are shown as shadowed zones”. By “shadowed zones”, do you mean the error bars? The font and markers are also small and difficult to read.
- Extended Data Table 2: What are the listed \pm values?
- Extended Data Fig. 3: Please define the error bars in panel (d). The font and markers in this figure are also small and difficult to read. Have you tested the hypothesis that there is a linear relationship between the scaling factor and temperature anomaly (i.e., computed the p-value)?

Author Rebuttals to Third Revision:

Color code: original (black), our replies (light blue)

Referee #5 (Remarks to the Author):

General comments

Thank you for your detailed responses to my comments on the previous version. The additional text that has been added to the introduction (L94—108) significantly clarifies the physical intuition behind the “residual” signal that you are isolating.

Authors: Thank you very much for taking the time to review our paper and for your excellent comments. We appreciate your encouraging feedback and are grateful for your valuable suggestions, which we have addressed one by one. Our responses are shown below in light blue.

Previously, my main comment related to the conclusion that SMB models systematically underestimate runoff by 20% in warm years based on the annual scale factors derived by this method. In your response and the updated methods, you have clarified the meaning of each term in the forward model, but I am still not entirely convinced that the conclusions should be stated so strongly. I would be more comfortable if you could soften the conclusions by acknowledging that there must be some uncertainty in your inversions. Perhaps some of the 20% scaling could be attributed to complex hydrologic process which are, understandably, not considered in the methods here. I do agree that your results provide some evidence that SMB may be underestimated in warm years, and this should be investigated in more detail in future studies.

Authors: Thank you very much. We have acknowledged that there are uncertainties in the estimated scaling factors and have done our best to quantify them. It is worth to note that an assumption that 20% of the scaling factor may reflect a “complex hydrologic process” means that this hypothetical process may explain all the scaling [this is because $1.2 \pm 20\%$ gives a range from $1.2 \cdot (1 - 0.2) = 0.96$ to $1.2 \cdot (1 + 0.2) = 1.44$]. This is unlikely because hydrological processes play an only minor role in the course of winter (several months after the melt season). At the same time, we have already shown that the usage of only winter

months as input has only a minor impact onto the estimated scaling factors (see Extended Data Table 4 and the associated discussion). This finding can be understood from the fact that we scale the modelled runoff, not a water mass. A scaling of the runoff affects the total mass of the GrIS in a long-term run (e.g., more runoff during the melt season implies a lower mass of the GrIS, and this effect sustains in the winter to follow). Thus, we decided not to mention a hypothetical “complex hydrologic process” as a possible cause of the adopted scaling.

Statistics and uncertainties

- Fig. 2: Can you briefly (in a few words) elaborate on “The red shadowing depicts the one-sigma uncertainty”? I.e., something along the lines of “The red shadowing depicts the one-sigma uncertainty from 100 Monte Carlo runs”? The same shading should be defined for Extended Data Fig. 6.

Authors: Thank you very much for the comment. The red shadowing depicts the one-sigma uncertainty which is computed as the root-sum-square of the standard deviations of noise from all possible sources considered in this study, i.e., GNSS data and models of ATM, NTOL, LWS, SMB, TEM, and GWS signals. Please see the section *Uncertainties of the mean annual cycle of residual vertical displacements* in Lines 416-421 of the main text. The same is applied to Extended Data Fig. 6.

- Fig. 4: “Uncertainties from 100 Monte Carlo runs are shown as shadowed zones”. By “shadowed zones”, do you mean the error bars? The font and markers are also small and difficult to read.

Authors: Thank you very much for the valuable comments. The “shadowed zones” is modified to “error bars”. We have re-plotted all the figures to make them clear.

- Extended Data Table 2: What are the listed \pm values?

Authors: Thank you very much for the comments. The uncertainties are computed from 100 Monte Carlo runs to describe the meltwater storage within GrIS. We have clarified it in Lines 922-924 in the main text.

- Extended Data Fig. 3: Please define the error bars in panel (d). The font and markers in this figure are also small and difficult to read. Have you tested the hypothesis that there is a linear relationship between the scaling factor and temperature anomaly (i.e., computed the p-value)?

Authors: Thank you very much for the comments. Uncertainties from 100 Monte Carlo runs are shown as error bars in Extended Data Fig. 3. We have increased the font and marker in the revised main text. We have tested the hypothesis of the linear relationship and found that the observed slope of the blue line is statistically-significant (the two-tailed p-value is $2.7 \cdot 10^{-9}$). Please see Lines 947-949 in the main text.